# Learning to Abstain from uninformative data

## Abstract

Learning and decision making in domains with naturally high noise-to-signal ratios – such as Finance or Healthcare – can be challenging yet extremely important. In this paper, we study a problem of learning and decision making under a general noisy generative process. The distribution has a significant proportion of uninformative data with high noise in label, while part of the data contains useful information represented by low label noise. This dichotomy is present during both training and inference, which requires the proper handling of uninformative data at testing time. We propose a novel approach to learn under these conditions via a loss inspired by the selective learning theory. By minimizing the loss, the model is guaranteed to make a near-optimal decision by distinguishing informative data from the uninformative data and making predictions. We build upon the strength of our theoretical guarantees by describing an iterative algorithm, which jointly optimizes both a predictor and a selector, and evaluate its empirical performance under a variety of settings.

## 1 Introduction

Despite the success of machine learning in computer vision (Krizhevsky et al., 2009; He et al., 2016a; Huang et al., 2017) and natural language processing (Vaswani et al., 2017; Devlin et al., 2018), the power of ML is yet to make significant impact in other areas such as finance and public health. One major challenge is the inherently high noise-to-signal ratio in certain domains. In financial statistical arbitrage, the spread between two assets are usually modeled using Orstein-Uhlembeck processes (Øksendal, 2003; Avellaneda & Lee, 2010). Spread behaves almost purely random near zero and are naturally unpredictable. They become predictable in certain rare pockets/scenarios. For example, when spread exceeds certain threshold, with high probability it will move toward zero, making arbitrage possible. In cancer research, due to limited resources, only a small number of the most popular gene mutations are routinely tested for differential diagnosis and prognosis. However, due to the long tail distribution of mutation frequencies across genes, these popular gene mutations can only capture a small proportion of the relevant list of driver mutations of a patient (Reddy et al., 2017). For a significant number of patients, the tested gene mutations may not be in the relevant list of driver mutations and its relationship w.r.t. the outcome may appear completely random. Identifying these patients automatically will justify additional gene mutation testing.

These high noise-to-signal ratio datasets pose new challenges to learning. New methods are required to deal with large fraction of uninformative/high-noise data in both training and testing stages. The source of uninformative data can be either due to the random nature of the data generating process, or due to the fact that the real causing factor is not captured during data collection. Direct application of standard supervised learning methods to such datasets is both challenging and unwarranted. Deep neural networks are even more affected by the presence of noise, due to their strong memorization power (Zhang et al., 2017a): they are likely to overfit the noise and make overly confident predictions where weak/no real structure exists.

In this paper, we propose a novel method for learning on datasets where a significant portion of content has high noise. Instead of forcing the classifier to make predictions for every sample, we learn to decide whether a datapoint is informative or not. Our idea is inspired by the classic selective prediction problem (Chow, 1957), in which one learns to select a subset of the data and only predict on that subset. However, the goal of selective prediction is very different from ours. A selective prediction method pursues a balance between coverage (i.e. proportion of the data selected) and conditional accuracy on the selected data, and does not explicitly model the underlying generative

process. In particular, the aforementioned balance needs to be specified by a human expert, as opposed to being derived directly from the data. In our problem, we assume that uninformative data is an integral part of the underlying generative process and needs to be accounted for. By definition, no learning method, no matter how powerful, can successfully make predictions on uninformative data. Our goal is therefore to identify these uninformative/high noise samples, and at the same time, to train a classifier that suffers less from the noisy data.

Our method learns a *selector*, $g$, to approximate the optimal indicator function of informative data, $g^*$. We assume that $g^*$ exists as a part of the data generation process, but it is never revealed to us, even during training. Instead of direct supervision, we therefore must rely on the predictor's mistakes to train the selector. To achieve this goal, we propose a novel *selector loss* enforcing that (1) the selected data best fits the predictor, and (2) the portion of the data where we abstain from forecasting, does not contain many correct predictions. This loss function is quite different from the loss in classic selective prediction, which penalizes all unselected data equally.

We theoretically analyze our method under a general noisy data generation process which follows the standard data dependent label noise model (Massart & Nédélec, 2006; Hanneke, 2009). We distinguish informative/uninformative data via a gap in label noise ratio. A major contribution of this paper is the derivation of theoretical guarantees for the empirical minimizer of our loss. A minimax-optimal sample complexity bound for approximating the optimal selector is provided. We show that optimizing the selector loss can recover nearly all the informative data in a PAC fashion (Valiant, 1984). This guarantee holds even in a challenging setting where the uninformative data has purely random labels, and dominates the training set.

This theoretical guarantee empowers us to expand to a more realistic setting where sample size is limited, and the initial predictor is not sufficiently close to the ground truth. Our method extends to an iterative algorithm, in which both the predictor and the selector are progressively optimized. The selector is improved by optimizing our novel selector loss. Meanwhile, the predictor is improved by optimizing the empirical risk, re-weighted based on the selector's output; uninformative samples identified by the selector will be down-weighed. Experiments on both synthetic and real-world datasets demonstrate the merit of our method compared to existing baselines.

## 2 RELATED WORK

**Learning with untrusted data** aims to recover the ground truth model from a partially corrupted dataset. Different noise models for untrusted data have been studied, including random label noise (Bylander, 1994; Natarajan et al., 2013; Han et al., 2018; Yu et al., 2019; Zheng et al., 2020; Zhang et al., 2020), Massart Noise (Massart & Nédélec, 2006; Awasthi et al., 2015; Hanneke, 2009; Hanneke & Yang, 2015; Yan & Zhang, 2017; Diakonikolas et al., 2019; 2020; Cheng et al., 2020; Xia et al., 2020; Zhang & Li, 2021) and adversarial noise (Kearns & Li, 1993; Kearns et al., 1994; Kalai et al., 2008; Klivans et al., 2009; Awasthi et al., 2017). Our noise model is similar to General Massart Noise (Massart & Nédélec, 2006; Hanneke, 2009; Diakonikolas et al., 2019), where the label noise is *data dependent* and label can be generated via a purely random coin flipping. The major distinct formulation in our noisy generative model is the existence of some uninformative data with high noise in label compared to informative data with low noise in label. We characterize such uninformative/informative data structure via non-vanishing label noise ratio gap. While there exists long history of literature studying learning classifiers with label noise in the training stage (Thulasidasan et al., 2019; Cheng et al., 2020; Xia et al., 2020), we are the first work to investigate learning a model for inference stage under label noise setting. We study the case where label noise is an integral part of the generative process and thus will appear during inference stage as well, where it must be detected and discarded once more. We view this as a realistic setup in industries like Finance and Healthcare.

**Selective learning** is an active research area (Chow, 1957; 1970; El-Yaniv et al., 2010; Kalai et al., 2012; Nan & Saligrama, 2017; Ni et al., 2019; Acar et al., 2020; Gangrade et al., 2021a). It extends the classic selective prediction problem and studies how to select a subset of data for different learning tasks, and has also been generalized to other problems, e.g., learning to defer human expert (Madras et al., 2018; Mozannar & Sontag, 2020). We can summarize existing methods into 4 categories: Monte Carlo sampling based methods (Gal & Ghahramani, 2016; Kendall & Gal, 2017; Pearce et al., 2020), margin based methods (Fumera & Roli, 2002; Bartlett & Wegkamp, 2008; Grandvalet et al., 2008; Wegkamp et al., 2011; Zhang et al., 2018), confidence based methods (Wiener & El-Yaniv,

2011; Geifman & El-Yaniv, 2017; Jiang et al., 2018) and customized selective loss (Cortes et al., 2016; Geifman & El-Yaniv, 2019; Liu et al., 2019; Gangrade et al., 2021b). Notably, several works propose customized losses, and incorporate them into neural networks. In (Geifman & El-Yaniv, 2019), the network maintains an extra output neuron to indicate rejection of datapoints. Liu et al. (2019) uses Gambler loss where a cost term is associated with each output neuron and a doubling-rate-like loss function is used to balance rejections and predictions. Thulasidasan et al. (2019) also applies an extra output neuron for identifying noise label to improve the robustness in learning. Huang et al. (2020) adopts a progressive label smoothing method which prevents DNN from overfitting and improves selective risk when applied to selective classification task. Cortes et al. (2016) perform data selection with an extra model and introduce a selective loss that helps maximize the coverage ratio, thus trading off a small fraction of data for a better precision. Sharing a similar spirit with (Kalai et al., 2012), (Gangrade et al., 2021b) applies an one-side prediction method to model high confidence region for each individual class, and maximizes coverage while maintains a low risk level.

Existing works on selective prediction are all motivated by the trade off between accuracy and coverage - i.e. one wants to make safe prediction to achieve higher precision while maintaining a reasonable recall. Our paper is the first to investigate the case where some (or even majority) of the data is uninformative, and thus must be discarded at test time. Unlike the selective prediction, there is a latent ground truth indicator function of whether a data point should be selected or not. Our method is guaranteed to identify those uninformative samples.

## 3 PROBLEM FORMULATION

In this section, we describe our model for the inherently-noisy data generation process that we aim to study.

**Definition 1** (Noisy Generative Process). *We define* Noisy Generative Process *by the following notation* $\boldsymbol{x} \sim \mathcal{D}_\alpha$ *where*

$$\mathcal{D}_\alpha \equiv \begin{cases} \boldsymbol{x} \sim \mathcal{D}_U & \text{with prob. } 1 - \alpha \quad \textbf{\textit{(Uninformative)}} \\ \boldsymbol{x} \sim \mathcal{D}_I & \text{with prob. } \alpha \quad\quad \textbf{\textit{(Informative)}}. \end{cases} \tag{1}$$

*Let the ground truth labeling function* $f^* : \mathcal{X} \to \{+1, -1\}$ *be in hypothesis class* $\mathcal{F}$. *Let* $\Omega_\mathcal{D} \subseteq \mathbb{R}^d$ *be the support of* $\mathcal{D}_\alpha$. *Suppose* $\{\Omega_U, \Omega_I\}$ *is a partition of* $\Omega_\mathcal{D}$. *Let* $\lambda(\boldsymbol{x}) \in (\lambda, \frac{1}{2}]$ *with* $\lambda > 0$, *the latent informative/uninformative status* $z \in \{+1, -1\}$ *has posterior distribution:*

$$\mathbb{P}[z = 1 | \boldsymbol{x}] \equiv \begin{cases} \frac{1}{2} - \lambda(\boldsymbol{x}), & \text{if } \boldsymbol{x} \in \Omega_U \\ \frac{1}{2} + \lambda(\boldsymbol{x}), & \text{if } \boldsymbol{x} \in \Omega_I. \end{cases} \tag{2}$$

*The observed data* $(\boldsymbol{x}, y)$ *is generated according to:*

$$\begin{aligned} &\boldsymbol{x} \sim \mathcal{D}_\alpha; \\ &z \sim \mathbb{P}[z | \boldsymbol{x}]; \\ &y \equiv \begin{cases} Bernoulli(0.5), & \text{if } z = -1 \\ f^*(\boldsymbol{x}), & \text{if } z = 1. \end{cases} \end{aligned} \tag{3}$$

Since $\lambda(\boldsymbol{x}) > 0$, $\boldsymbol{x}$ from $\Omega_U$ has a lower chance to be observed with true label compared to $\Omega_I$, thus can be viewed as uninformative data, in a relative sense. On the contrary, $x$ from $\Omega_I$ can be viewed as informative data. Our Noisy Generative Process follows standard data dependent label noise, e.g., Massart Noise (Massart & Nédélec, 2006) and Benign Label Noise (Hanneke, 2009; Hanneke & Yang, 2015; Diakonikolas et al., 2019). Indeed, one can always choose $\lambda(\boldsymbol{x}) \in [0, \frac{1}{2}]$ and $\alpha \in [0, 1]$ to replicate General Massart noise. Compared to classical label noise models, the assumption $\lambda(\boldsymbol{x}) > \lambda$ introduces a label noise ratio gap, which distinguishes the informative and uninformative data. In Equation 3, the $Bernoulli(0.5)$ label noise serves as a proxy for "white noise" in label corruption. When $\lambda(\boldsymbol{x}) = \frac{1}{2}$ and $\boldsymbol{x} \in \Omega_U$, $Bernoulli(0.5)$ random label noise can be viewed as the strongest known non-adversarial label noise, of both theoretical and practical interest (Diakonikolas et al., 2019). Such $Bernoulli(0.5)$ random label noise could happen when hard-to-classify examples are shown to human annotator (Klebanov & Beigman, 2010), or when fluctuations in financial market closely resemble random walks (Tsay, 2005).

A typical setting that is studied in this work is the case that both value of $\alpha$ and $\lambda$ are non-vanishing, i.e., there are significant fraction of uninformative data (large $\alpha$) and the label noise ratio gap is distinguishable between informative and uninformative data (large $\lambda$).

Next definition describes a recoverable condition of the optimal function for the latent informative/uninformative status $z$.

**Definition 2** ($\mathcal{G}$-realizable ). *Given support $\Omega$ and $\lambda(\boldsymbol{x}) \in (\lambda, \frac{1}{2}]$, let the posterior distribution of $z$ be defined in Equation* (2). *We say $\Omega$ is $\mathcal{G}$-realizable if there exists $g^* \in \mathcal{G} : \mathcal{X} \to \{+1, -1\}$ satisfying $g^*(\boldsymbol{x}) = 2\mathbb{1}\{\mathbb{P}[z = 1|\boldsymbol{x}] > \frac{1}{2}\} - 1$.*

Ideally, one wish to select all informative data where signal dominates noise. This can be done via recovering $g^*(\cdot)$, which we view as the *ground truth selector* we wish to recover. The $\mathcal{G}$-realizable condition is analogous to the realizability condition (Massart & Nédélec, 2006; Hanneke & Yang, 2015) in classical label noise problem. The major difference and challenge in recovering $g^*(\cdot)$ compared to learning a classifier, is that there is no direct observation on the informative/non-informative status $z$. The major contribution of this work is proposing a natural selector risk which recovers $g^*(\cdot)$ without observing latent variable $z$.

Having introduced the data generation process, we now describe the learning task:

**Assumption 1.** *Data $S_n = \{\boldsymbol{x}_i, y_i\}_{i=1}^N$ is i.i.d generated according to the* Noisy Generative Process *(Definition 1), with $f^* \in \mathcal{F}$ and support $\Omega$ satisfies $\mathcal{G}$-realizable condition.*

Given the above assumption, we are interested in the following learning task:

**Problem 1** (Abstain from Uninformative Data). *Under Assumption 1 with i.i.d observations from $\mathcal{D}_\alpha$, given $\widehat{f} \in \mathcal{F}$ sufficiently close to $f^*(\boldsymbol{x})$, we aim to learn a selector $\widehat{g} \in \mathcal{G}$ that is close to $g^*(\boldsymbol{x})$.*

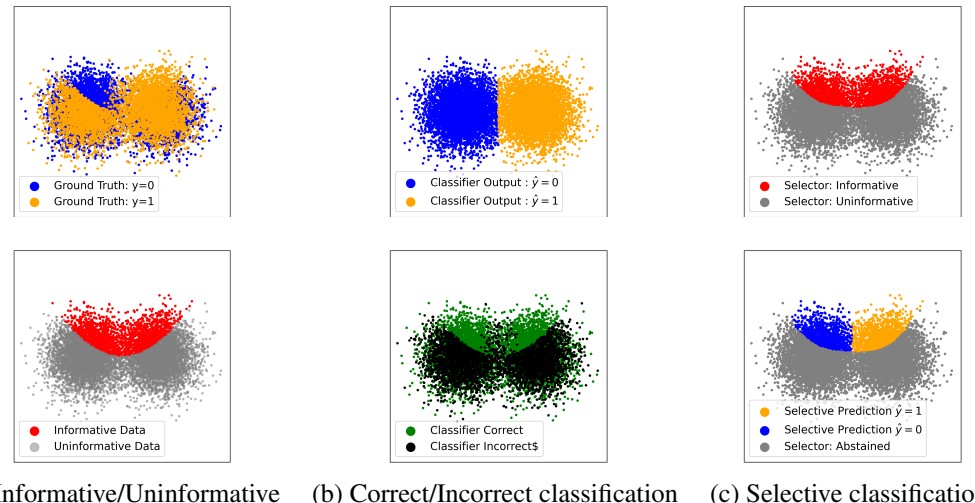

(a) Informative/Uninformative data     (b) Correct/Incorrect classification result     (c) Selective classification result

Figure 1: Illustration of the learning strategy with mixture of Gaussian data. We replace the 0-1 loss with hinge loss and train SVM models for $f$ and $g$. (a) upper panel shows the original dataset and bottom panel shows the region of informative (easy) and uninformative (hard) data. (b) shows that the classifier has high accuracy in the informative region, but low accuracy in the uninformative region. In (c), the selector trained with $\widehat{f}$ successfully recovers informative support thus resulting in low selective risk, and we abstain from making a prediction elsewhere.

## 4 OUR METHOD

In this section, we present our approach for learning and abstaining in the presence of uninformative data (Problem 1). The main challenge is that the latent informative/uninformative status of a datapoint is unknown. Our main idea is to introduce a novel yet natural *selector loss* function that trains a selector based on the performance of the best predictor (Section 4.1). In Section 4.1, we present our main theoretical result. We show that, given any reasonably good classifier, finding a selector minimizing the proposed selector loss, we can solve Problem 1, with minimax-optimal sample complexity. Inspired by the theoretical results, in Section 6, we propose a heuristic algorithm that iteratively optimizes the predictor and the selector.

### 4.1 SELECTOR LOSS

In an idealized setting, when access to latent informative/uninformative variables $\{(\boldsymbol{x}_i, z_i)\}_{i=1}^n$ is available, recovering $g^*$ shares a similar spirit with learning classfier under label noise. It suffices to minimize following classical classification risk :

$$\textit{Non-Realizable Risk}(g; S_n) \equiv \sum_{i=1}^n \mathbb{1}\{g(\boldsymbol{x}_i) \neq z_i\} \tag{4}$$

However, in practice $z$ is never revealed. To learn a selector without direct supervision, we have to leverage the performance of a predictor $f$. We propose to replace $z$ in the Equation 4 with a *pseudo-informative label* $\mathbb{1}\{f(\boldsymbol{x}) \neq y\}$, which has randomness coming from $z$ and noisy label $y$.

**Definition 3** (Selector Loss). *Given $f \in \mathcal{F}$ and its selector $g \in \mathcal{G}$, we define the following empirical version of weighted $0$-$1$ type risk w.r.t $g(\cdot)$ as selector risk:*

$$R_{S_n}(g; f, \beta) \equiv \sum_{i=1}^n \{\beta \mathbb{1}\{f(\boldsymbol{x}_i) \neq y_i\}\mathbb{1}\{g(\boldsymbol{x}_i) > 0\} + \mathbb{1}\{f(\boldsymbol{x}_i) = y_i\}\mathbb{1}\{g(\boldsymbol{x}_i) \leq 0\}\} \tag{5}$$

The selector loss is also a natural metric to evaluate the quality of the selector. This loss penalizes when (1) the predictor makes a correct prediction on a datapoint that the selector considers uninformative and abstains from, or (2) the predictor makes an incorrect prediction on a datapoint that the selector considers informative. Intuitively, the loss will drive the selector to partition the domain into informative and uninformative regions. Within the informative region, the predictor is supposed to fit the data well, and should be more accurate. Meanwhile, within the uninformative region, the label is random and the predictor is supposed to be more prone to error.

Note that there are two types of errors penalized in the selector loss: an incorrect prediction on a selected datapoint, $(f(\boldsymbol{x}) \neq y) \wedge (g(\boldsymbol{x}) > 0)$, and a correct prediction on an unselected datapoint, $(f(\boldsymbol{x}) = y) \wedge (g(\boldsymbol{x}) \leq 0)$. Since the label noise is non-adversarial, $y$ tends to have higher probability of coincidence with $f^*(\boldsymbol{x})$, introducing imbalance on the pseudo-informative label. We thus use $\beta$ to weigh these two types of errors in the loss. An analysis can be found in section A.1 on the choice of $\beta$. Our theoretical analysis suggests that for a wide range of $\beta$, the accuracy of the selector is guaranteed. Empirical study also shows stability with regard to these choices.

**Learning a selector with the novel loss.** To learn a selector, one can follow standard procedure e.g., empirical risk minimization(ERM), to estimate a predictor $\widehat{f}$ with reasonable quality. The selector can be estimated by minimizing the selector loss $\widehat{g} = \arg\min_{g \in \mathcal{G}} R_{S_n}(g, \widehat{f}, \beta)$, conditioned on the estimated predictor $\widehat{f}$.

In Figure 1, we show an example of using the ERM strategy using SVM with $0$-$1$ loss replaced by hinge loss. In this case, the losses are all convex and the empirical minimizers $\widehat{f}$ and $\widehat{g}$ can be computed exactly.

In practice, however, empirical minimization is not always possible, as optimization for complex models (e.g., DNNs) and non-convex losses remains open. We therefore propose a heuristic algorithm in the spirit of our theoretical results - it jointly learns $f$ and $g$ by minimizing the selector loss and a reweighed classification risk iteratively (see Section 6).

## 5 MINIMAX-OPITMAL RISK BOUND

In this section we present our main theoretical results. The main result can be summarized in following (informal) statement.

**Main Result (Informal)** For any reasonably good predictor $\widehat{f}$, with sufficient data, the selector $\widehat{g}$ estimated using $\widehat{g} = \arg\min_{g \in \mathcal{G}} R_{S_n}(g, \widehat{f}, \beta)$ is sufficiently close to the targets $g^*$ with high probability.

**Remark 1.** *The toolkit we use in the proof is a Bernstein type inequality for fast generalization rate under margin condition (Massart & Nédélec, 2006; Van Erven et al., 2015; Li & Liu, 2021). We also provide an information theoretic lower bound construction in section A.2 to show our risk bound is minimax-optimal. Our construction of the lower bound is motivated from (Ehrenfeucht et al., 1989;*

*Blumer et al., 1989) and Le Cam's method (Yu, 1997). Due to space constraints, detailed proofs of our theorems are provided in the Appendix. We also present our extension from finite hypothesis class to VC-class using Local Rademacher Average tools (Bartlett et al., 2005) in section A.6. In the analysis, we do not pursue risk bounds for learning $f^*$ since it has been thorough studied in existing literatures (Blum et al., 2016; Mohri et al., 2018; Bartlett et al., 2005; Massart & Nédélec, 2006). Instead, the theorem admits any classifier $\widehat{f}$ that is close to $f^*$ in a PAC fashion, providing additional flexibility in choosing classifier $\widehat{f}$.*

**Theorem 1** (Risk Bound). *Let $S_n = \{(\boldsymbol{x}_i, y_i)\}_{i=1}^n$ be i.i.d sample from Data Generative Process described in Definition 1 under Assumption 1, with $f^*(\cdot) \in \mathcal{F}$ and $g^*(\cdot) \in \mathcal{G}$, $|\mathcal{F}| < \infty$ $|\mathcal{G}| < \infty$. Given $\lambda$, let $\beta \in \left[\frac{3-2\lambda}{1+2\lambda} + \lambda, min(\frac{3+2\lambda}{1-2\lambda} - \frac{\lambda}{1-4\lambda^2}, 10)\right]$. For any $\widehat{f}(\cdot) \in \mathcal{F}$, let $\widehat{g} = \underset{g \in \mathcal{G}}{\arg\min}\, R_{S_n}(g; \widehat{f}, \beta)$. Then for any $\varepsilon > 0$, there is a $\delta > 0$ such that the following holds: For $n \geq max\{\frac{32\beta^2 \log(\frac{|\mathcal{G}|}{\delta})}{\lambda\varepsilon}, \frac{24\beta \log(\frac{|\mathcal{F}|}{\delta})}{\varepsilon}\}$, and for $\widehat{f}$ that satisfies one of the following condition:*

- *For any $\widehat{f}(\cdot) \in \mathcal{F}$ that $\mathbb{E}_{\boldsymbol{x}}[\widehat{f}(\boldsymbol{x}) \neq f^*(\boldsymbol{x})] \leq \frac{\varepsilon}{8\beta}$ with prob at least $1 - \delta$,*

- *If $\lambda = \frac{1}{2}$, for any $\widehat{f}(\cdot) \in \mathcal{F}$ that $\mathbb{E}_{\boldsymbol{x}}[\widehat{f}(\boldsymbol{x}) \neq f^*(\boldsymbol{x})|\boldsymbol{x} \in \Omega_I] \leq \frac{\varepsilon}{8\beta\alpha}$ with prob at least $1 - \delta$,*

*The following holds with probability at least $1 - 2\delta$:*

$$R(\widehat{g}; f^*, \beta) - R(g^*; f^*, \beta) \leq \varepsilon$$

**Remark 2.** *The assumption that $\mathbb{E}_{\boldsymbol{x}}[\widehat{f}(\boldsymbol{x}) \neq f^*(\boldsymbol{x})] \leq \varepsilon$ could be achieved via an ERM on classification loss $\sum_{i=1}^n \mathbb{1}\{f(\boldsymbol{x}_i) \neq y_i\}$ under some margin condtions (Massart & Nédélec, 2006; Bartlett et al., 2005). In practice, one can also apply some methods beyond ERM to obtain $\widehat{f}$ (Namkoong & Duchi, 2017; Zhang et al., 2017b; Huang et al., 2020). In particular, in case $\lambda = \frac{1}{2}$, the data in support $\Omega_U$ is un-learnable as $y$ are purely random. While approximating $f^*$ on the full support is not possible in general, one can control the conditional risk $\mathbb{E}_{\boldsymbol{x}}[\widehat{f}(\boldsymbol{x}) \neq f^*(\boldsymbol{x})|\boldsymbol{x} \in \Omega_I]$ via a standard ERM schema (see proof in appendix Section A.5). We stress that Theorem 1 holds for any classifier that is close to $f^*$, even the case where $\widehat{f}$ and $\widehat{g}$ are trained on the same dataset.*

**Corollary 1** (Recovering $g^*$). *Given conditions in Theorem 1, if we choose $\beta = 3$, we have:*

$$\mathbb{E}_{\boldsymbol{x}}[\mathbb{1}\{\widehat{g}(\boldsymbol{x}) \neq g^*(\boldsymbol{x})\}] \leq \frac{4\varepsilon(1 + 2\lambda)}{\lambda} \tag{6}$$

Corollary 1 suggests that by minimizing the empirical version of the loss from Definitions 3, one can recover $g^*$ in a PAC fashion. The theoretical guarnatee holds even under a very challenging case were $\alpha > 0.5$ and $\lambda = \frac{1}{2}$, .e.g, majority of the data have purely random labels.

The analysis of the selector loss (Theorem 1) relies on the quality of the classifier $\widehat{f}$. But since we know that $\widehat{g}$ is able to abstain from uninformative data, we can retrain $\widehat{f}$ beyond standard ERM, with up weighted informative data, therefore improving the accuracy of $\widehat{f}$. Such circular logic naturally leads to a practical iterative algorithm that we present in the next section.

---

**Algorithm 1** `Iterative Soft Abstain`

---

1: **Input:** Data set $S_n = \{(\boldsymbol{x}_1, y_1), ..., (\boldsymbol{x}_n, y_n))\}$, weight parameter:$\beta$, random initial $f^0$ and $g^0$, initial sample weights $\gamma_i^0 = \frac{1}{n}, \forall i \in [n]$, meta learning rate $\eta$, number of iterations $T$

2:

3: **for** $t \leftarrow 1, \cdots, T$ **do**

4:      Optimize loss to update predictor $f^t : \sum_{i=1}^n \gamma_i^t \{y_i \log(f(\boldsymbol{x}_i)) + (1 - y_i)\log(1 - f(\boldsymbol{x}_i))\}$

5:      Approximate the 'pseudo-informative label' : $z_i^t = \mathbb{1}\{\mathbb{1}\{f^t(\boldsymbol{x}_i) > 0.5\} = y_i\}$

6:      Optimize loss to update selector $g^t : \sum_{i=1}^n \{z_i^t \log(g(\boldsymbol{x}_i)) + \beta(1 - z_i^t)\log(1 - g(\boldsymbol{x}_i))\}$

7:      Update sample weights using $g^t : \gamma_i^{t+1} = \frac{\gamma_i^t(1 + \eta\mathbb{1}\{g^t(\boldsymbol{x}_i) > 0.5\})}{\sum_{j=1}^n \gamma_j^t(1 + \eta\mathbb{1}\{g^t(\boldsymbol{x}_j) > 0.5\})}$.

8: **end for**

9: **Output:** $f^T, g^T$

---

## 6 A PRACTICAL HEURISTIC ALGORITHM

Motivated by our theoretical analysis, we propose a practical algorithm sharing a similar spirit with the selector loss. From a computational standpoint, we replace the binary loss by cross-entropy loss instead and require that both $f$ and $g$ have continuous-valued output, ranging between 0 and 1 instead of binary output $\{+1, -1\}$. The labels $y$ also needs to be processed so that the values are in the $\{0, 1\}$ range. We also relax the requirement for minimization oracles, allowing the practical algorithm to jointly optimize the predictor and the selector in an iterative manner. At each iteration, we update the predictor using the informative data selected by the selector, and then update the selector based on the predictor's output. See Algorithm 1 for the pseudo-code. A pictorial example of Algorithm 1's performance can be found in Figure 3 of the Appendix.

During the joint optimization process, the predictor is counting on the selector to upweigh informative data. By putting more effort on the informative data, we wish to improve the performance of predictor via learning beyond ERM (Zhang et al., 2017b; Ren et al., 2018; Huang et al., 2020). However, the initial selector is not trustworthy. To update the predictor $f$, we turn to a so-called soft abstention scheme: use a weight vector $\gamma$ that progressively down-weighs samples abstained by $g$, in the spirit of multiplicative weights update (MWU) algorithms (Cesa-Bianchi & Lugosi, 2006; Arora et al., 2012). Specifically, we increase the weight of $i$-th sample $\gamma_i$ if the selector accepts $\boldsymbol{x}_i$: $\gamma_i = \gamma_i(1 + \eta \cdot \mathbb{1}\{g(\boldsymbol{x}_i) > 0.5\})$ and then normalize so that $\sum_{i=1}^{n} \gamma_i = 1$. We call this a soft abstention approach because the algorithm decreases the weight of uninformative data gradually. We count on MWU mechanism to serve as a soft version of the selector, allowing the classifier to put less effort on learning uninformative data.

## 7 EXPERIMENTS

In this section, we test the efficacy of our heuristic algorithm (Algorithm 1) on publicly-available datasets. The empirical study aims to answer following questions.

$Q_1$ : *Is Algorithm 1 able to approximate ground truth selector $g^*$ ?*
The Answer is yes. The empirical result on Semi-synthetic dataset (Figure 2 in Section 7.1) suggests that Algorithm 1 recovers ground truth selector $g^*$ within reasonable error range.

$Q_2$ : *How does Algorithm 1 compare to baselines on semi-synthetic dataset in recovering ground truth selector $g^*$?*
The results are presented in table 1 in Section 7.2. All baselines are simply not equipped with the functionality to distinguish informative/uninformative data automatically. They all suffers from poor estimation of $\alpha$, the proportion of informative data in the dataset, which must be given as a hyper-parameter. However, our method does not require such prior information and provably recovers $\alpha$ ( which is implied by recovery of $g^*$), thus consistently behaves well.

$Q_3$ : *How does Algorithm 1 work on real world datasets compared to selective learning baselines?*
On real world datasets, Algorithm 1 consistently gains competitive performance against other baselines in low coverage regime, e.g., the proportion of data chosen by selector $\leq 20\%$. These empirical results suggest that that our method is good at picking out strongly informative data.

**Baselines.** We compare our method to two of the recently proposed selective learning algorithms. (1) **SelectiveNet** (Geifman & El-Yaniv, 2019), which integrates an extra neuron as a data selector in the output layer and also introduces a loss term to control the coverage ratio; (2) **DeepGambler** (Liu et al., 2019), which also maintains an extra neuron for abstention and uses a doubling-rate-like loss term (i.e., gambler loss) to train the model. (3) We also create a third baseline that selects data using model prediction confidence, which we refer to as **Confidence**. The intuition behind this heuristic baseline is that informative data should have higher confidence compared to uninformative data.

**Experiment Details and Ablation Study:** Due to limited space, all experiment setting details are given in Appendix Section B.1. Detailed ablation studies are provided in Appendix Section C.

### 7.1 EXPERIMENTS USING SEMI-SYNTHETIC DATA FOR $Q_1$

Table 1: Synthetic Data Experiment on MNIST+Fashion and SVHN

| Dataset | Uninformative Num. | Informative Num. | Criterion | Confidence | SLNet | DeepGambler | Ours |
|---|---|---|---|---|---|---|---|
| MNIST + Fashion | 60000 | 15000 | SR(%) | 59.32±0.83 | 45.44±0.68 | 45.37±1.29 | **10.35±0.31** |
| | | | Precision | 0.46±0.79 | 0.61±0.01 | 0.61±0.01 | **1.00±0.00** |
| | | | Recall | 0.94±0.07 | **1.00±0.00** | **1.00±0.00** | 0.85±0.01 |
| | 60000 | 30000 | SR(%) | 39.23±0.47 | 28.15±0.63 | 27.78±0.67 | **12.03±1.06** |
| | | | Precision | 0.68±0.56 | 0.80±0.01 | 0.80±0.00 | **1.00±0.00** |
| | | | Recall | 0.98±0.00 | **1.00±0.00** | **1.00±0.00** | 0.92±0.04 |
| | 60000 | 45000 | SR(%) | 28.92±0.49 | 20.22±0.58 | 19.09±0.41 | **12.58±2.00** |
| | | | Precision | 0.79±0.29 | 0.88±0.86 | 0.89±0.00 | **1.00±0.00** |
| | | | Recall | 0.99±0.00 | **1.00±0.00** | **1.00±0.00** | 0.97±0.03 |
| | 60000 | 60000 | SR(%) | 21.88±0.39 | 15.38±0.32 | 14.20±0.62 | **11.61±0.79** |
| | | | Precision | 0.86±0.46 | 0.92±0.22 | 0.94±0.00 | **1.00±0.00** |
| | | | Recall | **1.00±0.00** | **1.00±0.00** | **1.00±0.00** | 0.97±0.01 |
| SVHN | 33800 | 9200 | SR(%) | 64.91±0.89 | 48.61±23.37 | 18.24±2.00 | **7.03±0.01** |
| | | | Precision | 0.53±0.01 | 0.52±0.22 | 0.80±0.02 | **0.93±0.01** |
| | | | Recall | **0.98±0.00** | 0.89±0.10 | 0.96±0.01 | 0.85±0.01 |
| | 33800 | 18300 | SR(%) | 47.64±0.89 | 28.15±22.74 | 12.03±0.99 | **5.41±0.97** |
| | | | Precision | 0.70±0.01 | 0.75±0.18 | 0.88±0.02 | **0.95±0.01** |
| | | | Recall | **0.99±0.00** | 0.96±0.10 | 0.98±0.01 | 0.88±0.01 |
| | 33800 | 26200 | SR(%) | 36.53±1.57 | 12.96±1.70 | 8.65±0.44 | **4.05±0.67** |
| | | | Precision | 0.79±0.01 | 0.88±0.01 | 0.91±0.01 | **0.97±0.01** |
| | | | Recall | **0.99±0.00** | 0.98±0.00 | 0.98±0.00 | 0.89±0.01 |
| | 33800 | 28400 | SR(%) | 34.48±1.15 | 12.10±1.21 | 8.31±0.64 | **4.43±0.86** |
| | | | Precision | 0.80±0.01 | 0.89±0.01 | 0.92±0.01 | **0.96±0.01** |
| | | | Recall | **0.99±0.00** | 0.98±0.00 | 0.98±0.00 | 0.89±0.01 |

**Dataset Construction**: We explicitly control the support of informative/uninformative data. For MNIST+FashionMNIST dataset, images from MNIST are set to be uninformative and images from Fashion-MNIST are set to be informative. For SVHN(Netzer et al., 2011) dataset, class 5-9 are set to be uninformative and class 0-4 are set to be informative. Datasets are constructed with different values of informative data fraction $\alpha$ and label noise ratio gap $\lambda$ according to the noisy generative process. We inject label noise accordingly to Definition 1 by setting $\lambda(\boldsymbol{x}) = \lambda$. We shuffle the labels of informative/uninformative data according to different values of $\lambda$ and mix informative/uninformative data according to $\alpha$. In particular, we choose $\alpha \in \{25\%, 50\%\}$ and $\lambda \in \{0.3, 0.35, 0.4\}$.

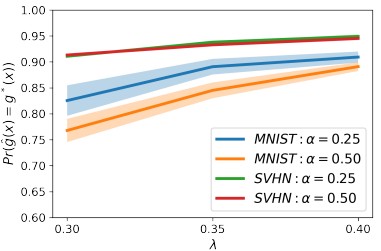

Figure 2: Recover $g^*$ using Algorithm 1 under different label noise ratio gap $\lambda$.

**Results and Discussion.** The average accuracy of the selector given by Algorithm 1 compared to the ground truth selector is presented in Figure 2. As one can observe, Algorithm 1 recovers $g^*$ within reasonable error range. In addition, the accuracy improves as $\lambda$ increases, which supports our bound in Equation 1. MNIST+FashionMNIST data turns out to be more challenging than SVHN in recovering $g^*$. We believe this is because informative data in MNIST+FashionMNIST has 10-classes, which is more challenging for learning a predictor $\widehat{f}$ compared to SVHN with 5 classes. The quality of selector $\widehat{g}$ suffers from imprecise $\widehat{f}$.

## 7.2 EXPERIMENTS USING SEMI-SYNTHETIC DATA FOR $Q_2$

**Dataset Construction**: The construction of informative/uninformative data follows Section 7.1. We uniformly shuffle the labels for uninformative data and keep original labels of informative data. Informative and uninformative datasets are mixed in different proportions as proxy for noisy generative process with different $\alpha$. The construction of dataset mimics the noisy generative process with $\lambda = \frac{1}{2}$. The choice of $\lambda$ ensures that the informative data has no label noise. Such noiseless setting allows three baselines to estimate/set $\alpha$ to the best of their ability, according to accuracy of estimated predictor $\widehat{f}$.

**Evaluation Metric.** We use three criteria to jointly evaluate a selective learning outcome. (1) Selective risk (SR). Selective risk is the empirical risk measured over data points selected by the algorithm. This is a metric that is also adopted in (Geifman & El-Yaniv, 2019; Liu et al., 2019). (2) Precision. Precision is the proportion of true informative data point among all the data picked out by the selector. (3) Recall. Recall is the proportion of true informative samples picked out by the

selector out of all the informative samples in the dataset. SR evaluates the quality of the classifier, Precision and Recall are the standard ML metrics for the selector. An ideal algorithm should have low SR, high precision and high recall.

**Results and discussion.** Table 1 presents the results of MNIST+FashionMNIST and SVHN dataset with different fractions ($\alpha$) of informative data. The selective learning methods, SelectiveNet and DeepGambler, perform poorly because they require prior information $\alpha$ to be given as input. The estimation of $\alpha$ turns out to be very challenging in practice due to the presence of noisy uninformative data. In contrast, our method provably recovers $\alpha$ automatically and is robust against the choice of hyper-parameter ($\beta$). The ablation study in Appendix Fig 7 exhibits this stability: our method consistently behaves well given different values of $\beta$. A thorough exploration of the estimation of $\alpha$ and corresponding performance of baselines is provided in Appendix Tables 10,11 and 12. We also provide ablation study on the MWU mechanism in Figure 4 and Table 13, showing its ability to weight up informative data and improve algorithm's performance.

## 7.3 EXPERIMENTS USING REAL-WORLD DATA FOR $Q_3$

In this section, we report our empirical study on 3 publicly-available datasets: (1) breast ultrasound images (BUS) (Al-Dhabyani et al., 2020), (2) lending club dataset (LC), and (3) Oxford realized volatility (Volatility) data set (Heber et al., 2009). We aim to demonstrate the potential application of the proposed algorithm in real-world application and its advantages in selecting useful information out of noisy dataset.

**Evaluation Metric.** Unlike synthetic experiments, in real-world data set, the ground-truth labels showing which data is informative and which is uninformative is not available. Metrics like precision/recall are not applicable. Instead, we report the selective risk of each algorithm given different coverage level. Specifically, we pick testing data points that have top coverage% selective confidence given by each selector, and calculate the testing selective risk at different coverage level accordingly.

**Result and Discussion.** From Table 2, we can see our method gains competitive performance against other baselines at low coverage level. This suggests that our method is especially good at picking out strongly informative data. Strongly informative data are easier to learn thus the classifier is more consistent with the ground-truth model. Such low risk regime can be captured by our selector loss, leading to low selective risk.

Table 2: Real-World Experiments: Selective Risk v.s Coverage

| Dataset | Coverage | Confidence | SLNet | DeepGambler | Ours |
|---------|----------|------------|-------|-------------|------|
| Volatility | 0.02 | 0.072±0.002 | 0.074±0.000 | 0.267±0.033 | **0.046±0.002** |
| | 0.05 | 0.088±0.002 | 0.091±0.000 | 0.281±0.013 | **0.073±0.003** |
| | 0.10 | **0.118±0.005** | 0.127±0.003 | 0.327±0.012 | **0.116±0.004** |
| | 0.20 | **0.160±0.003** | 0.200±0.004 | 0.374±0.008 | 0.192±0.005 |
| BUS | 0.02 | 0.040±0.007 | 0.014±0.020 | **0.000±0.000** | **0.000±0.000** |
| | 0.05 | 0.040±0.007 | 0.014±0.020 | **0.000±0.000** | **0.000±0.000** |
| | 0.10 | 0.040±0.007 | **0.014±0.020** | 0.000±0.000 | 0.042±0.059 |
| | 0.20 | **0.040±0.007** | **0.063±0.026** | 0.073±0.015 | 0.083±0.029 |
| LC | 0.02 | 0.469±0.014 | 0.212±0.022 | **0.153±0.024** | **0.136±0.013** |
| | 0.05 | 0.286±0.008 | 0.207±0.015 | **0.170±0.026** | 0.177±0.013 |
| | 0.10 | 0.239±0.008 | **0.218±0.009** | 0.190±0.024 | 0.221±0.010 |
| | 0.20 | 0.248±0.005 | 0.251±0.004 | **0.218±0.021** | 0.271±0.007 |

## 8 CONCLUSION AND FUTURE WORK

In this work, we take the first step towards principled learning in domains where a lot of data is naturally uninformative/highly noisy and should be discarded in learning and inference stage. We propose a general noisy generative process that formally describes such setting. A novel loss is designed for the training of the selector model with theoretical guarantees. Based on this loss, we design a heuristic algorithm that jointly learns the predictor and selector. Our empirical study support merit of our methods. We believe the Noisy Generative Process can be generalize to solve different problem, such as active learning (Cohn et al., 1994) and out of distribution generalization (Arjovsky et al., 2019). We look forward to these extensions in future work.

## 9 REPRODUCIBILITY STATEMENT

We describe a synthetic data generation procedure, evaluation metrics in Appendix section 7. For the convenience of the reader to reproduce the experiment, we also summarize the experiment setting and give implementation details in section B.1. The source code as well as parameters to reproduce the experimental results will be made available together with the publication of the paper.

## 10 ETHICS STATEMENT

This paper focuses on a theoretical discussion about learning from data that contains different portion of non-informative samples. Our experiments only use publicly available datasets. Our discussion, analysis, or data shouldn't raise any ethics-related issues. The learning method proposed in this paper, however, can be potentially used in applications with fairness and privacy concerns. It our future efforts in this area, we aim to address and resolve possible negative impact.

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

## A  THEORETIC RESULTS DETAILS

In this appendix section, we present the missing proofs as well as additional empirical results.

### A.1  PRELIMINARY

We describe the risk for selector loss on $(\boldsymbol{x}, y) \sim \mathcal{X} \times \mathcal{Y} \subset \mathbb{R}^d \times \{+1, -1\}$.

$$R(g; f, \beta) := \mathbb{E}_{\boldsymbol{x},y}\big[\beta \mathbb{1}\{g(\boldsymbol{x}) = 1\}\mathbb{1}\{f(\boldsymbol{x}) \neq y\} + \mathbb{1}\{g(\boldsymbol{x}) \neq 1\}\mathbb{1}\{f(\boldsymbol{x}) = y\}\big] \tag{7}$$

The choice of $\beta$ should ensure that given Bayes optimal classifier $f^*(\cdot)$, $g^*(\cdot)$ is the minimizer for the selector risk $R(g; f^*, \beta)$. We have that given $f^*$, the risk gap between any selector $g$ and $g^*$ is $R(g; f^*, \beta) - R(g^*; f^*, \beta)$ could be written as:

$$
\begin{aligned}
&R(g; f^*, \beta) - R(g^*; f^*, \beta) \\
&= \mathbb{E}_{\boldsymbol{x},y}\big[\beta \mathbb{1}\{g(\boldsymbol{x}) = 1\}\mathbb{1}\{g^*(\boldsymbol{x}) = 1\}\mathbb{1}\{f^*(\boldsymbol{x}) \neq y\} \\
&\qquad + \beta \mathbb{1}\{g(\boldsymbol{x}) = 1\}\mathbb{1}\{g^*(\boldsymbol{x}) \neq 1\}\mathbb{1}\{f^*(\boldsymbol{x}) \neq y\} \\
&\qquad + \mathbb{1}\{g(\boldsymbol{x}) \neq 1\}\mathbb{1}\{g^*(\boldsymbol{x}) = 1\}\mathbb{1}\{f^*(\boldsymbol{x}) = y\}\big] \\
&\qquad + \mathbb{1}\{g(\boldsymbol{x}) \neq 1\}\mathbb{1}\{g^*(\boldsymbol{x}) \neq 1\}\mathbb{1}\{f^*(\boldsymbol{x}) = y\}\big] \\
&\quad - \mathbb{E}_{\boldsymbol{x},y}\big[\beta \mathbb{1}\{g(\boldsymbol{x}) = 1\}\mathbb{1}\{g^*(\boldsymbol{x}) = 1\}\mathbb{1}\{f^*(\boldsymbol{x}) \neq y\} \\
&\qquad + \beta \mathbb{1}\{g(\boldsymbol{x}) \neq 1\}\mathbb{1}\{g^*(\boldsymbol{x}) = 1\}\mathbb{1}\{f^*(\boldsymbol{x}) \neq y\} \\
&\qquad + \mathbb{1}\{g(\boldsymbol{x}) \neq 1\}\mathbb{1}\{g^*(\boldsymbol{x}) \neq 1\}\mathbb{1}\{f^*(\boldsymbol{x}) = y\} \\
&\qquad + \mathbb{1}\{g(\boldsymbol{x}) = 1\}\mathbb{1}\{g^*(\boldsymbol{x}) \neq 1\}\mathbb{1}\{f^*(\boldsymbol{x}) = y\}\big] \\
&= \mathbb{E}_{\boldsymbol{x}}\Big[\Big\{\beta\Big(\frac{1}{4} + \frac{\lambda(\boldsymbol{x})}{2}\Big) - \frac{3}{4} + \frac{\lambda(\boldsymbol{x})}{2}\Big\}\mathbb{1}\{g(\boldsymbol{x}) = 1\}\mathbb{1}\{g^*(\boldsymbol{x}) \neq 1\}\Big] \\
&\quad + \mathbb{E}_{\boldsymbol{x}}\Big[\Big\{\frac{3}{4} + \frac{\lambda(\boldsymbol{x})}{2} - \beta\Big(\frac{1}{4} - \frac{\lambda(\boldsymbol{x})}{2}\Big)\Big\}\mathbb{1}\{g(\boldsymbol{x}) \neq 1\}\mathbb{1}\{g^*(\boldsymbol{x}) = 1\}\Big]
\end{aligned}
\tag{8}
$$

Since $\lambda(\boldsymbol{x})$ is data dependent, to ensure that $R(g, f^*, \beta) \geq R(g^*; f^*, \beta)$ for all $g \in \mathcal{G}$, it suffices to pick $\beta\big(\frac{1}{4} + \frac{\lambda(\boldsymbol{x})}{2}\big) - \frac{3}{4} + \frac{\lambda(\boldsymbol{x})}{2} > 0$ and $\frac{3}{4} + \frac{\lambda(\boldsymbol{x})}{2} - \beta\big(\frac{1}{4} - \frac{\lambda(\boldsymbol{x})}{2}\big) > 0$, we need $\beta \geq \sup_{\boldsymbol{x}} \frac{3 - 2\lambda(\boldsymbol{x})}{1 + 2\lambda(\boldsymbol{x})}$ and $\beta \leq \inf_{\boldsymbol{x}} \frac{3 + 2\lambda(\boldsymbol{x})}{1 - 2\lambda(\boldsymbol{x})}$ which implies that it suffices to pick $\beta \in \big[\frac{3 - 2\lambda}{1 + 2\lambda}, \frac{3 + 2\lambda}{1 - 2\lambda}\big]$.

Assuming $\lambda(\boldsymbol{x}) \geq \lambda$, we pick $\beta$ within certain margin of the above interval: by picking $\left[\frac{3-2\lambda}{1+2\lambda} + \lambda, \frac{3+2\lambda}{1-2\lambda} - \frac{\lambda}{1-4\lambda^2}\right]$ we have

$$R(g; f^*, \beta) - R(g^*; f^*, \beta) \geq \frac{\lambda}{4(1+2\lambda)} \mathbb{E}_{\boldsymbol{x}}[\mathbb{1}\{g(\boldsymbol{x}) \neq g^*(\boldsymbol{x})\}] \tag{9}$$

### A.2 PROOF OF INFORMATION THEORETIC LOWER BOUND

In this section we quantify the hardness of recovering $g^*$, from an information theoretic perspective. Let $(\boldsymbol{x}, y) \sim \mathcal{X} \times \mathcal{Y} \subset \mathbb{R}^d \times \{+1, -1\}$, we set $\mathcal{X} : \{\tau \cdot \boldsymbol{e} | \boldsymbol{e} \in \{\boldsymbol{e}^1, ..., \boldsymbol{e}^d\}, |\tau| \leq 1\}$ where $e^j$ represents the $j$-th cannonical basis. Let $\boldsymbol{w}$ be vector of ones, $\boldsymbol{w} = \mathbb{1}$, we set $f^*(\boldsymbol{x}) = 2\mathbb{1}\{\boldsymbol{w}^\top \boldsymbol{x} > 0\} - 1$. Let $\mathcal{G}$ be the hypothesis class that contains $g^*(\boldsymbol{x})$. In our lower bound construction $|\mathcal{G}| = 2^d$ and $g(\boldsymbol{x}) = \sum_{j=1}^d \mathbb{1}\{\boldsymbol{x}^\top \boldsymbol{e}^j \neq 0\}\{-\mathbb{1}\{\|\boldsymbol{x}\| \geq 1-\alpha\} \cdot \zeta^j + \mathbb{1}\{\|\boldsymbol{x}\| \leq \alpha\} \cdot \zeta^j - \mathbb{1}\{\alpha \leq \|\boldsymbol{x}\| \leq 1-\alpha\}\}$ where $\zeta \in \{+1, -1\}^d$. For example, suppose for each $j = 1, \ldots, d$, $\boldsymbol{e}^j$ is the vector with the $j$th entry one and the other entries zero. Then for $\boldsymbol{x} = \tau \cdot \boldsymbol{e}^j$ for some $j \in \{1, \ldots, d\}$, $f^*(\boldsymbol{x})$ is 1 if $\tau$ is positive and it is $-1$ if $\tau$ is negative. Moreover, $g(x)$ is $-\zeta^j$ if $\|\tau\| \geq 1 - \alpha$, it is $\zeta^j$ if $\|\tau\| \leq \alpha$, and it is $-1$ otherwise.

Let $\boldsymbol{\sigma} \in \{+1, -1\}^d$ be a $d$-dimensional Rademacher vector and we set $g^*(\boldsymbol{x}) = \sum_{j=1}^d \mathbb{1}\{\boldsymbol{x}^\top \boldsymbol{e}^j \neq 0\}\{-\mathbb{1}\{\|\boldsymbol{x}\| \geq 1-\alpha\} \cdot \sigma^j + \mathbb{1}\{\|\boldsymbol{x}\| \leq \alpha\} \cdot \sigma^j - \mathbb{1}\{\alpha \leq \|\boldsymbol{x}\| \leq 1-\alpha\}\}$ where $\boldsymbol{\sigma} \in \{+1, -1\}^d$. In another word, the support of $\mathcal{D}_\alpha = \cup_{j=1}^d \Omega^j$ where $\Omega^j : \{\boldsymbol{x} | \boldsymbol{x} = \tau \cdot \boldsymbol{e}^j\}$. If $\sigma^j = -1$, the informative part of $\Omega^j$ is $\{\boldsymbol{x} | \|x\| \geq 1 - \alpha\}$ otherwise the informative part of $\Omega^j$ becomes $\{\boldsymbol{x} | \|x\| \leq \alpha\}$. Assuming $f^*(\boldsymbol{x})$ is deterministic and let $S = \{(\boldsymbol{x}_i, y_i)\}_{i=1}^n$ be generated from following process which we denote as $\mathcal{Q}$:

$$\boldsymbol{\sigma} \sim Unif\{+1, -1\}^d$$

$$g^*(\boldsymbol{x}) = \sum_{j=1}^d \left\{ \mathbb{1}\{\boldsymbol{x}^\top \boldsymbol{e}^j \neq 0\}\{-\mathbb{1}\{\|\boldsymbol{x}\| \geq 1-\alpha\} \cdot \sigma^j \right.$$

$$\left. + \mathbb{1}\{\|\boldsymbol{x}\| \leq \alpha\} \cdot \sigma^j - \mathbb{1}\{\alpha \leq \|\boldsymbol{x}\| \leq 1-\alpha\}\} \right\}$$

Generate $S = \{(\boldsymbol{x}_i, y_i)\}_{i=1}^n$ according to:

$$j \sim \begin{cases} j = 1, & \text{with prob } 1 - \frac{\varepsilon}{\lambda} \\ j \sim Unif\{2, ..., d\} & \text{with prob } \frac{\varepsilon}{\lambda}. \end{cases} \tag{10}$$

$$\tau \sim Unif[-1, 1]$$

$$\boldsymbol{x} = \tau \cdot e^j$$

$$y = \begin{cases} f^*(\boldsymbol{x}), & \text{with prob } \frac{3}{4} + \frac{g^*(\boldsymbol{x})\lambda}{2} \\ -f^*(\boldsymbol{x}) & \text{with prob } \frac{1}{4} - \frac{g^*(\boldsymbol{x})\lambda}{2} \end{cases}$$

Let $\mathcal{A}$ be *any* (potentially randomized) algorithm that takes dataset $S_{\boldsymbol{\sigma}}$ as input where $S_{\boldsymbol{\sigma}}$ is generated from the process described in Equation 10. Let $\widehat{g}$ be the hypothesis ouput of algorithm $\mathcal{A}$. For a parameter $\beta$ we define

$$R(\mathcal{A}(S_{\boldsymbol{\sigma}}), \beta) = R(\widehat{g}(\boldsymbol{x}), f^*, \beta) = \mathbb{E}_{\boldsymbol{x}, y}\left[\beta \mathbb{1}\{\widehat{g}(\boldsymbol{x}) = 1\}\mathbb{1}\{f^*(\boldsymbol{x}) \neq y\} + \mathbb{1}\{\widehat{g}(\boldsymbol{x}) \neq 1\}\mathbb{1}\{f^*(\boldsymbol{x}) = y\}\right]. \tag{11}$$

**Theorem 2.** *Consider the noisy generative process defined in Definition 1 with $\Omega$ being $\mathcal{G}$-realizable. For any $\varepsilon \leq \lambda$, to achieve*

$$\mathbb{E}_{S_n}[R(\mathcal{A}(S_n), f^*, \beta) - R(g^*, f^*, \beta)] \leq \frac{\varepsilon}{8(1+2\lambda)}$$

*with $\beta \in \left[\frac{3-2\lambda}{1+2\lambda} + \lambda, \frac{3+2\lambda}{1-2\lambda} - \frac{\lambda}{1-4\lambda^2}\right]$ ,any algorithm $\mathcal{A}$ will take at least $\frac{\log(|\mathcal{G}|)}{\lambda\varepsilon}$ many samples.*

*Proof.* The lower bound construction is presented in Equation 10. Note when $\lambda = \frac{1}{2}$, $y$ becomes purely random. Our lower bound construction in case $\lambda = \frac{1}{2}$ works for any $f$ that is consistent with

$f^*$ on $\Omega_I$. From equation (9) the risk gap $R(\widehat{g}, f^*, \beta) - R(g^*, f^*, \beta)$ averaged over $\boldsymbol{\sigma}$ and $S_{\boldsymbol{\sigma}}$ can be written as

$$
\begin{aligned}
&\mathbb{E}_{\boldsymbol{\sigma}} \mathbb{E}_{S_{\boldsymbol{\sigma}}} [R(\widehat{g}, f^*, \beta) - R(g^*, f^*, \beta)] \\
&\geq \frac{\lambda}{4(1+2\lambda)} \mathbb{E}_{\boldsymbol{\sigma}} \mathbb{E}_{S_{\boldsymbol{\sigma}}} \left[ \mathbb{E}_{\boldsymbol{x}} [\mathbb{1}\{\widehat{g}(\boldsymbol{x}) \neq g^*(\boldsymbol{x})\}] \Big| \boldsymbol{\sigma} \right] \\
&\geq \frac{\lambda}{4(1+2\lambda)} \mathbb{E}_{\boldsymbol{\sigma}} \mathbb{E}_{S_{\boldsymbol{\sigma}}} \left\{ \sum_{j=2}^{d} \mathbb{P}_{\boldsymbol{x}} [\boldsymbol{x} \in \Omega^j] \mathbb{P}_{\boldsymbol{x}} \left[ \widehat{g}(\boldsymbol{x}) \neq g^*(\boldsymbol{x}) \big| \boldsymbol{x} \in \Omega^j \right] \Big| \boldsymbol{\sigma} \right\} \\
&\geq \frac{\varepsilon}{4(1+2\lambda)d} \sum_{j=2}^{d} \mathbb{E}_{\boldsymbol{\sigma}} \left\{ \mathbb{E}_{S_{\boldsymbol{\sigma}}} \left[ \mathbb{P}_{\boldsymbol{x}} \left[ \widehat{g}(\boldsymbol{x}) \neq g^*(\boldsymbol{x}) \big| \boldsymbol{x} \in \Omega^j \right] \right] \Big| \boldsymbol{\sigma} \right\}
\end{aligned}
\tag{12}
$$

In the last inequality we use the fact that $\mathbb{P}_{\boldsymbol{x}}[\boldsymbol{x} \in \boldsymbol{\Omega}^j] = \epsilon/(\lambda d)$ for $j \geq 2$.

Let $\boldsymbol{\sigma}^{/j}$ be a Rademacher vector conditional on coordinates $\{1, ..., j-1, j+1, ...d\}$. Let $\boldsymbol{\sigma}^{\{-j\}}$ be a vector equal to $\boldsymbol{\sigma}$ except at the $j$th entry. Above equation becomes:

$$
\begin{aligned}
&\mathbb{E}_{\boldsymbol{\sigma}} \mathbb{E}_{S_{\boldsymbol{\sigma}}} [R(\widehat{g}, f^*, \beta) - R(g^*, f^*, \beta)] \\
&\geq \frac{\varepsilon}{8(1+2\lambda)d} \sum_{j=2}^{d} \mathbb{E}_{\boldsymbol{\sigma}^{/j}} \left\{ \mathbb{E}_{S_{\boldsymbol{\sigma}}} \left[ \mathbb{P}_{\boldsymbol{x}} \left[ \widehat{g}(\boldsymbol{x}) \neq g^*(\boldsymbol{x}) \big| \boldsymbol{x} \in \Omega^j \right] \right] \right. \\
&\quad + \mathbb{E}_{S_{\boldsymbol{\sigma}^{\{-j\}}}} \left[ \mathbb{P}_{\boldsymbol{x}} \left[ \widehat{g}(\boldsymbol{x}) \neq g^*(\boldsymbol{x}) \big| \boldsymbol{x} \in \Omega^j \right] \right] \Big| \boldsymbol{\sigma}^{/j} \right\} \\
&= \frac{1}{2^{d-1}} \sum_{\boldsymbol{\sigma}^{/j} \in \{+1,-1\}^{d-1}} \frac{\varepsilon}{8(1+2\lambda)d} \sum_{j=2}^{d} \left\{ \mathbb{P}_{S_{\boldsymbol{\sigma}}, \boldsymbol{x}} \left[ \widehat{g}(\boldsymbol{x}) \neq g^*(\boldsymbol{x}) \big| \boldsymbol{x} \in \Omega^j \right] \right. \\
&\quad + \mathbb{P}_{S_{\boldsymbol{\sigma}^{\{-j\}}}, \boldsymbol{x}} \left[ \widehat{g}(\boldsymbol{x}) \neq g^*(\boldsymbol{x}) \big| \boldsymbol{x} \in \Omega^j \right] \right\}
\end{aligned}
\tag{13}
$$

We make our notation more specific. Let $\mathcal{A}(S_{\boldsymbol{\sigma}}) = \widehat{g}_{\boldsymbol{\sigma}}$ and $\mathcal{A}(S_{\boldsymbol{\sigma}^{\{-j\}}}) = \widehat{g}_{\boldsymbol{\sigma}^{-j}}$. Notice that $g^*(x)$ also depends on $\sigma$. We let $g^*_{\boldsymbol{\sigma}}$ be $g^*(x)$ conditioned on $\sigma$ and $g^*_{\boldsymbol{\sigma}^{-j}}$ be $g^*(x)$ conditioned on $\sigma^{\{-j\}}$. In particular, for all $\boldsymbol{x} \in \Omega^j$, $g^*_{\boldsymbol{\sigma}^{-j}}(\boldsymbol{x}) \neq g^*_{\boldsymbol{\sigma}}(\boldsymbol{x})$ could happen only when $\alpha \geq \|\boldsymbol{x}\|$ or $\|\boldsymbol{x}\| \geq 1 - \alpha$. So equation 13 becomes

$$
\begin{aligned}
&\mathbb{E}_{\boldsymbol{\sigma}} \mathbb{E}_{S_{\boldsymbol{\sigma}}} [R(\widehat{g}, f^*, \beta) - R(g^*, f^*, \beta)] \\
&\geq \frac{1}{2^{d-1}} \sum_{\boldsymbol{\sigma}^{/j} \in \{+1,-1\}^{d-1}} \frac{\varepsilon}{8(1+2\lambda)d} \sum_{j=2}^{d} \left\{ \mathbb{P}_{S_{\boldsymbol{\sigma}}, \boldsymbol{x}} \left[ \widehat{g}_{\boldsymbol{\sigma}}(\boldsymbol{x}) \neq g^*_{\boldsymbol{\sigma}}(\boldsymbol{x}) \big| \boldsymbol{x} \in \Omega^j \right] \right. \\
&\quad + \mathbb{P}_{S_{\boldsymbol{\sigma}^{\{-j\}}}, \boldsymbol{x}} \left[ \widehat{g}_{\boldsymbol{\sigma}^{-j}}(\boldsymbol{x}) \neq g^*_{\boldsymbol{\sigma}^{-j}}(\boldsymbol{x}) \big| \boldsymbol{x} \in \Omega^j \right] \right\} \\
&= \frac{1}{2^{d-1}} \sum_{\boldsymbol{\sigma}^{/j} \in \{+1,-1\}^{d-1}} \frac{\alpha\varepsilon}{8(1+2\lambda)d} \sum_{j=2}^{d} \left\{ \mathbb{P}_{S_{\boldsymbol{\sigma}}, \boldsymbol{x}} \left[ \widehat{g}_{\boldsymbol{\sigma}}(\boldsymbol{x}) \neq g^*_{\boldsymbol{\sigma}}(\boldsymbol{x}) \big| \boldsymbol{x} \in \Omega^j, \alpha \geq \|\boldsymbol{x}\|, or, \|\boldsymbol{x}\| \geq 1 - \alpha \right] \right. \\
&\quad + \mathbb{P}_{S_{\boldsymbol{\sigma}^{\{-j\}}}, \boldsymbol{x}} \left[ \widehat{g}_{\boldsymbol{\sigma}^{-j}}(\boldsymbol{x}) \neq g^*_{\boldsymbol{\sigma}^{-j}}(\boldsymbol{x}) \big| \boldsymbol{x} \in \Omega^j, \alpha \geq \|\boldsymbol{x}\|, or, \|\boldsymbol{x}\| \geq 1 - \alpha \right] \right\} \\
&= \frac{1}{2^{d-1}} \sum_{\boldsymbol{\sigma}^{/j} \in \{+1,-1\}^{d-1}} \frac{\alpha\varepsilon}{8(1+2\lambda)d} \sum_{j=2}^{d} \left\{ \mathbb{P}_{S_{\boldsymbol{\sigma}}, \boldsymbol{x}} \left[ \widehat{g}_{\boldsymbol{\sigma}}(\boldsymbol{x}) \neq g^*_{\boldsymbol{\sigma}}(\boldsymbol{x}) \big| \boldsymbol{x} \in \Omega^j, \alpha \geq \|\boldsymbol{x}\|, or, \|\boldsymbol{x}\| \geq 1 - \alpha \right] \right. \\
&\quad + \mathbb{P}_{S_{\boldsymbol{\sigma}^{\{-j\}}}, \boldsymbol{x}} \left[ \widehat{g}_{\boldsymbol{\sigma}^{-j}}(\boldsymbol{x}) \neq -g^*_{\boldsymbol{\sigma}}(\boldsymbol{x}) \big| \boldsymbol{x} \in \Omega^j, \alpha \geq \|\boldsymbol{x}\|, or, \|\boldsymbol{x}\| \geq 1 - \alpha \right] \right\}
\end{aligned}
\tag{14}
$$

Next we make Equation 14 independent of $x$.

$$
\begin{aligned}
&\mathbb{E}_{\boldsymbol{\sigma}}\mathbb{E}_{S_{\boldsymbol{\sigma}}}[R(\widehat{g}, f^*, \beta) - R(g^*, f^*, \beta)] \\
&\geq \frac{1}{2^{d-1}} \sum_{\boldsymbol{\sigma}/j \in \{+1,-1\}^{d-1}} \frac{\alpha\varepsilon}{8(1+2\lambda)d} \sum_{j=2}^{d} \Big\{ \mathbb{P}_{S_{\boldsymbol{\sigma}},\boldsymbol{x}}\big[\widehat{g}_{\boldsymbol{\sigma}}(\boldsymbol{x}) \neq g_{\boldsymbol{\sigma}}^*(\boldsymbol{x}) \big| \boldsymbol{x} \in \Omega^j, \alpha \geq \|\boldsymbol{x}\|, or, \|\boldsymbol{x}\| \geq 1-\alpha\big] \\
&\quad + \mathbb{P}_{S_{\boldsymbol{\sigma}\{-j\}},\boldsymbol{x}}\big[\widehat{g}_{\boldsymbol{\sigma}^{-j}}(\boldsymbol{x}) \neq -g_{\boldsymbol{\sigma}}^*(\boldsymbol{x}) \big| \boldsymbol{x} \in \Omega^j, \alpha \geq \|\boldsymbol{x}\|, or, \|\boldsymbol{x}\| \geq 1-\alpha\big] \Big\} \\
&= \frac{1}{2^{d-1}} \sum_{\boldsymbol{\sigma}/j \in \{+1,-1\}^{d-1}} \frac{\alpha\varepsilon}{8(1+2\lambda)d} \sum_{j=2}^{d} \Big\{ \mathbb{E}_{S_{\boldsymbol{\sigma}},\boldsymbol{x}}\big[\mathbb{1}\{\widehat{g}_{\boldsymbol{\sigma}}(\boldsymbol{x}) \neq g_{\boldsymbol{\sigma}}^*(\boldsymbol{x})\} \big| \boldsymbol{x} \in \Omega^j, \alpha \geq \|\boldsymbol{x}\|, or, \|\boldsymbol{x}\| \geq 1-\alpha\big] \\
&\quad + \mathbb{E}_{S_{\boldsymbol{\sigma}\{-j\}},\boldsymbol{x}}\big[\mathbb{1}\{\widehat{g}_{\boldsymbol{\sigma}^{-j}}(\boldsymbol{x}) \neq -g_{\boldsymbol{\sigma}}^*(\boldsymbol{x})\} \big| \boldsymbol{x} \in \Omega^j, \alpha \geq \|\boldsymbol{x}\|, or, \|\boldsymbol{x}\| \geq 1-\alpha\big] \Big\} \\
&= \frac{1}{2^{d-1}} \sum_{\boldsymbol{\sigma}/j \in \{+1,-1\}^{d-1}} \frac{\alpha\varepsilon}{8(1+2\lambda)d} \sum_{j=2}^{d} \Big\{ \mathbb{E}_{S_{\boldsymbol{\sigma}},\boldsymbol{x}}\big[\mathbb{1}\{\widehat{g}_{\boldsymbol{\sigma}} \neq g_{\boldsymbol{\sigma}}^*\} \big| \boldsymbol{x} \in \Omega^j, \alpha \geq \|\boldsymbol{x}\|, or, \|\boldsymbol{x}\| \geq 1-\alpha\big] \\
&\quad + \mathbb{E}_{S_{\boldsymbol{\sigma}\{-j\}},\boldsymbol{x}}\big[\mathbb{1}\{\widehat{g}_{\boldsymbol{\sigma}^{-j}} \neq -g_{\boldsymbol{\sigma}}^*\} \big| \boldsymbol{x} \in \Omega^j, \alpha \geq \|\boldsymbol{x}\|, or, \|\boldsymbol{x}\| \geq 1-\alpha\big] \Big\} \\
&\overset{*}{=} \frac{1}{2^{d-1}} \sum_{\boldsymbol{\sigma} \in \{+1,-1\}^{d-1}} \frac{\alpha\varepsilon}{8(1+2\lambda)d} \sum_{j=2}^{d} \Big\{ \mathbb{P}_{S_{\boldsymbol{\sigma}}}\big[\widehat{g}_{\boldsymbol{\sigma}} \neq g_{\boldsymbol{\sigma}}^*\big] + \mathbb{P}_{S_{\boldsymbol{\sigma}\{-j\}}}\big[\widehat{g}_{\boldsymbol{\sigma}^{-j}} \neq -g_{\boldsymbol{\sigma}}^*\big] \Big\} \\
&= \frac{1}{2^{d-1}} \sum_{\boldsymbol{\sigma} \in \{+1,-1\}^{d-1}} \frac{\alpha\varepsilon}{8(1+2\lambda)d} \sum_{j=2}^{d} \Big\{ 1 - \mathbb{P}_{S_{\boldsymbol{\sigma}}}\big[\widehat{g}_{\boldsymbol{\sigma}} \neq -g_{\boldsymbol{\sigma}}^*\big] + \mathbb{P}_{S_{\boldsymbol{\sigma}\{-j\}}}\big[\widehat{g}_{\boldsymbol{\sigma}^{-j}} \neq -g_{\boldsymbol{\sigma}}^*\big] \Big\} \\
&\geq \frac{1}{2^{d-1}} \sum_{\boldsymbol{\sigma}/j \in \{+1,-1\}^{d-1}} \frac{\alpha\varepsilon}{8(1+2\lambda)d} \sum_{j=2}^{d} \Big\{ 1 - \|Q_{\sigma}^{(n)} - Q_{\sigma\{-j\}}^{(n)}\|_{TV} \Big\}
\end{aligned}
$$

(15)

where $Q_{\sigma}^{(n)}, Q_{\sigma\{-j\}}^{(n)}$ is the product distribution of $n$ samples for $S_{\sigma}$ and $S_{\sigma^{-j}}$. The last step of inequality follows from the Le Cam's method. In the Equation $*$ we use the fact that for all $\boldsymbol{x}$ s.t., $\|x\| \leq \alpha$ or $\|x\| \geq 1-\alpha$, $\mathbb{1}\{\widehat{g}_{\boldsymbol{\sigma}^j}(\boldsymbol{x}) \neq g_{\boldsymbol{\sigma}^j}^*(\boldsymbol{x})\} = \mathbb{1}\{\widehat{g}_{\boldsymbol{\sigma}^j} \neq g_{\boldsymbol{\sigma}^j}^*\}$ is free of $\boldsymbol{x}$.

Let $Q_{\sigma}$ be distribution of $S_{\sigma}$ and $Q_{\sigma\{-j\}}$ be distribution of $S_{\sigma^{-j}}$. The total variation distance can be bounded using the Hellinger distance, which is denoted as $\mathcal{H}(\cdot, \cdot)$. Below we bound the TV distance using Hellinger distance.

$$
\begin{aligned}
&\|Q_{\sigma}^{(n)} - Q_{\sigma\{-j\}}^{(n)}\|_{TV} \\
&\leq \mathcal{H}(Q_{\sigma}^{(n)}, Q_{\sigma\{-j\}}^{(n)})\sqrt{1 - \frac{\mathcal{H}^2(Q_{\sigma}^{(n)}, Q_{\sigma\{-j\}}^{(n)})}{4}} \\
&\leq \sqrt{n}\mathcal{H}(Q_{\sigma}, Q_{\sigma\{-j\}})\sqrt{1 - \frac{\mathcal{H}^2(Q_{\sigma}^{(n)}, Q_{\sigma\{-j\}}^{(n)})}{4}} \\
&\overset{\mathcal{H}^2(Q_{\sigma}^{(n)}, Q_{\sigma\{-j\}}^{(n)}) \leq n\mathcal{H}^2(Q_{\sigma}, Q_{\sigma\{-j\}})}{\leq} \sqrt{n}\mathcal{H}(Q_{\sigma}, Q_{\sigma\{-j\}})
\end{aligned}
$$

(16)

Now we bound the Hellinger distance.

$$
\begin{aligned}
&\mathcal{H}^2(Q_\sigma, Q_{\sigma\{-j\}}) \\
&= \int_{\boldsymbol{x},y} \left(\sqrt{Q_\sigma(\boldsymbol{x},y)} - \sqrt{Q_{\sigma\{-j\}}(\boldsymbol{x},y)}\right)^2 d\boldsymbol{x}dy \\
&= \int_{\boldsymbol{x}\in\Omega^j, \|\boldsymbol{x}\|\le\alpha} \int_{y=f^*(\boldsymbol{x})} \left(\sqrt{Q_\sigma(\boldsymbol{x},y)} - \sqrt{Q_{\sigma\{-j\}}(\boldsymbol{x},y)}\right)^2 d\boldsymbol{x}dy \\
&\quad + \int_{\boldsymbol{x}\in\Omega^j, \|\boldsymbol{x}\|\ge 1-\alpha} \int_{y=f^*(\boldsymbol{x})} \left(\sqrt{Q_\sigma(\boldsymbol{x},y)} - \sqrt{Q_{\sigma\{-j\}}(\boldsymbol{x},y)}\right)^2 d\boldsymbol{x}dy \\
&\quad + \int_{\boldsymbol{x}\in\Omega^j, \|\boldsymbol{x}\|\le\alpha} \int_{y\ne f^*(\boldsymbol{x})} \left(\sqrt{Q_\sigma(\boldsymbol{x},y)} - \sqrt{Q_{\sigma\{-j\}}(\boldsymbol{x},y)}\right)^2 d\boldsymbol{x}dy \\
&\quad + \int_{\boldsymbol{x}\in\Omega^j, \|\boldsymbol{x}\|\ge 1-\alpha} \int_{y\ne f^*(\boldsymbol{x})} \left(\sqrt{Q_\sigma(\boldsymbol{x},y)} - \sqrt{Q_{\sigma\{-j\}}(\boldsymbol{x},y)}\right)^2 d\boldsymbol{x}dy \\
&= \frac{\alpha\varepsilon}{d\lambda}\left\{\left(\sqrt{\frac{3}{4}+\frac{\lambda}{2}} - \sqrt{\frac{3}{4}-\frac{\lambda}{2}}\right)^2 + \left(\sqrt{\frac{1}{4}+\frac{\lambda}{2}} - \sqrt{\frac{1}{4}-\frac{\lambda}{2}}\right)^2\right\} \\
&\le \frac{3\alpha\varepsilon\lambda}{d}
\end{aligned}
\tag{17}
$$

Thus we can bound the total variation distance as:

$$
\|Q_\sigma^{(n)} - Q_{\sigma\{-j\}}^{(n)}\|_{TV} \le \sqrt{\frac{3n\alpha\varepsilon\lambda}{d}}
\tag{18}
$$

Note inequality 18 together with inequality 13 we have and inequality 15

$$
\begin{aligned}
&\mathbb{E}_\sigma \mathbb{E}_{S_\sigma}[R(\mathcal{A}(S_\sigma),\beta) - R(g^*,f^*,\beta)] \\
&= \mathbb{E}_\sigma \mathbb{E}_{S_\sigma}[R(\widehat{g},f^*,\beta) - R(g^*,f^*,\beta)] \\
&\ge \frac{d-1}{d}\frac{\alpha\varepsilon}{8(1+2\lambda)}\left(1 - \sqrt{\frac{3n\alpha\varepsilon\lambda}{d}}\right)
\end{aligned}
\tag{19}
$$

Above implies $\sup_{\boldsymbol{\sigma}\in\{+1,-1\}^d} \mathbb{E}_{S_\sigma}[R(\mathcal{A}(S_\sigma),f^*,\beta) - R(g^*,f^*,\beta)] \ge \mathbb{E}_\sigma \mathbb{E}_{S_\sigma}[R(\mathcal{A}(S_\sigma),\beta) - R(g^*,f^*,\beta)] \ge \frac{d-1}{d}\frac{\alpha\varepsilon}{8(1+2\lambda)}\left(1 - \sqrt{\frac{3n\alpha\varepsilon\lambda}{d}}\right)$. Since $|\mathcal{G}| = 2^d$, any algorithm $\mathcal{A}$ will needs number of samples at least $n = \Omega\left(\frac{\log|\mathcal{G}|}{\lambda\varepsilon\alpha}\right)$ so that there is a hope to achieve

$$
\sup_\sigma \mathbb{E}_{S_\sigma}[R(\mathcal{A}(S_\sigma),\beta)] - R(g^*,f^*,\beta) \le \frac{\alpha\varepsilon}{32(1+2\lambda)}.
$$

Replacing $\alpha\varepsilon$ with $\alpha$ finishes the proof.

**Remark 3.** *From the second inequality in Equation 12, it can be observed that the construction of information theoretic lower bound for risk function $R(g,\beta)$ can also be applied to construction an $\Omega(\log(|\mathcal{G}|/(\lambda\varepsilon)))$ sample complexity lower bound for $\mathbb{E}_{\boldsymbol{x}}[g(\boldsymbol{x}) \ne g^*(\boldsymbol{x})]$. Thus our Corollary 1 also achieves minimax-optimal rate for recovering $g^*$ for family of Noise Generative Process.*

$\square$

### A.3 PROOF OF SAMPLE COMPLEXITY UPPER BOUND

Here we prove Theorem 1 in which we bound the risk gap $R(g; f^*, \beta) - R(g^*; f^*, \beta)$. Recall that the empirical version of the selector loss is

$$
R_{S_n}(g; f, \beta) = \frac{1}{n}\sum_{i=1}^n \left\{\beta\mathbb{1}\{g(\boldsymbol{x}_i) = 1\}\mathbb{1}\{f(\boldsymbol{x}_i) \ne y_i\} + \mathbb{1}\{g(\boldsymbol{x}_i) = 1\}\mathbb{1}\{f(\boldsymbol{x}_i) = y_i\}\right\}.
$$

Our high level approach is as follows. We first analyze the gap between $R_{S_n}(\widehat{g}; f^*, \beta)$ and $R_{S_n}(g^*; f^*, \beta)$ and provide an upper bound for it. Then we use this upper bound to get an upper bound for the gap between $R(\widehat{g}; f^*, \beta)$ and $R(g^*; f^*, \beta)$ using concentration properties and Bernstein inequality.

**CASE I: $\widehat{f}(\cdot) \in \mathcal{F}$ and $\mathbb{E}_{\boldsymbol{x}}[\widehat{f}(\boldsymbol{x}) \neq f^*(\boldsymbol{x})] \leq \frac{\varepsilon}{8\beta}$ with probability at least $1 - \delta$.** To upper bound $R_{S_n}(\widehat{g}; f^*, \beta) - R_{S_n}(g^*; f^*, \beta)$, we use $R_{S_n}(\widehat{g}; \widehat{f}, \beta)$ and $R_{S_n}(g^*; \widehat{f}, \beta)$ as a middle step. Since $\widehat{g}$ is the empirical risk minimizer, we have

$$R_{S_n}(\widehat{g}; \widehat{f}, \beta) \leq R_{S_n}(g^*; \widehat{f}, \beta). \tag{20}$$

Next we leverage on the fact that $\widehat{f}$ is consistent with $f$ to establish an inequality in following fashion:

$$R_{S_n}(\widehat{g}; f^*, \beta) \leq R_{S_n}(g^*; f^*, \beta) + const \cdot \varepsilon$$

Note we have that

$$R_{S_n}(\widehat{g}; \widehat{f}, \beta)$$
$$= \frac{1}{n} \sum_{i=1}^{n} \mathbb{1}\left\{\widehat{f}(\boldsymbol{x}_i) \neq f^*(\boldsymbol{x}_i)\right\}\left\{\beta \mathbb{1}\{\widehat{g}(\boldsymbol{x}_i) = 1\}\mathbb{1}\{\widehat{f}(\boldsymbol{x}_i) \neq y_i\} + \mathbb{1}\{\widehat{g}(\boldsymbol{x}_i) = 1\}\mathbb{1}\{\widehat{f}(\boldsymbol{x}_i) = y_i\}\right\}$$
$$+ \frac{1}{n} \sum_{i=1}^{n} \mathbb{1}\left\{\widehat{f}(\boldsymbol{x}_i) = f^*(\boldsymbol{x}_i)\right\}\left\{\beta \mathbb{1}\{\widehat{g}(\boldsymbol{x}_i) = 1\}\mathbb{1}\{\widehat{f}(\boldsymbol{x}_i) \neq y_i\} + \mathbb{1}\{\widehat{g}(\boldsymbol{x}_i) = 1\}\mathbb{1}\{\widehat{f}(\boldsymbol{x}_i) = y_i\}\right\} \tag{21}$$

Recall that

$$R_{S_n}(\widehat{g}; f^*, \beta) = \frac{1}{n} \sum_{i=1}^{n} \left[\beta \mathbb{1}\{\widehat{g}(\boldsymbol{x}_i) = 1\}\mathbb{1}\{f^*(\boldsymbol{x}_i) \neq y_i\} + \mathbb{1}\{\widehat{g}(\boldsymbol{x}_i) = 1\}\mathbb{1}\{f^*(\boldsymbol{x}_i) = y_i\}\right].$$

So

$$R_{S_n}(\widehat{g}; \widehat{f}, \beta)$$
$$= R_{S_n}(\widehat{g}; f^*, \beta)$$
$$- \frac{1}{n} \sum_{i=1}^{n} \mathbb{1}\left\{\widehat{f}(\boldsymbol{x}_i) \neq f^*(\boldsymbol{x}_i)\right\}\left\{\beta \mathbb{1}\{\widehat{g}(\boldsymbol{x}_i) = 1\}\mathbb{1}\{f^*(\boldsymbol{x}_i) \neq y_i\} + \mathbb{1}\{\widehat{g}(\boldsymbol{x}_i) = 1\}\mathbb{1}\{f^*(\boldsymbol{x}_i) = y_i\}\right\}$$
$$+ \frac{1}{n} \sum_{i=1}^{n} \mathbb{1}\left\{\widehat{f}(\boldsymbol{x}_i) \neq f^*(\boldsymbol{x}_i)\right\}\left\{\beta \mathbb{1}\{\widehat{g}(\boldsymbol{x}_i) = 1\}\mathbb{1}\{\widehat{f}(\boldsymbol{x}_i) \neq y_i\} + \mathbb{1}\{\widehat{g}(\boldsymbol{x}_i) = 1\}\mathbb{1}\{\widehat{f}(\boldsymbol{x}_i) = y_i\}\right\}$$
$$\geq R_{S_n}(\widehat{g}; f^*, \beta) - \frac{\beta - 1}{n} \sum_{i=1}^{n} \mathbb{1}\left\{\widehat{f}(\boldsymbol{x}_i) \neq f^*(\boldsymbol{x}_i)\right\} \tag{22}$$

Recall that in the theorem assumptions we have $\mathbb{E}_{\boldsymbol{x}}[\widehat{f}(\boldsymbol{x}) \neq f^*(\boldsymbol{x})] \leq \frac{\varepsilon}{8\beta}$ with probability at least $1 - \delta$. By Lemma 2, if $n \geq \frac{24\beta^2 \log(|\mathcal{F}|/\delta)}{\varepsilon}$ we have with probability at least $1 - \delta$, $\frac{1}{n}\sum_{i=1}^{n} \mathbb{1}\{\widehat{f}(\boldsymbol{x}_i) \neq f^*(\boldsymbol{x}_i)\} \leq \frac{\varepsilon}{4\beta}$, so we have

$$R_{S_n}(\widehat{g}; \widehat{f}, \beta) \geq R_{S_n}(\widehat{g}; f^*, \beta) - \varepsilon/4$$

With a similar approach we get that

$$R_{S_n}(g^*; \widehat{f}, \beta) \leq R_{S_n}(g^*; f^*, \beta) + \varepsilon/4$$

Thus using (20) we have following inequality holds with probability at least $1 - \delta$

$$R_{S_n}(\widehat{g}; f^*, \beta) \leq R_{S_n}(g^*; f^*, \beta) + \varepsilon/2. \tag{23}$$

To get a bound for $R(\widehat{g}; f^*, \beta) - R(g^*; f^*, \beta)$, we first define $\ell(g; f, \boldsymbol{x}, y) = \beta \mathbb{1}\{g(\boldsymbol{x}) = 1\}\mathbb{1}\{f(\boldsymbol{x}) \neq y\} + \mathbb{1}\{g(\boldsymbol{x}) = 1\}\mathbb{1}\{f(\boldsymbol{x}) = y\}$. Note that at this point we think of $\beta$ as fixed and so we have not included it in the arguments of $\ell(\cdot)$ for simplicity.

Observe that $R_{S_n}(g, f^*, \beta) = \frac{1}{n} \sum_{i=1}^n \ell(g; f^*, \boldsymbol{x}_i, y)$ and $R(g, f^*, \beta) = \mathbb{E}_{\boldsymbol{x}, y}\left[\ell(g; f^*, \boldsymbol{x}, y)\right]$ for any $g$. First we have the following simple inequality directly taken from (23).

$$
\begin{aligned}
&R(\widehat{g}, f^*, \beta) - R(g^*, f^*, \beta) \\
&= \mathbb{E}_{\boldsymbol{x}, y}\ell(\widehat{g}; f^*, \boldsymbol{x}, y) - \mathbb{E}_{\boldsymbol{x}, y}\ell(g^*; f^*, \boldsymbol{x}, y) \\
&\leq R_{S_n}(g^*; f^*, \beta) - R_{S_n}(\widehat{g}; f^*, \beta) - (\mathbb{E}_{\boldsymbol{x}, y}\ell(g^*; f^*, \boldsymbol{x}, y) - \mathbb{E}_{\boldsymbol{x}, y}\ell(\widehat{g}; f^*, \boldsymbol{x}, y)) + \varepsilon/2
\end{aligned}
\tag{24}
$$

By defining $\Delta\ell(g^*, g, \boldsymbol{x}, y) = \ell(g^*; f^*, \boldsymbol{x}, y) - \ell(g; f^*, \boldsymbol{x}, y)$ for any $g$, we can express the above inequality as follows:

$$
\begin{aligned}
&\mathbb{E}_{\boldsymbol{x}, y}\ell(\widehat{g}; f^*, \boldsymbol{x}, y) - \mathbb{E}_{\boldsymbol{x}, y}\ell(g^*; f^*, \boldsymbol{x}, y) \\
&\leq \frac{1}{n} \sum_{i=1}^n \Delta\ell(g^*, \widehat{g}, \boldsymbol{x}_i, y) - \mathbb{E}_{\boldsymbol{x}, y}\Delta\ell(g^*; \widehat{g}, \boldsymbol{x}, y) + \varepsilon/2
\end{aligned}
\tag{25}
$$

To bound $\frac{1}{n} \sum_{i=1}^n \Delta\ell(g^*, \widehat{g}, \boldsymbol{x}_i, y) - \mathbb{E}_{\boldsymbol{x}, y}\Delta\ell(g^*; \widehat{g}, \boldsymbol{x}, y)$ with high probability over all $S_n$, we need to find a bound on $\frac{1}{n} \sum_{i=1}^n \Delta\ell(g^*, g, \boldsymbol{x}_i, y) - \mathbb{E}_{\boldsymbol{x}, y}\Delta\ell(g^*; g, \boldsymbol{x}, y)$ that is true for all $g$ simultaneously with high probability. We have :

$$
\begin{aligned}
&\mathbb{P}_{S_n}\left[\exists g \in \mathcal{G}, \left\{\frac{1}{n} \sum_{i=1}^n \Delta\ell(g^*, g; \boldsymbol{x}_i, y_i) - \mathbb{E}_{\boldsymbol{x}, y}[\Delta\ell(g^*, g; \boldsymbol{x}, y)]\right.\right. \\
&\qquad \left.\left. \geq \frac{n}{\beta} \log(|\mathcal{G}|/\delta) + \sqrt{\frac{2\boldsymbol{Var}(\Delta(g^*, g; \boldsymbol{x}, y)) \log(|\mathcal{G}|/\delta)}{n}}\right\}\right] \\
&\leq \sum_{g \in \mathcal{G}} \mathbb{P}_{S_n}\left[\frac{1}{n} \sum_{i=1}^n \Delta\ell(g^*, g; \boldsymbol{x}_i, y_i) - \mathbb{E}_{\boldsymbol{x}, y}[\Delta\ell(g^*, g; \boldsymbol{x}, y)]\right. \\
&\qquad \left. \geq \frac{n}{\beta} \log(|\mathcal{G}|/\delta) + \sqrt{\frac{2\boldsymbol{Var}(\Delta(g^*, g; \boldsymbol{x}, y)) \log(|\mathcal{G}|/\delta)}{n}}\right]
\end{aligned}
\tag{26}
$$

Now to bound $\frac{1}{n} \sum_{i=1}^n \Delta\ell(g^*, g, \boldsymbol{x}_i, y) - \mathbb{E}_{\boldsymbol{x}, y}\Delta\ell(g^*; g, \boldsymbol{x}, y)$ we use Bernstein inequality. For that we need to bound $\boldsymbol{Var}_{\boldsymbol{x}, y}[\Delta\ell(g^*, g, \boldsymbol{x}, y)]$. We first expand $\Delta\ell(g^*, g, \boldsymbol{x}, y)$.

$$
\begin{aligned}
\Delta\ell(g^*, g, \boldsymbol{x}, y) &= \ell(g^*; f^*, \boldsymbol{x}, y) - \ell(g; f^*, \boldsymbol{x}, y) \\
&= \beta \mathbb{1}\{g^*(\boldsymbol{x}) = 1\}\mathbb{1}\{f^*(\boldsymbol{x}) \neq y\} + \mathbb{1}\{g^*(\boldsymbol{x}) = 1\}\mathbb{1}\{f^*(\boldsymbol{x}) = y\} \\
&\quad - \beta \mathbb{1}\{g(\boldsymbol{x}) = 1\}\mathbb{1}\{f^*(\boldsymbol{x}) \neq y\} - \mathbb{1}\{g(\boldsymbol{x}) = 1\}\mathbb{1}\{f^*(\boldsymbol{x}) = y\}
\end{aligned}
$$

So we have

$$
\begin{aligned}
&\Delta^2\ell(g^*, g, \boldsymbol{x}, y) \\
&= \beta^2\left[\mathbb{1}\{g^*(\boldsymbol{x}) = 1\} + \mathbb{1}\{g(\boldsymbol{x}) = 1\} - 2\mathbb{1}\{g(\boldsymbol{x}) = 1\}\mathbb{1}\{g^*(\boldsymbol{x}) = 1\}\right]\mathbb{1}\{f^*(\boldsymbol{x}) \neq y\} \\
&\quad + \left[\mathbb{1}\{g^*(\boldsymbol{x}) = 1\} + \mathbb{1}\{g(\boldsymbol{x}) = 1\} - 2\mathbb{1}\{g(\boldsymbol{x}) = 1\}\mathbb{1}\{g^*(\boldsymbol{x}) = 1\}\right]\mathbb{1}\{f^*(\boldsymbol{x}) = y\} \\
&= \left(\beta^2\mathbb{1}\{f^*(\boldsymbol{x}) \neq y\} + \mathbb{1}\{f^*(\boldsymbol{x}) = y\}\right)\left[\mathbb{1}\{g(\boldsymbol{x}) = 1\}\mathbb{1}\{g^*(\boldsymbol{x}) \neq 1\} + \mathbb{1}\{g(\boldsymbol{x}) \neq 1\}\mathbb{1}\{g^*(\boldsymbol{x}) = 1\}\right] \\
&\leq \beta^2\mathbb{1}\{g^*(\boldsymbol{x}) \neq g(\boldsymbol{x})\}
\end{aligned}
\tag{27}
$$

Hence we conclude that $\boldsymbol{Var}_{\boldsymbol{x}, y}[\Delta\ell(g^*, g, \boldsymbol{x}, y)] \leq \mathbb{E}_{\boldsymbol{x}, y}\Delta^2\ell(g^*, g, \boldsymbol{x}, y) \leq \beta^2\mathbb{E}_{\boldsymbol{x}}[\mathbb{1}\{g^*(\boldsymbol{x}) \neq g(\boldsymbol{x})\}]$. On the other hand, Equation 8 implies that $R(g; f^*, \beta) - R(g^*; f^*, \beta) \geq \frac{\lambda}{1+2\lambda}\mathbb{E}_{\boldsymbol{x}}[\mathbb{1}\{g^*(\boldsymbol{x}) \neq g(\boldsymbol{x})\}]$. Thus we can use the following inequality to achieve fast rate of convergence using the Bernstein Inequality:

$$
\boldsymbol{Var}_{\boldsymbol{x}, y}[\Delta\ell(g^*, g; \boldsymbol{x}, y)] \leq \frac{\beta^2(1 + 2\lambda)}{\lambda}\{R(g; f^*, \beta)) - R(g^*; f^*, \beta)\}.
\tag{28}
$$

We use following version of the Bernstein Inequality, with $X_1, ..., X_n$ i.i.d random variable uniformly bounded by $b$:

$$\mathbb{P}\left(\frac{1}{n}\sum_{i=1}^{n}X_i - \mathbb{E}[X] < \frac{b}{n}\log(1/\delta) + \sqrt{\frac{2\boldsymbol{Var(X)}\log(1/\delta)}{n}}\right) \geq 1-\delta$$

Using union bounds, the Bernstein Inequality implies that with probability for all $g \in \mathcal{G}$ simultaneously:

$$\mathbb{P}_{S_n}\left[\left\{\frac{1}{n}\sum_{i=1}^{n}\Delta\ell(g^*, g; \boldsymbol{x}_i, y_i) - \mathbb{E}_{\boldsymbol{x},y}[\Delta\ell(g^*, g; \boldsymbol{x}, y)]\right\}\right.$$
$$\left.\leq \frac{\beta}{n}\log\left(|\mathcal{G}|/\delta\right) + \sqrt{\frac{2\boldsymbol{Var}(\Delta(g^*, g; \boldsymbol{x}, y))\log\left(|\mathcal{G}|/\delta\right)}{n}}\right] \geq 1-\delta \quad (29)$$

Thus by applying inequality 29 with $\widehat{g}$ we have:

$$\mathbb{P}_{S_n}\left[\left\{\frac{1}{n}\sum_{i=1}^{n}\Delta\ell(g^*, \widehat{g}; \boldsymbol{x}_i, y_i) - \mathbb{E}_{\boldsymbol{x},y}[\Delta\ell(g^*, \widehat{g}; \boldsymbol{x}, y)]\right\}\right.$$
$$\left.\leq \frac{\beta}{n}\log\left(|\mathcal{G}|/\delta\right) + \sqrt{\frac{2\boldsymbol{Var}(\Delta(g^*, \widehat{g}; \boldsymbol{x}, y))\log\left(|\mathcal{G}|/\delta\right)}{n}}\right] \geq 1-\delta$$

By inequality 23, we have $\frac{1}{n}\sum_{i=1}^{n}\Delta\ell(g^*, \widehat{g}; \boldsymbol{x}_i, y_i) = R_{S_n}(g^*; f^*, \beta) - R_{S_n}(\widehat{g}; f^*, \beta) \geq -\varepsilon/2$ holds with probability $1-\delta$. Note $R(\widehat{g}; f^*, \beta)) - R(g^*; f^*, \beta) = -\mathbb{E}_{\boldsymbol{x},y}[\Delta\ell(g^*, \widehat{g}; \boldsymbol{x}, y)]$. So we have

$$\mathbb{P}_{S_n}\left[\left\{R(\widehat{g}; f^*, \beta) - R(g^*; f^*, \beta)\right\}\right.$$
$$\left.\leq \frac{\beta}{n}\log\left(|\mathcal{G}|/\delta\right) + \sqrt{\frac{2\beta^2(1+2\lambda)\{R(\widehat{g}; f^*, \beta)) - R(g^*; f^*, \beta)\}\log\left(|\mathcal{G}|/\delta\right)}{\lambda n}} + \varepsilon/2\right]$$
$$\geq 1-2\delta$$

The choice of $n \geq \frac{16\beta^2\log(\frac{|\mathcal{G}|}{\delta})}{\lambda\varepsilon}$ ensures that with probability at least $1-2\delta$, $R(\widehat{g}; f^*, \beta) - R(g^*; f^*, \beta) \leq \varepsilon$.

**CASE II: $\lambda = \frac{1}{2}$, $\widehat{f}(\cdot) \in \mathcal{F}$ that satisfies $\mathbb{E}_{\boldsymbol{x}}[\widehat{f}(\boldsymbol{x}) \neq f^*(\boldsymbol{x})|\boldsymbol{x} \in \Omega_I] \leq \frac{\varepsilon}{8\beta\alpha\beta}$ with probability at least $1-\delta$.** When $\lambda = \frac{1}{2}$, achieving $\mathbb{E}_{\boldsymbol{x}}[\widehat{f}(\boldsymbol{x}) \neq f^*(\boldsymbol{x})] \leq \frac{\varepsilon}{8\beta}$ is in general impossible. One can only approximate $f^*(\cdot)$ on the informative support $\Omega_I$ since $y$ is generated by coin flipping when $\boldsymbol{x} \in \Omega_U$. For simplicity of analysis, we introduce a 'pseudo' version of $f^*(\cdot)$ denoted as $\widetilde{f}^*$. Let $\widetilde{\mathcal{F}}$ be following hypothesis class:

$$\left\{\widetilde{f}\Big|\widetilde{f}(\boldsymbol{x}) = \begin{cases} f_1(\boldsymbol{x}), & \boldsymbol{x} \in \Omega_U \\ f_2(\boldsymbol{x}), & \boldsymbol{x} \in \Omega_I \end{cases}, f_1 \in \mathcal{F}, f_2 \in \mathcal{F}\right\}$$

and we let $\widetilde{f}^*(\cdot)$ be:

$$\widetilde{f}^*(\boldsymbol{x}) = \begin{cases} \widehat{f}(\boldsymbol{x}), & \boldsymbol{x} \in \Omega_U \\ f^*(\boldsymbol{x}), & \boldsymbol{x} \in \Omega_I \end{cases}$$

Clearly, $\widetilde{f}^* \in \widetilde{\mathcal{F}}$. Note such hypothesis class is only introduced in analysis and is potentially impractical. The cardinality of hypothesis class $|\widetilde{\mathcal{F}}| \leq |\mathcal{F}|^2$. The construction of $\widetilde{f}^*$ is to make

$R(g(\boldsymbol{x}), \widetilde{f}^*, \beta) - R(g^*(\boldsymbol{x}), \widetilde{f}^*, \beta) \geq 0$ for all $g \in \mathcal{G}$. To see this:

$$
\begin{aligned}
& R(g(\boldsymbol{x}), \widetilde{f}^*, \beta) - R(g^*(\boldsymbol{x}), \widetilde{f}^*, \beta) \\
=& \mathbb{E}_{\boldsymbol{x},y}\big[\beta \mathbb{1}\{g(\boldsymbol{x}) = 1\}\mathbb{1}\{g^*(\boldsymbol{x}) \neq 1\}\mathbb{1}\{\widetilde{f}^*(\boldsymbol{x}) \neq y\} \\
& -\mathbb{1}\{g(\boldsymbol{x}) = 1\}\mathbb{1}\{g^*(\boldsymbol{x}) \neq 1\}\mathbb{1}\{\widetilde{f}^*(\boldsymbol{x}) = y\} \\
& +\mathbb{1}\{g(\boldsymbol{x}) \neq 1\}\mathbb{1}\{g^*(\boldsymbol{x}) = 1\}\mathbb{1}\{\widetilde{f}^*(\boldsymbol{x}) = y\} \\
& -\beta \mathbb{1}\{g(\boldsymbol{x}) \neq 1\}\mathbb{1}\{g^*(\boldsymbol{x}) = 1\}\mathbb{1}\{\widetilde{f}^*(\boldsymbol{x}) \neq y\}\big] \\
=& \left\{\frac{\beta}{2} - \frac{1}{2}\right\}\mathbb{E}_{\boldsymbol{x}}\big[\mathbb{1}\{g(\boldsymbol{x}) = 1\}\mathbb{1}\{g^*(\boldsymbol{x}) \neq 1\} \\
& +\mathbb{E}_{\boldsymbol{x}}\big[\mathbb{1}\{g(\boldsymbol{x}) \neq 1\}\mathbb{1}\{g^*(\boldsymbol{x}) = 1\}\big] \\
\geq& 0
\end{aligned}
\tag{30}
$$

Meanwhile $\widetilde{f}^*(\cdot)$ also satisfies the property that $\mathbb{E}_{\boldsymbol{x}}[\widetilde{f}^*(\boldsymbol{x}) \neq \widehat{f}(\boldsymbol{x})] \leq \frac{\varepsilon}{8\beta}$ with probability at least $1 - \delta$. Thus by Lemma 2, if $n \geq \frac{24\beta^2 \log(\frac{|\mathcal{F}|}{\delta})}{\varepsilon}$ we have with probability at least $1 - \delta$, $\frac{1}{n}\sum_{i=1}^{n} \mathbb{1}\{\widehat{f}(\boldsymbol{x}_i) \neq \widetilde{f}^*(\boldsymbol{x}_i)\} \leq \frac{\varepsilon}{8\beta}$.

The rest of the proof is the same as the proof in **CASE I** by replacing $f^*$ with $\widetilde{f}^*$, leveraging on the fact that $R(g^*(\boldsymbol{x}), \widetilde{f}^*, \beta) = R(g^*(\boldsymbol{x}), f^*, \beta)$.

**Remark 4.** *Let us point out that our proposed selective strategy is different from the consistent selective strategy in (El-Yaniv et al., 2010). Instead of rejecting by looking for consistent output from all hypothesis in the version space, our approach deals with one single reasonably accurate hypothesis (the empirical minimizer). We leverage empirical mistakes made by the predictor in order to learn a selector, aiming to reject (only) the mistakes in a data driven manner. This avoids dealing with the issues found in Theorem 14 in (El-Yaniv et al., 2010), where the selector fails to select any data points.*

**Remark 5.** *In (Cortes et al., 2016), a second hypothesis for the selector is introduced and analyzed, and at the same time, multiple commonly used loss functions are scrutinized and generalization results are provided. The major difference between this work and (Cortes et al., 2016; Geifman & El-Yaniv, 2019) is the motivation pertaining to selective learning. While in (Cortes et al., 2016; Geifman & El-Yaniv, 2019) the selective loss is designed from a coverage ratio perspective, i.e. one wants to trade coverage ratio for a higher precision (selective loss), our approach is designed to distinguish data that is naturally unlearnable and unpredictable. This difference leads to an alternative theoretical result. While the analysis in (Cortes et al., 2016) focuses on selective risk, our theoretical analysis focuses on the quality of the selector in distinguishing informative/uninformative data, without adjusting rejection cost given by human.*

## A.4 MISSING PROOF FOR COROLLARY 6

It can be easily verified that $\beta = 3$ is in the interval $\beta \in \left[\frac{3-2\lambda}{1+2\lambda} + \lambda, \min(\frac{3+2\lambda}{1-2\lambda} - \frac{\lambda}{1-4\lambda^2}, 10)\right]$. By the choice of $\beta$, from (9) we have

$$
R(\widehat{g}, f^*, \beta) - R(g^*, f^*, \beta) \geq \frac{\lambda}{4(1+2\lambda)}\mathbb{E}_{\boldsymbol{x}}[\mathbb{1}\{\widehat{g}(\boldsymbol{x}) \neq g^*(\boldsymbol{x})\}],
$$

together with the conclusion in Theorem 1 that

$$
R(\widehat{g}; f^*, \beta) - R(g^*; f^*, \beta) \leq \varepsilon
$$

we can conclude that Equation 6 holds.

## A.5 MISSING PROOF FOR CONTROLLING CONDITIONAL RISK $\mathbb{E}_{\boldsymbol{x}}[\widehat{f}(\boldsymbol{x}) \neq f^*(\boldsymbol{x})|\boldsymbol{x} \in \Omega_I]$

**Lemma 1** (Sauer–Shelah Lemma(See (Blum et al., 2016; Mohri et al., 2018; Sauer, 1972))). *Let $d_{vc}(\mathcal{G})$ be the VC-dimension of hypothesis class $\mathcal{G}$, for all $n \in \mathbb{N}$,*

$$
\mathcal{B}_{\mathcal{G}}(n) \leq \sum_{i=0}^{d_{vc}} \binom{n}{i} \leq \left(\frac{en}{d_{vc}(\mathcal{G})}\right)^{d_{vc}(\mathcal{G})}
$$

**Definition 4** (Growth Function(Vapnik & Chervonenkis, 2015)). *Let $\mathcal{G}$ be the hypothesis class of function $f$ and $\mathcal{F}_{\boldsymbol{x}_1,...,\boldsymbol{x}_n} = \{(f(\boldsymbol{x}_1),...,f(\boldsymbol{x}_n)) : f \in \mathcal{F}\} \subseteq \{+1,-1\}^n$. The growth function is defined to be the maximum number of ways in which $n$ points can be classified by the function class: $\mathcal{B}_{\mathcal{F}}(n) = \sup_{\boldsymbol{x}_1,...,\boldsymbol{x}_n} |\mathcal{F}_{\boldsymbol{x}_1,...,\boldsymbol{x}_n}|$.*

**Theorem 3.** *For every $\varepsilon > 0$, there is a $\delta > 0$ such that under Assumption 1, given a set of samples $S_n = \{(\boldsymbol{x}_1,y_1),...,(\boldsymbol{x}_n,y_n)\}$ drawn i.i.d. from the Noisy Generative Process and*

$$\widehat{f} = \arg\min_{f \in \mathcal{F}} \sum_{i=1}^{n} \mathbb{1}\{f(\boldsymbol{x}_i) \neq y_i\},$$

*if $n$ is chosen such that*

$$n \geq \frac{32\left[d_{VC}(\mathcal{F})\log(\frac{1}{\varepsilon}) + \log(\frac{1}{\delta})\right]}{\epsilon^2\alpha^2},$$

*then with probability at least $1 - 2\delta$:*

$$\mathbb{E}_{\boldsymbol{x},y}[\widehat{f}(\boldsymbol{x}) \neq y] \leq \frac{1}{2}(1-\alpha) + 2\epsilon\alpha.$$

*Furthermore,*

$$\mathbb{P}_{\boldsymbol{x}}[f^*(\boldsymbol{x}) \neq \widehat{f}(\boldsymbol{x})|\boldsymbol{x} \in \Omega_I] \leq 2\epsilon$$

*Proof.* We first bound the probability of the event that $\mathbb{E}_{\boldsymbol{x},y}[\widehat{f}(\boldsymbol{x}) \neq y] \leq \frac{1}{2}(1-\alpha) + 2\epsilon\alpha$.

By Lemma 3 and Hoeffding inequality we have:

$$\mathbb{P}_{S_n}[\sup_{f \in \mathcal{F}} |\frac{1}{n}\sum_{i=1}^{n} \mathbb{1}\{f(\boldsymbol{x}_i) \neq y_i\} - \mathbb{E}_{\boldsymbol{x},y}[\mathbb{1}\{f(\boldsymbol{x}) \neq y\}]| \geq t] \leq 4\mathcal{B}_{\mathcal{F}}(2n)e^{-\frac{nt^2}{32}} \tag{31}$$

By setting $t = \alpha\epsilon$ and $n \geq \frac{32(4\log(\mathcal{B}_{\mathcal{F}}(2n)) + \log(\frac{1}{\delta}))}{\alpha^2\epsilon^2}$ we have with probability of at least $1 - \delta$:

$$\frac{1}{n}\sum_{i=1}^{n} \mathbb{1}\{\widehat{f}(\boldsymbol{x}_i) \neq y_i\} - \mathbb{E}_{\boldsymbol{x},y}[\mathbb{1}\{\widehat{f}(\boldsymbol{x}) \neq y\}] \leq \frac{\alpha\epsilon}{2}$$

The term $\mathcal{B}_{\mathcal{F}}(2n)$ could be bounded by Sauer's lemma. Next we apply the fact that $\widehat{f} = \arg\min_{f \in \mathcal{F}} \frac{1}{n}\sum_{i=1}^{n} \mathbb{1}\{\widehat{f}(\boldsymbol{x}_i) \neq y_i\}$. We have:

$$\mathbb{E}_{\boldsymbol{x},y}[\mathbb{1}\{\widehat{f}(\boldsymbol{x}) \neq y\}] \leq \frac{\alpha\epsilon}{2} + \frac{1}{n}\sum_{i=1}^{n} \mathbb{1}\{\widehat{f}(\boldsymbol{x}_i) \neq y_i\} \leq \frac{\alpha\epsilon}{2} + \frac{1}{n}\sum_{i=1}^{n} \mathbb{1}\{f^*(\boldsymbol{x}_i) \neq y_i\}$$

Since $\frac{1}{n}\sum_{i=1}^{n} \mathbb{1}\{f^*(\boldsymbol{x}_i) \neq y_i\} \leq \frac{1}{2}(1-\alpha) + \epsilon\alpha$ with failure probability at most $\delta$ (Lemma 5), we have with probability at least $1 - 2\delta$:

$$\mathbb{E}_{\boldsymbol{x},y}[\mathbb{1}\{\widehat{f}(\boldsymbol{x}) \neq y\}] \leq \frac{1}{2}(1-\alpha) + 2\epsilon\alpha.$$

Next we prove the claim that:

$$\mathbb{P}_{\boldsymbol{x} \sim \mathcal{D}_I}[f^*(\boldsymbol{x}) \neq \widehat{f}(\boldsymbol{x})] \leq 2\epsilon.$$

Since $\mathbb{E}_{\boldsymbol{x},y}[\mathbb{1}\{\widehat{f}(\boldsymbol{x}) \neq y\}] \leq \frac{1}{2}(1-\alpha) + 2\epsilon\alpha$:

$$\begin{aligned}
&\mathbb{E}_{\boldsymbol{x},y}[\mathbb{1}\{\widehat{f}(\boldsymbol{x}) \neq y\}] \\
=&\mathbb{E}_{(\boldsymbol{x},y) \sim \mathcal{D}_\alpha}[\mathbb{1}\{\widehat{f}(\boldsymbol{x}) \neq y\}] \\
=&\underbrace{\mathbb{E}_{(\boldsymbol{x},y) \sim \mathcal{D}_\alpha}[\mathbb{1}\{\widehat{f}(\boldsymbol{x}) \neq y\}|\boldsymbol{x} \in \Omega_U]}_{\frac{1}{2}}\underbrace{\mathbb{P}_{(\boldsymbol{x},y) \sim \mathcal{D}_\alpha}[\boldsymbol{x} \in \Omega_U]}_{1-\alpha} \\
&+\underbrace{\mathbb{E}_{(\boldsymbol{x},y) \sim \mathcal{D}_\alpha}[\mathbb{1}\{\widehat{f}(\boldsymbol{x}) \neq y\}|\boldsymbol{x} \in \Omega_I]}_{\mathbb{P}_{(\boldsymbol{x},y) \sim \mathcal{D}_\alpha}[\mathbb{1}\{\widehat{f}(\boldsymbol{x}) \neq f^*(\boldsymbol{x})\}|\boldsymbol{x} \in \Omega_I]}\underbrace{\mathbb{P}_{(\boldsymbol{x},y) \sim \mathcal{D}_\alpha}[\boldsymbol{x} \in \Omega_I]}_{\alpha} \\
=&\frac{1}{2}(1-\alpha) + \alpha\mathbb{P}_{\boldsymbol{x} \sim \mathcal{D}_\alpha}[\widehat{f}(\boldsymbol{x}) \neq f^*(\boldsymbol{x})|\boldsymbol{x} \in \Omega_I] \\
\leq&\frac{1}{2}(1-\alpha) + 2\epsilon\alpha \\
\Longrightarrow&\mathbb{P}_{\boldsymbol{x} \sim \mathcal{D}_\alpha}[\mathbb{1}\{\widehat{f}(\boldsymbol{x}) \neq f^*(\boldsymbol{x})\}|\boldsymbol{x} \in \Omega_I] \leq 2\epsilon
\end{aligned} \tag{32}$$

$\square$

## A.6 EXTENTION TO VC-CLASS

In order to leverage the margin condition of distribution of $z$ to obtain a minimax-optimal generalization rate, we leverage on the Local Rademacher Average tool. Our analysis tool largely follows from (Bousquet et al., 2003; Bartlett et al., 2005). Throughout this section, $\lesssim$ and $\gtrsim$ represent as shorthand for the $\leq$ and $\geq$ that ignores universal constants.

**Definition 5** ($L_2$-Covering Number). *(Wellner et al., 2013) Let $\boldsymbol{x}_{1:n}$ be set of points. A set of $U \subseteq \mathbb{R}^n$ is an $\varepsilon$-cover w.r.t $L_2$-norm of $\mathcal{F}$ on $x_{1:n}$, if $\forall f \in \mathcal{F}$, $\exists u \in U$, s.t. $\sqrt{\frac{1}{n}\sum_{i=1}^n |[u]_i - f(x_i)|^2} \leq \varepsilon$, where $[u]_i$ is the $i$-th coordinate of $u$. We define the covering number $\mathcal{N}_2(\varepsilon, \mathcal{F}, \boldsymbol{x}_{1:n})$ :*

$$\mathcal{N}_2(\varepsilon, \mathcal{F}, \boldsymbol{x}_{1:n}) := \min\{|U|: U \text{ is an } \varepsilon\text{-cover of } \mathcal{F} \text{ on } x_{1:n}\}$$

*Let $\mathcal{N}_2(\varepsilon, \mathcal{F}, n)$ be the maximum cardinality of $\mathcal{N}_2(\varepsilon, \mathcal{F}, \boldsymbol{x}_{1:n})$ over all $\boldsymbol{x}_{1:n}$. Formally $\mathcal{N}_2(\varepsilon, \mathcal{F}, n)$ is defined as:*

$$\mathcal{N}_2(\varepsilon, \mathcal{F}, n) := \sup_{\boldsymbol{x}_{1:n} \in \mathcal{X}^n} \min\{|U|: U \text{ is an } \varepsilon\text{-cover of } \mathcal{F} \text{ on } x_{1:n}\}$$

**Definition 6** (Local Rademacher Average (Bartlett et al., 2005; Bousquet et al., 2003)). *Let $\sigma_{1:n}$ be Rademacher sequence of length $n$, the Empirical Local Rademacher Complexity at distributional and empirical radius $r \geq 0$ for the class $\mathcal{F}$ are defined as*

$$\mathcal{R}_n(\mathcal{F}, Pf^2 \leq r) \equiv \mathbb{E}_{\sigma_{1:n}}\big[\sup_{f \in \mathcal{F}, \mathbb{E}_{\boldsymbol{x}} f(\boldsymbol{x})^2 \leq r} \frac{1}{n}\sum_{i=1}^n \sigma_i f(\boldsymbol{x}_i)\big]$$

$$\mathcal{R}_n(\mathcal{F}, P_n f^2 \leq r) \equiv \mathbb{E}_{\sigma_{1:n}}\big[\sup_{f \in \mathcal{F}, \frac{1}{n}\sum_{i=1}^n f(\boldsymbol{x}_i)^2 \leq r} \frac{1}{n}\sum_{i=1}^n \sigma_i f(\boldsymbol{x}_i)\big]$$

*and their distributional Average as: $\mathcal{R}(\mathcal{F}, Pf^2 \leq r) \equiv \mathbb{E}_{S_n}[\mathcal{R}_n(\mathcal{F}, Pf^2 \leq r)]$ and $\mathcal{R}(\mathcal{F}, P_n f^2 \leq r) \equiv \mathbb{E}_{S_n}[\mathcal{R}_n(\mathcal{F}, P_n f^2 \leq r)]$.*

**Definition 7** (Star Hull). *(Bartlett et al., 2005; Bousquet et al., 2003) The star hull of set of functions $\mathcal{F}$ is defined as*

$$*\mathcal{F} \equiv \{\alpha f : f \in \mathcal{F}, \alpha \in [0, 1]\}$$

**Definition 8** (Sub-Root Function). *(Bartlett et al., 2005; Massart & Nédélec, 2006; Bousquet et al., 2003) A function $\psi : \mathbb{R} \to \mathbb{R}$ is sub-root if*

- *$\psi$ is non-decreasing*

- *$\psi$ is non-negative*

- *$\psi(r)/\sqrt{r}$ is non-increasing*

*And we say $r^*$ is a fixed point of $\psi$ if $\psi(r^*) = r^*$.*

**Theorem 4.** *[Risk Bound VC-Class] Let $S_n = \{(\boldsymbol{x}_i, y_i)\}_{i=1}^n$ be i.i.d sample from Data Generative Process described in Definition 1 under Assumption 1, with $f^*(\cdot) \in \mathcal{F}$ and $g^*(\cdot) \in \mathcal{G}$ with VC-dimension $d_{VC}(\mathcal{F}) < \infty$ $d_{VC}(\mathcal{G}) < \infty$. Given $\lambda$, let $\beta \in \big[\frac{3-2\lambda}{1+2\lambda} + \lambda, \min(\frac{3+2\lambda}{1-2\lambda} - \frac{\lambda}{1-4\lambda^2}, 10)\big]$. For any $\widehat{f}(\cdot) \in \mathcal{F}$, let $\widehat{g} = \arg\min_{g \in \mathcal{G}} R_{S_n}(g; \widehat{f}, \beta)$. Then for any $\varepsilon > 0$, there is a $\delta > 0$ such that the following holds: For*

$$n \gtrsim \max\big\{\frac{\beta^4 d_{VC}(\mathcal{G})\log(\frac{1}{\varepsilon}) + \beta^4 \log(\frac{1}{\delta})}{\lambda\varepsilon}, \frac{\beta d_{VC}(\mathcal{F})\log(\frac{d_{VC}(\mathcal{F})}{\varepsilon}) + \beta\log(\frac{1}{\delta})}{\varepsilon}\big\}.$$

*and for $\widehat{f}$ that satisfies one of the following condition:*

- *For any $\widehat{f}(\cdot) \in \mathcal{F}$ that satisfies $\mathbb{E}_{\boldsymbol{x}}[\widehat{f}(\boldsymbol{x}) \neq f^*(\boldsymbol{x})] \lesssim \frac{\varepsilon}{\beta}$ with probability at least $1 - \delta$,*

- *If $\lambda = \frac{1}{2}$, for any $\widehat{f}(\cdot) \in \mathcal{F}$ that satisfies $\mathbb{E}_{\boldsymbol{x}}[\widehat{f}(\boldsymbol{x}) \neq f^*(\boldsymbol{x})|\boldsymbol{x} \in \Omega_I] \lesssim \frac{\varepsilon}{\beta\alpha}$ with probability at least $1 - \delta$,*

*The following holds with probability at least $1 - 3\delta$:*

$$R(\widehat{g}; f^*, \beta) - R(g^*; f^*, \beta) \lesssim \varepsilon$$

*Proof.* The major difference from the proof for Theorem 1 is the fact that $\mathcal{G}$ and $\mathcal{F}$ are not finite hypothesis class. To achieve fast generalization rate, we leverage the Local Rademacher Complexity Tool from (Bartlett et al., 2005).

**CASE I** : $\widehat{f}(\cdot) \in \mathcal{F}$ and $\mathbb{E}_{\boldsymbol{x}}[\widehat{f}(\boldsymbol{x}) \neq f^*(\boldsymbol{x})] \lesssim \frac{\varepsilon}{\beta}$ with probability at least $1 - \delta$.

We use a proof similar to the one in Theorem 1 up to Equation 21. Since $\mathcal{F}$ is a VC-class, we will invoke Lemma 8 instead of Lemma 2. Since $n \gtrsim \frac{\beta(d_{VC}(\mathcal{F})\log(\frac{1}{\varepsilon}) + \log(\frac{1}{\delta}))}{\varepsilon}$, it can be achieved with probability at least $1 - \delta$ that

$$R_{S_n}(\widehat{g}; \widehat{f}, \beta) \geq R_{S_n}(\widehat{g}; f^*, \beta) - \varepsilon/4$$

and

$$R_{S_n}(g^*; \widehat{f}, \beta) \leq R_{S_n}(g^*; f^*, \beta) + \varepsilon/4.$$

Thus following hold with probability at least $1 - \delta$:

$$R_{S_n}(\widehat{g}; f^*, \beta) \leq R_{S_n}(g^*; f^*, \beta) + \frac{\varepsilon}{2}. \tag{33}$$

Next we turn to bound the risk gap using $R(\widehat{g}; f^*, \beta) - R(g^*; f^*, \beta)$ using concentration property of inequality 33. Similar to the proof in Theorem 1, we define $\ell(g; f, \boldsymbol{x}, y) = \beta \mathbb{1}\{g(\boldsymbol{x}) = 1\}\mathbb{1}\{f(\boldsymbol{x}) \neq y\} + \mathbb{1}\{g(\boldsymbol{x}) \neq 1\}\mathbb{1}\{f(\boldsymbol{x}) = y\}$. Based on $\ell$, we define following hypothesis class:

$$\Delta \circ \ell \circ \mathcal{G} \equiv \left\{ \Delta\ell(g; g^*, \boldsymbol{x}, y) = \ell(g; f^*, \boldsymbol{x}, y) - \ell(g^*; f^*, \boldsymbol{x}, y) : g \in \mathcal{G} \right\}. \tag{34}$$

To invoke Lemma 6, we need to establish some hypothesis class $\mathcal{H}$ that satisfies condition $\boldsymbol{Var}[h] \leq B\mathbb{E}[h]$. Next we show $\Delta \circ \ell \circ \mathcal{G}$ satisfies the condition that $\boldsymbol{Var}[h] \leq B\mathbb{E}[h]$ and thus we can apply Lemma 6 with $\mathcal{H} = \Delta \circ \ell \circ \mathcal{G}$. To begin with, one can apply Equation 27 to show that,

$$\mathbb{1}\{g^*(\boldsymbol{x}) \neq g(\boldsymbol{x})\} \leq \Delta^2\ell(g; g^*, \boldsymbol{x}, y) \leq \beta^2\mathbb{1}\{g^*(\boldsymbol{x}) \neq g(\boldsymbol{x})\}$$

Above implies that $\boldsymbol{Var}_{\boldsymbol{x},y}[\Delta\ell(g^*, g, \boldsymbol{x}, y)] \leq \mathbb{E}_{\boldsymbol{x},y}\Delta^2\ell(g^*, g, \boldsymbol{x}, y) \leq \beta^2\mathbb{E}_{\boldsymbol{x}}[\mathbb{1}\{g^*(\boldsymbol{x}) \neq g(\boldsymbol{x})\}]$.

On the other hand, Equation 8 implies that

$$R(g; f^*, \beta) - R(g^*; f^*, \beta) \geq \frac{\lambda}{1 + 2\lambda}\mathbb{E}_{\boldsymbol{x}}[\mathbb{1}\{g^*(\boldsymbol{x}) \neq g(\boldsymbol{x})\}].$$

Thus we have following holds:

$$\boldsymbol{Var}_{\boldsymbol{x},y}[\Delta\ell(g; g^*, \boldsymbol{x}, y)] \leq \frac{\beta^2(1 + 2\lambda)}{\lambda}\mathbb{E}_{\boldsymbol{x},y}\{\Delta\ell(g; g^*, \boldsymbol{x}, y)\} \tag{35}$$

Thus we can apply Lemma6 with $\mathcal{H} = \Delta \circ \ell \circ \mathcal{G}$, $T(h) = \mathbb{E}[h^2]$ and $B = \frac{\beta^2(1+2\lambda)}{\lambda}$.

Now we find a subroot function $\psi(r)$ that

$$\psi(r) \geq \frac{\beta^2(1 + 2\lambda)}{\lambda}\mathbb{E}\mathcal{R}_n\{\Delta\ell(g; g^*) \in \mathcal{H} : \mathbb{E}[h^2] \leq r\}.$$

To find $\psi(r)$, we show some analysis on the Local Rademacher Average $\mathbb{E}\mathcal{R}_n\{\Delta\ell(g; g^*) \in \mathcal{H} : \mathbb{E}[h^2] \leq r\}$.

$$
\begin{aligned}
\mathbb{E}\mathcal{R}_n(\Delta \circ \ell \circ \mathcal{G}, r) =& \mathbb{E}_{S_n \sigma_{1:n}} \big[ \sup_{g \in \mathcal{G}, \mathbb{E}_{\boldsymbol{x},y}[\Delta^2 \ell(g;g^*)] \leq r} \frac{1}{n} \sum_{i=1}^n \sigma_i \Delta \ell(g; g^*) \big] \\
\leq& \mathbb{E}_{S_n \sigma_{1:n}} \big[ \underbrace{\sup_{g \in \mathcal{G}, \mathbb{E}_{\boldsymbol{x}}[\mathbb{1}(g \neq g^*)] \leq r} \frac{1}{n} \sum_{i=1}^n \sigma_i \Delta \ell(g; g^*) \big]}_{\mathbb{1}(g \neq g^*) \leq \Delta^2 \ell(g;g^*)} \\
\leq& \beta \mathbb{E}_{S_n \sigma_{1:n}} \big[ \underbrace{\sup_{g \in \mathcal{G}, \mathbb{E}_{\boldsymbol{x}}[\mathbb{1}(g \neq g^*)] \leq r} \frac{1}{n} \sum_{i=1}^n \sigma_i \mathbb{1} g(\boldsymbol{x}_i) \neq g^*(\boldsymbol{x}_i)) \big]}_{\substack{|\Delta \ell(g_1;g^*) - \Delta \ell(g_2;g^*)| \leq \beta |\mathbb{1}(g_1 \neq g^*) - \mathbb{1}(g_2 \neq g^*)| \\ \text{Talagrand Contraction Inequality (Ledoux \& Talagrand, 1991)}}}
\end{aligned}
\tag{36}
$$

In the last inequality, we use the fact that

$$
|\Delta \ell(g_1; g^*) - \Delta \ell(g_2; g^*)| \leq |\ell(g_1) - \ell(g_2)| \leq \beta |\mathbb{1}(g_1 \neq g_2)| = \beta |\mathbb{1}(g_1 \neq g^*) - \mathbb{1}(g_2 \neq g^*)|.
$$

Define $\mathbb{1} \circ \mathcal{G} \equiv \mathbb{1}\{g(\boldsymbol{x}) \neq g^*(\boldsymbol{x}), g \in \mathcal{G}\}$. The indicator function $\mathbb{1}\{g(\boldsymbol{x}) \neq g^*(\boldsymbol{x})\}$ is a Boolean function taking $g$ as input, thus $d_{VC}(\mathbb{1} \circ \mathcal{G}) \leq d_{VC}(\mathcal{G})$. Thus we have

$$
\begin{aligned}
&\frac{\beta^2 (1 + 2\lambda)}{\lambda} \mathbb{E}\mathcal{R}_n \{\Delta \ell(g; g^*) \in \mathcal{H} : \mathbb{E}[h^2] \leq r\} \\
\leq& \frac{\beta^3 (1 + 2\lambda)}{\lambda} \mathbb{E}\mathcal{R}_n \{\mathbb{1}\{g(\boldsymbol{x}) \neq g^*(\boldsymbol{x})\} \in \mathbb{1} \circ \mathcal{G} : \mathbb{E}[\mathbb{1}\{g(\boldsymbol{x}) \neq g^*(\boldsymbol{x})\}] \leq r\}
\end{aligned}
\tag{37}
$$

Above implies that we can pick $\psi(r)$ to be

$$
\psi(r) = \frac{\beta^3 (1 + 2\lambda)}{\lambda} \mathbb{E}\mathcal{R}_n \{ * \mathbb{1} \circ \mathcal{G} : \mathbb{E}[\mathbb{1}\{g(\boldsymbol{x}) \neq g^*(\boldsymbol{x})\}] \leq r\} + \frac{+11\beta^2 \log n}{n}
\tag{38}
$$

By Equation 49, we have:

$$
\mathbb{E}_{\boldsymbol{x},y}[\Delta \ell(\widehat{g}; g^*, \boldsymbol{x}, y)] \leq \frac{2}{n} \sum_{i=1}^n \Delta \ell(\widehat{g}; g^*; \boldsymbol{x}_i, y_i) + \frac{1500\lambda}{\beta^2} r^* + \frac{\log(1/\delta)(11\beta + \frac{52}{\lambda})}{n}
\tag{39}
$$

By inequality 33, we have $\frac{1}{n} \sum_{i=1}^n \Delta \ell(\widehat{g}; g^*; \boldsymbol{x}_i, y_i) = R_{S_n}(\widehat{g}; f^*, \beta) - R_{S_n}(g^*; f^*, \beta) \leq \varepsilon/2$ holds with probability $1 - \delta$. By Lemma 7 we have $r^* \lesssim \frac{\beta^6}{\lambda^2} \frac{d_{VC}(\mathcal{G}) \log n}{n}$. Plugging in Equation 49 we have that $n \gtrsim \frac{\beta^4 (d_{VC}(\mathcal{G}) \log(\frac{1}{\varepsilon}) + \log(1/\delta))}{\lambda \varepsilon}$ suffices to achieve $\mathbb{E}_{\boldsymbol{x},y}[\Delta \ell(\widehat{g}; g^*, \boldsymbol{x}, y)] \lesssim \varepsilon$.

**CASE II:** $\lambda = \frac{1}{2}$, $\widehat{f}(\cdot) \in \mathcal{F}$ that satisfies $\mathbb{E}_{\boldsymbol{x}}[\widehat{f}(\boldsymbol{x}) \neq f^*(\boldsymbol{x}) | \boldsymbol{x} \in \Omega_I] \leq \frac{\varepsilon}{8\alpha\beta}$ with probability at least $1 - \delta$.
The proof is similar to the one in Theorem 1 except for that we need to bound the VC-dimension of pseudo hypothesis class $\mathcal{F}$. Since $\widetilde{f}$ can be viewed as Boolean function given $f_1(\boldsymbol{x}), f_2(\boldsymbol{x})$ as input, with two hypothesis $f_1, f_2 \in \mathcal{F}$, by Lemma 3.2.3 in (Blumer et al., 1989) we know $d_{VC}(\widetilde{\mathcal{F}}) \leq 2 d_{VC}(\mathcal{F}) \log(d_{VC}(\mathcal{F}))$. The rest of the proof follows from the one in Theorem 1. $\square$

*Next we present our extension of information theoretic lower bound to VC-class. The lower bounds suggest that the risk bound in Theorem 4 is tight up to some logarithmic factor.*

**Theorem 5.** *There exists noisy generative process defined in Definition 1 with $\Omega$ being $\mathcal{G}$-realizable, for any $\varepsilon \leq \lambda$, to achieve*

$$
\mathbb{E}_{S_n}[R(\mathcal{A}(S_n), f^*, \beta) - R(g^*, f^*, \beta)] \leq \frac{\varepsilon}{8(1 + 2\lambda)}
$$

*with $\beta \in \left[ \frac{3-2\lambda}{1+2\lambda} + \lambda, \frac{3+2\lambda}{1-2\lambda} - \frac{\lambda}{1-4\lambda^2} \right]$, any algorithm $\mathcal{A}$ will take at least $\frac{d_{VC}(\mathcal{G})}{\log(d_{VC}(\mathcal{G}))\lambda\varepsilon}$ many samples.*

*Proof.* The proof follows from the proof of Theorem 2 except for the fact that we need to have an upper bound on the VC-dimension of our hypothesis construction $\mathcal{G}$. Since $\mathcal{G}$ consists of composition of interval hypothesis and each individual interval has VC-dimension at most 3. By Lemma 3.2.3 in (Blumer et al., 1989) we know $d_{VC}(\mathcal{G}) \leq 6d \log(d)$ which implies a $\frac{d_{VC}(\mathcal{G})}{\log(d_{VC}(\mathcal{G}))\lambda\varepsilon}$ lower bound. $\square$

## A.7 TECHNICAL LEMMAS

**Lemma 2.** *Let $S_n = \{(\boldsymbol{x}_i, y_i)\}$ be i.i.d sample from Data Generative Process described in Definition 1. For every $\varepsilon > 0$, there exist a $\delta > 0$ such that if $n \geq \frac{3\log(\frac{|\mathcal{F}|}{\delta})}{\varepsilon}$, following inequality holds simultaneously for all $f \in \mathcal{F}$ with $|\mathcal{F}| < \infty$ with probability at least $1 - \delta$*

$$\frac{1}{n}\sum_{i=1}^{n}\mathbb{1}\{f(\boldsymbol{x}_i) \neq f^*(\boldsymbol{x}_i)\} < \mathbb{E}_{\boldsymbol{x}}\mathbb{1}\{f(\boldsymbol{x}) \neq f^*(\boldsymbol{x})\} + \varepsilon \tag{40}$$

*Proof.* By taking union bound one can ensure that

$$\mathbb{P}_{S_n}\left[\sup_{f\in\mathcal{F}}\left\{\left|\sum_{i=1}^{n}\mathbb{1}\{f(\boldsymbol{x}_i) \neq f^*(\boldsymbol{x}_i)\} - n\mathbb{E}_{\boldsymbol{x}}\mathbb{1}\{f(\boldsymbol{x}) \neq f^*(\boldsymbol{x})\}\right| \geq n\mathbb{E}_{\boldsymbol{x}}\mathbb{1}\{f(\boldsymbol{x}) \neq f^*(\boldsymbol{x})\} + n\varepsilon\right\}\right]$$

$$\leq\mathbb{P}_{S_n}\left[\bigcup_{f\in\mathcal{F}}\left\{\left|\sum_{i=1}^{n}\mathbb{1}\{f(\boldsymbol{x}_i) \neq f^*(\boldsymbol{x}_i)\} - n\mathbb{E}_{\boldsymbol{x}}\mathbb{1}\{f(\boldsymbol{x}) \neq f^*(\boldsymbol{x})\}\right| \geq n\mathbb{E}_{\boldsymbol{x}}\mathbb{1}\{f(\boldsymbol{x}) \neq f^*(\boldsymbol{x})\} + n\varepsilon\right\}\right]$$

$$\leq\sum_{f\in\mathcal{F}}\mathbb{P}_{S_n}\left[\left|\sum_{i=1}^{n}\mathbb{1}\{f(\boldsymbol{x}_i) \neq f^*(\boldsymbol{x}_i)\} - n\mathbb{E}_{\boldsymbol{x}}\mathbb{1}\{f(\boldsymbol{x}) \neq f^*(\boldsymbol{x})\}\right| \geq n\mathbb{E}_{\boldsymbol{x},y}\mathbb{1}\{f(\boldsymbol{x}) \neq f^*(\boldsymbol{x})\} + n\varepsilon\right]$$
$$\tag{41}$$

We next apply following version of Chernoff inequality with $a \geq 1$: Let $X = \sum_{i=1}^{n} X_i$ where $X_i \in \{0, 1\}$. Then

$$\mathbb{P}[X \geq (1+a)\mathbb{E}X] \leq \exp\left(-\frac{a^2}{2+a}\mathbb{E}X\right) \leq \exp\left(-\frac{a}{3}\mathbb{E}X\right)$$

$$\mathbb{P}[X \leq (1-a)\mathbb{E}X] \leq \exp\left(-\frac{a^2}{2}\mathbb{E}X\right) \leq \exp\left(-\frac{a}{3}\mathbb{E}X\right)$$

So we have

$$\mathbb{P}[|X - \mathbb{E}X| \geq a\mathbb{E}X] \leq \exp\left(-\frac{a}{3}\mathbb{E}X\right)$$

For any fixed $f \in \mathcal{F}$, let $a = \varepsilon/\mathbb{E}_{\boldsymbol{x}}\mathbb{1}[f(\boldsymbol{x}) \neq f^*(\boldsymbol{x})]$, by Chernoff Inequality we have

$$\mathbb{P}_{S_n}\left[\left|\sum_{i=1}^{n}\mathbb{1}\{f(\boldsymbol{x}_i) \neq f^*(\boldsymbol{x}_i)\} - n\mathbb{E}_{\boldsymbol{x},y}\mathbb{1}\{f(\boldsymbol{x}) \neq f^*(\boldsymbol{x})\}\right| \geq n\mathbb{E}_{\boldsymbol{x}}\mathbb{1}\{f(\boldsymbol{x}) \neq f^*(\boldsymbol{x})\} + n\varepsilon\right]$$

$$\leq \exp\left(-\frac{n\mathbb{E}_{\boldsymbol{x}}\mathbb{1}\{f(\boldsymbol{x}) \neq f^*(\boldsymbol{x})\}a}{3}\right) = \exp\left(-\frac{n\varepsilon}{3}\right)$$
$$\tag{42}$$

Using (41) and setting $\delta = |\mathcal{F}|\exp(-n\epsilon/3)$ finishes the proof. $\qquad\square$

**Lemma 3.** *Suppose $S_n = \{(\boldsymbol{x}_1, y_1), ..., (\boldsymbol{x}_n, y_n)\}$ are i.i.d sampled , $L(f, x, y) \in [0, b]$ and $L_{S_n}(f) = \frac{1}{n}\sum_{i=1}^{n} L(f, \boldsymbol{x}_i, y_i)$. Given parameter $t$ such that*

$$nt^2 \geq 2b^2$$

*then we have:*

$$\mathbb{P}_{S_n \sim \mathcal{D}}[\sup_{f\in\mathcal{F}}|L_{S_n}(f) - L(f)| \geq t] \leq 4\mathcal{B}_{\mathcal{F}}(2n)e^{-\frac{nt^2}{4b^2}}$$

**Proof**: For sample sets $S_n$ and $S'_n$, if we have $|L_{S_n}(f) - L(f)| \geq t$ and $|L_{S'_n}(f) - L(f)| \leq \frac{t}{2}$ then we get that $|L_{S_n} - L_{S'_n}| \geq \frac{t}{2}$. Thus we have

$$\mathbb{1}\{\sup_{f\in\mathcal{F}}|L_{S_n}(f) - L(f)| \geq t\} \cdot \mathbb{1}\{\sup_{f\in\mathcal{F}}|L_{S'_n}(f) - L(f)| \leq \frac{t}{2}\}$$

$$\leq \mathbb{1}\{\sup_{f\in\mathcal{F}}|L_{S_n}(f) - L_{S'_n}(f)| \geq \frac{t}{2}\}$$
$$\tag{43}$$

Taking expectation w.r.t $S_n \sim \mathcal{D}$ and $S'_n \sim \mathcal{D}$ we have

$$\mathbb{P}_{S_n \sim \mathcal{D}}\big[\sup_{f \in \mathcal{F}} |L_{S_n}(f) - L(f)| \geq t\big] \cdot \mathbb{P}_{S'_n \sim \mathcal{D}}\big[\sup_{f \in \mathcal{F}} |L_{S'_n}(f) - L(f)| \leq \frac{t}{2}\big]$$

$$\leq \mathbb{P}_{S_n, S'_n \sim \mathcal{D}}\big[\sup_{f \in \mathcal{F}} |L_{S_n}(f) - L_{S'_n}(f)| \geq \frac{t}{2}\big] \tag{44}$$

Next we lower bound $\mathbb{P}\big[\sup_{f \in \mathcal{F}} |L_{S_n}(f) - L(f)| \geq \frac{t}{2}\big]$. Since $L(f, x, y) \in [0, b]$ and so $Var(L(f, x, y)) \leq b^2/4$, using $nt^2 \geq 2b^2$ we have that:

$$\mathbb{P}_{S_n \sim \mathcal{D}}\big[\sup_{f \in \mathcal{F}} |L_{S_n}(f) - L(f)| \geq \frac{t}{2}\big] \leq \frac{4Var(L_{S_n})}{nt^2} \leq \frac{1}{2}$$

So we have $\mathbb{P}_{S'_n}\big[\sup_{f \in \mathcal{F}} |L_{S'_n}(f) - L(f)| \leq \frac{t}{2}\big] \geq \frac{1}{2}$. Combining this inequality with (44) we have

$$\mathbb{P}_{S_n \sim \mathcal{D}}[\sup_{f \in \mathcal{F}} |L_{S_n}(f) - L(f)| \geq t]$$

$$\leq 2\mathbb{P}_{S_n, S'_n \sim \mathcal{D}}[\sup_{f \in \mathcal{F}} |L_{S_n}(f) - L_{S'_n}(f)| \geq \frac{t}{2}]$$

$$= 2\mathbb{P}_{S_n, S'_n \sim \mathcal{D}}[\sup_{f(\boldsymbol{x}) \in \mathcal{F}_{S_{2n}}} |L_{S_n}(f) - L_{S'_n}(f)| \geq \frac{t}{2}]$$

$$\leq 2\mathbb{P}_{S_{2n}}\big[\mathbb{P}_{S_n = S_{2n} - S'_n}[\sup_{f(\boldsymbol{x}) \in \mathcal{F}_{S_{2n}}} |L_{S_n}(f) - L_{S'_n}(f)| \geq \frac{t}{2}|S_{2n}]\big]$$

$$\leq 2\mathbb{P}_{S_{2n}}\big[\bigcup_{f(\boldsymbol{x}) \in \mathcal{F}_{S_{2n}}} \mathbb{P}_{S_n = S_{2n} - S'_n}[|L_{S_n}(f) - L_{S'_n}(f)| \geq \frac{t}{2}|S_{2n}]\big] \tag{45}$$

$$\leq 2\mathbb{P}_{S_{2n}}\big[2|\mathcal{F}_{S_{2n}}|e^{-\frac{nt^2}{4b^2}}|S_{2n}]\big]$$

$$\leq 2\mathbb{P}_{S_{2n}}\big[\sup_{S_{2n}} |\mathcal{F}_{S_{2n}}|e^{-\frac{nt^2}{4b^2}}|S_{2n}]\big]$$

$$\leq 2\sup_{S_{2n}} |\mathcal{F}_{S_{2n}}|\mathbb{P}_{S_{2n}}\big[e^{-\frac{nt^2}{4b^2}}]\big]$$

$$\leq 2\mathcal{B}_{\mathcal{F}}(2n)e^{-\frac{nt^2}{4b^2}}$$

**Lemma 4** (Hoeffding's Inequality). *Let $Z_1, ..., Z_n$ be independent bounded random variables with $Z_i \in [a, b]$ for all $i$, where $-\infty < a < b < \infty$. Then for all $t > 0$:*

$$\mathbb{P}(\frac{1}{n}|\sum_{i=1}^{n} Z_i - \mathbb{E}[Z_i]| \geq t) \leq 2e^{-\frac{2nt^2}{(b-a)^2}} \tag{46}$$

**Lemma 5.** *Consider a set of samples $S = \{(\boldsymbol{x}_1, y_1), ..., (\boldsymbol{x}_n, y_n)\}$ drawn i.i.d. from the Noisy Generative Process and $f^*$ in the hypothesis class $\mathcal{F}$ satisfying $f(\boldsymbol{x}) \in \{-1, +1\}$. If:*

$$n \geq \frac{3\log(\frac{1}{\delta})}{\epsilon^2 \alpha^2}$$

*Then we have with probability at least $1 - \delta$ :*

$$\frac{1}{n}\sum_{i=1}^{n} \mathbb{1}\{f^*(\boldsymbol{x}_i) \neq y_i\} \leq \frac{1}{2}(1 - \alpha) + \alpha\varepsilon \tag{47}$$

**Proof**:

Since $\mathbb{1}\{f(\boldsymbol{x}) \neq y\}$ is bounded in the interval $[0, 1]$ and given $f^* \in \mathcal{F}$, $\mathbb{1}\{f(\boldsymbol{x}_i) \neq y_i\}, i \in [n]$ form a set of $n$ independent random variables. By setting $b - a = 1, t = \alpha\epsilon$ in Equation 46, the choice of

$n$ ensures that $\frac{-2nt^2}{(b-a)^2} \leq 6\log(\delta)$. Thus

$$\mathbb{P}_{S_n \sim \mathcal{D}_\alpha}[|\frac{1}{n}\sum_{i=1}^n \mathbb{1}\{f^*(\boldsymbol{x}_i) \neq y_i\} - \mathbb{E}_{\boldsymbol{x},y}[\mathbb{1}\{f^*(\boldsymbol{x}) \neq y\}]| \geq \epsilon\alpha] \leq \delta.$$

where we have

$$
\begin{aligned}
&\mathbb{E}_{(\boldsymbol{x},y)\sim\mathcal{D}_\alpha}[\mathbb{1}\{f^*(\boldsymbol{x}) \neq y\}] \\
=&\underbrace{\mathbb{E}_{(\boldsymbol{x},y)\sim\mathcal{D}_\alpha}[\mathbb{1}\{f^*(\boldsymbol{x}) \neq y\}|\boldsymbol{x} \in \Omega_U]\mathbb{P}[\boldsymbol{x} \in \Omega_U]}_{\frac{1}{2}\mathbb{P}[\boldsymbol{x}\in\Omega_U]:\text{ Since } y \text{ is labeled by coin flipping in } \Omega_U} \\
&+\underbrace{\mathbb{E}_{(\boldsymbol{x},y)\sim\mathcal{D}_\alpha}[\mathbb{1}\{f^*(\boldsymbol{x}) \neq y\}|\boldsymbol{x} \in \Omega_I]\mathbb{P}[\boldsymbol{x} \in \Omega_I]}_{0:\text{ Since } y \text{ is labeled by } f^* \text{ with 0 Bayes Risk in } \Omega_I} \\
=&\frac{1}{2}(1-\alpha)
\end{aligned}
\tag{48}
$$

This way we have:

$$\mathbb{P}_{S_n \sim \mathcal{D}_\alpha}[|\frac{1}{n}\sum_{i=1}^n \mathbb{1}\{f^*(\boldsymbol{x}_i) \neq y_i\} - \frac{1}{2}(1-\alpha)| \geq \epsilon\alpha] \leq \delta.$$

which implies that Equation 47 holds with probability at least $1 - \delta$. $\qquad\square$

**Lemma 6** (Theorem 3.3 in (Bartlett et al., 2005)). *Let $\mathcal{F}$ be a class of functions with range in $[a, b]$ and assume that there are some functional $T : \mathcal{H} \to \mathbb{R}^+$ and some constant $B$ such that for every $h \in \mathcal{H}$, $\boldsymbol{Var}(h) \leq T(h) \leq BP[h]$. Let $\psi$ be a subroot function and $r^*$ be the fixed point of $\psi$. Assume the $\psi$ satisfies, for any $r \geq r^*$,*

$$\psi(r) \geq B\mathbb{E}\mathcal{R}_n\{h \in \mathcal{H} : T(h) \leq r\}$$

*Then with $c_1 = 704$ and $c_2 = 26$, for any $K > 1$ and every $t > 1$ with probability at least $1 - e^{-t}$,*

$$\forall h \in \mathcal{H}, P[h] \leq \frac{K}{K-1}P_n h + \frac{c_1 K}{B}r^* + \frac{t(11(b-a) + c_2 BK)}{n} \tag{49}$$

*Also with probability at least $1 - e^{-t}$,*

$$\forall h \in \mathcal{H}, P_n[h] \leq \frac{K+1}{K}Ph + \frac{c_1 K}{B}r^* + \frac{t(11(b-a) + c_2 BK)}{n} \tag{50}$$

*where $Pf = \mathbb{E}_{\boldsymbol{x}}[h(\boldsymbol{x})]$ and $P_n = \frac{1}{n}\sum_{i=1}^n h(\boldsymbol{x}_i)$.*

**Lemma 7.** *Given hypothesis class $\mathcal{F} : \mathcal{X} \to [-b, b]$ with some universal constant $b$ and its VC-dimension $d_{VC}(\mathcal{F}) < \infty$. Define following sub-root function with $B \geq 1$:*

$$\psi(r) = 100B\mathbb{E}\mathcal{R}_n\{*\mathcal{F}, r\} + \frac{11b^2 \log n}{n}.$$

*Let $r^*$ be fixed point of $\psi(r)$ so that $r^* = \psi(r^*)$, suppose $n \geq d_{VC}(\mathcal{F})$, we have*

$$r^* \lesssim \frac{B^2 d_{VC}(\mathcal{F})\log(\frac{n}{d_{VC}(\mathcal{F})})}{n}$$

*Proof.* The proof largely follows from the proof in Corollary 3.7 in (Bartlett et al., 2005). We include here for completeness. Since $f$ is uniformly bounded by $b$, for any $r \geq \psi(r)$, Corollart 2.2 in (Bartlett et al., 2005) implies that with probability at least $1 - \frac{1}{n}$, $\{f \in *\mathcal{F} : Pf^2 \leq r\} \subseteq \{f \in *\mathcal{F} : P_n f^2 \leq 2r\}$. Let $\mathcal{E}$ be event that $\{f \in *\mathcal{F} : Pf^2 \leq r\} \subseteq \{f \in *\mathcal{F} : P_n f^2 \leq 2r\}$ holds, above implies

$$
\begin{aligned}
&\mathbb{E}\mathcal{R}_n\{*\mathcal{F}, Pf^2 \leq r\} \\
\leq&\mathbb{P}[\mathcal{E}]\mathbb{E}[\mathcal{R}_n\{*\mathcal{F}, Pf^2 \leq r\}|\mathcal{E}] + \mathbb{P}[\mathcal{E}^c]\mathbb{E}[\mathcal{R}_n\{*\mathcal{F}, Pf^2 \leq r\}|\mathcal{E}^c] \\
\leq&\mathbb{E}[\mathcal{R}_n\{*\mathcal{F}, P_n f^2 \leq 2r\}] + \frac{b}{n}
\end{aligned}
\tag{51}
$$

Since $r^* = \psi(r^*)$, $r^*$ satisfies

$$r^* \leq 100B\mathbb{E}\mathcal{R}_n\{*\mathcal{F}, P_n f^2 \leq 2r^*\} + \frac{b + 11b^2 \log n}{n}. \tag{52}$$

Next we leverage Dudley's chaining bound (Dudley, 2014) to upper bound $\mathbb{E}\mathcal{R}_n\{*\mathcal{F}, P_n f^2 \leq 2r^*\}$ using integral of covering number. We first bound the covering number of a star hull of $\mathcal{F}$. It follows from (Bartlett et al., 2005) Corollary 3.7 that

$$\log \mathcal{N}_2(\varepsilon, \mathcal{F}, \boldsymbol{x}_{1:n}) \leq \log \left\{ \mathcal{N}_2\left(\frac{\varepsilon}{2}, \mathcal{F}, \boldsymbol{x}_{1:n}\right) \left(\lceil \frac{2}{\varepsilon} \rceil + 1\right) \right\}$$

And covering number $\log \mathcal{N}_2(\varepsilon, \mathcal{F}, n)$ can be bounded using VC-dimension of $\mathcal{F}$ using Haussler's bound on the covering number (Haussler, 1995; Wellner et al., 2013):

$$\log \mathcal{N}_2\left(\frac{\varepsilon}{2}, \mathcal{F}, n\right) \leq c_1 d_{VC} \log\left(\frac{1}{\varepsilon}\right)$$

where $c_1$ is some universal constant. Now we are ready to apply the chaining bound, it follows from Theorem B.7 (Bartlett et al., 2005) that

$$\begin{aligned}
&\mathbb{E}[\mathcal{R}_n(*\mathcal{F}, P_n f^2 \leq 2r^*)] \\
&\leq \frac{c_2}{\sqrt{n}}\mathbb{E}\int_0^{\sqrt{2r^*}} \sqrt{\log \mathcal{N}_2(\varepsilon, *\mathcal{F}, \boldsymbol{x}_{1:n})} d\varepsilon \\
&\leq \frac{c_2}{\sqrt{n}}\mathbb{E}\int_0^{\sqrt{2r^*}} \sqrt{\log \mathcal{N}_2\left(\frac{\varepsilon}{2}, \mathcal{F}, \boldsymbol{x}_{1:n}\right)\left(\lceil \frac{2}{\varepsilon} \rceil + 1\right)} d\varepsilon \\
&\leq c_3 \sqrt{\frac{d_{VC}(\mathcal{F})r^* \log(1/r^*)}{n}} \\
&\leq c_3 \sqrt{\frac{d_{VC}^2(\mathcal{F})}{n^2} + \frac{d_{VC}(\mathcal{F})r^* \log(n/ed_{VC}(\mathcal{F}))}{n}}
\end{aligned} \tag{53}$$

Where $c_2$ and $c_3$ are some universal constants. Together with Equation 52 one can solve for $r^* \lesssim \frac{B^2 d_{VC}(\mathcal{F})\log(\frac{n}{d_{VC}(\mathcal{F})})}{n}$ $\qquad \square$

**Lemma 8.** *Let $S_n = \{(\boldsymbol{x}_i, y_i)\}$ be i.i.d sample from Data Generative Process described in Definition 1. For every $\varepsilon > 0$, there exist a $\delta > 0$ such that if $n \gtrsim \frac{d_{VC}(\mathcal{F})\log(\frac{1}{\varepsilon}) + \log(\frac{1}{\delta})}{\varepsilon}$, following inequality holds simultaneously for all $f \in \mathcal{F}$ with $d_{VC}(\mathcal{F}) < \infty$, with probability at least $1 - \delta$*

$$\frac{1}{n}\sum_{i=1}^n \mathbb{1}\{f(\boldsymbol{x}_i) \neq f^*(\boldsymbol{x}_i)\} \lesssim \mathbb{E}_{\boldsymbol{x}}\mathbb{1}\{f(\boldsymbol{x}) \neq f^*(\boldsymbol{x})\} + \varepsilon \tag{54}$$

*Proof.* The proof invokes Lemma 6, in particular, the Equation 50. Let $\mathbb{1} \circ \mathcal{F} : \mathbb{1}\{f(\boldsymbol{x}) \neq f^*(\boldsymbol{x}), f \in \mathcal{F}\}$ be the hypothesis class $\mathcal{H}$ in Lemma 6. Since $f^*$ is a deterministic boolean function, it does not increase the number of points that can be shattered by $\mathcal{F}$. We have $d_{VC}(\mathbb{1} \circ \mathcal{F}) \leq d_{VC}(\mathcal{F})$. In particular, we choose the functional $T(\cdot) = \mathbb{E}[\cdot]$ and it is easy to verify that

$$\boldsymbol{Var}(\mathbb{1}\{f(\boldsymbol{x}) \neq f^*(\boldsymbol{x})\}) \leq \mathbb{E}_{\boldsymbol{x}}[\mathbb{1}\{f(\boldsymbol{x}) \neq f^*(\boldsymbol{x})\}] = \mathbb{E}_{\boldsymbol{x}}[\mathbb{1}^2\{f(\boldsymbol{x}) \neq f^*(\boldsymbol{x})\}].$$

Let $\psi(r) = 100\mathbb{E}\mathcal{R}_n\{*\mathcal{F}, \mathbb{E}f \leq r\} + \frac{11 \log n}{n}$. We have

$$\mathbb{E}\mathcal{R}_n\{\mathcal{F}, \mathbb{E}f^2 \leq r\} \leq \mathbb{E}\mathcal{R}_n\{*\mathcal{F}, \mathbb{E}f^2 \leq r\} \leq 100\mathbb{E}\mathcal{R}_n\{*\mathcal{F}, \mathbb{E}f^2 \leq r\} + \frac{11 \log n}{n} = \psi(r)$$

Since local Rademacher averages of the star-hull is sub-root function, we know for all $r \geq r^*$, $\psi(r) \geq \psi(r^*) = r^*$. By Equation 50 in Lemma 6 we have

$$\frac{1}{n}\sum_{i=1}^n \mathbb{1}\{f(\boldsymbol{x}_i) \neq f^*(\boldsymbol{x}_i)\} \leq 2\mathbb{E}_{\boldsymbol{x}}\mathbb{1}\{f(\boldsymbol{x}) \neq f^*(\boldsymbol{x})\} + 15r^* + \frac{\log(1/\delta) + 5200}{n}\varepsilon \tag{55}$$

Next we bound $r^*$. A direct application of Lemma 7 show that

$$r^* \lesssim \frac{d_{VC}(\mathbb{1} \circ \mathcal{F})\log(\frac{n}{d_{VC}(\mathbb{1}\circ\mathcal{F})})}{n} \lesssim \frac{d_{VC}(\mathcal{F})\log(\frac{n}{d_{VC}(\mathcal{F})})}{n}.$$

The rest of the proof follows from plugging $r^*$ in Equation 50 and removing absolute constants. $\quad \square$

A.8   ILLUSTRATIVE EXAMPLE FOR ALGORITHM 1

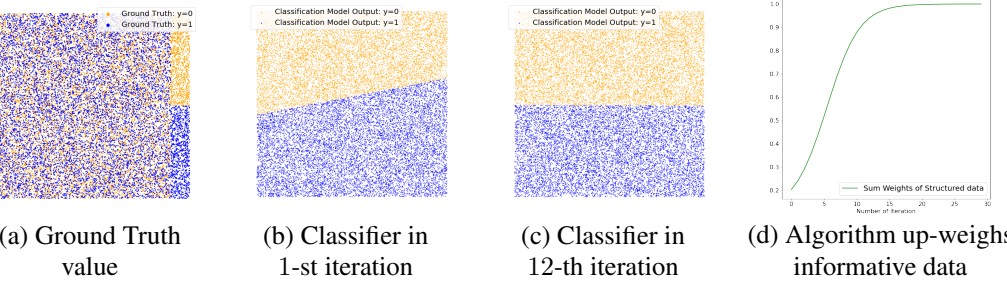

(a) Ground Truth
value

(b) Classifier in
1-st iteration

(c) Classifier in
12-th iteration

(d) Algorithm up-weighs
informative data

Figure 3: Illustration of Algorithm 1 when $\lambda = \frac{1}{2}$. a) shows a mix of informative/uninformative data. b) and c) show classifiers trained on weighted samples with different number of iterations. By up-weighing the informative datapoints, the algorithm progressively improves the classifier. d) shows the sum of weights of informative over total selected, i.e $\frac{\sum_{i:\boldsymbol{x}_i \in \Omega_I} \gamma_i}{\sum_i^n \gamma_i}$ ( See $\gamma$ in Algorithm 1): the algorithm converges to an all-informative dataset.

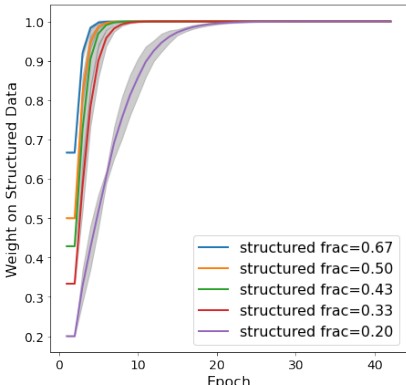

Figure 4: We show case the effecacy of the MWU mechanism on MNIST dataset. We plot the weight of informative data as a function of training epoch and the ratio of informative data. The $y$ axis is the percentage of weight put on the iformative data, i.e $\frac{\sum_{i:\boldsymbol{x}_i \in \Omega_I} \gamma_i}{\sum_i^n \gamma_i}$ in the notation of Algorithm 1.

## B   MORE EXPERIMENT RESULTS AND DETAILS

### B.1   EXPERIMENT SETTING AND IMPLEMENTATION DETAILS

**Extension to multi-class.** Our method extends to the multi-class setting naturally. In the case of $K$-class classification, our selector loss remains the same while the predictor becomes $f(\boldsymbol{x}) = f(\boldsymbol{x})_{1:K} : \mathcal{X} \to \Delta^K$ where $\Delta^K$ is the $K$-simplex. Meanwhile, we use multi-class cross entropy loss to train the classifier. The pseudo-informative label becomes $\widehat{z}_i = \mathbb{1}\{\arg\max_{k \in [K]} f(\boldsymbol{x}_i)_k = y_i\}$.

**Semi-Synthetic Experiment Setting.** For experiments in Section 7.1 and 7.2We use same backbone TinyCNN models for all baselines. It is a light-weight CNN with 2 convolutional layers and 3 fully connected layers . We adopt same training scheme for all baseline algorithm. We use Adam optimizer with learning rate 1e-3 and weight weight decay rate 1e-4. We train 220 epochs using batch size 196 for all baselines and the leanring rate is reduced by 50% at 45th epoch and 90th epoch.

For experiments in Section 7.2, we use the default hyper-parameters for every method as recommended in the respective original paper (i.e., internal selective learning-specific defaults, as reported in (Geifman & El-Yaniv, 2019; Liu et al., 2019), and $\beta = 2$ and MWU step-size $\eta = 2$ for our algorithm ). It simulates a practical scenario where hyper-parameter optimization is impossible, since the

ground truth regarding which datapoints are actually informative is never revealed. For SelectiveNet, the default hyper-parameters are the weight for coverage rate penalty $\lambda_{sl} = 32$ and the weight for selective net loss $a = 0.5$. For DeepGambler, the sole hyper-parameter "reward" should lie between 1 and 10, and we choose the default value recommended by the author which is 2.2.

For the SVHN experiment, we use ResNet18 (He et al., 2016b) as the backbone model for every candidate. We set the batch size to 128, and use Adam as the optimizer with learning rate 1e-3 and weight decay 1e-4. We train each algorithm for 162 epochs, and shrink our learning rate at both epoch 45 and 90 by half each time.

We assume that the ratio of informative data, $\alpha$, is unknown to all methods. This is necessary in practice; the ratio and strength of noise are not known in most real world scenarios. For SelectiveNet and DeepGambler, such ratio is a required input. To run these baselines, we first run the original backbone for 60 epochs, and then estimate $\alpha$ using backbone's training accuracy. Assume that the backbone fits all of the informative data perfectly and also makes some correct guesses on noisy data with probability $\frac{1}{\text{num of classes}}$, then the frequency estimation of $\alpha$ is $\hat{\alpha} = \frac{\text{num of classes}*\text{train acc}-1}{\text{num of classes}-1}$ , which is the estimation of $\alpha$ we give to baselines. Note such estimation can only be applied when $\lambda = \frac{1}{2}$.

**Real-World Experiment Setting.** We use same backbone model for all baseline methods. Specifically, we use a 1-layer LSTM for volatility data. We use VGG16 Simonyan & Zisserman (2014) for BUS data. We use a 3-layer multi-layer perceptron for LC data. We adopt same training scheme for all baseline algorithm. We use Adam optimizer with learning rate 1e-4 and weight decay rate 1e-4 for all experiments. The learning rate is reduced by 50% every 10 epochs. We use batchsize 128 for volatility data, batch size 32 for BUS data and batch size 256 for LC data, which is determined according to the size of the data. We further split the training set and perform 30 random hyper-parameter searching for each baseline. For BUS and LC, we set the training epoch to be 50. For financial time-series volatility data, since each baseline is quite sensitive to the running epoch, we add the training epoch as a hyper-parameter searching dimension during the HPO process.

## B.2    Real-World Dataset Description

The first dataset is the Oxford realized volatility (Volatility) (Heber et al., 2009) data set containing 5-min realized volatility of 31 stock indices from 2000 to 2022 which contains 155107 records in total. We use past volatility and returns as features, and the task here is to predict whether the next day volatility will be higher than current one, making it a binary classification. We choose data from 2000 to 2020 as our training set and the rest for the testing (2020 Jan. to 2021 Oct.). This data set is used as an example to show our algorithm's possible application in selectively forecasting financial time series.

The second one is the dataset of breast ultrasound images (BUS) (Al-Dhabyani et al., 2020). BUS contains 780 gray-scale breast ultrasound images among women in ages between 25 and 75 years old. These images have average size $500 \times 500$ pixels and can be categorized into 3 classes (487 benign, 210 malign and 133 healthy). We randomly choose 80% of data as training dataset and the rest 20% for testing. We are going to use this dataset as an example to show a possible application of our algorithm in automatic diagnosis. The machine can generate diagnosis result only on selected cases and deliver unsure cases to human expert for further investigation.

[1]The third one is the lending club dataset (LC). Lending club is a peer-to-peer lending company that matches borrowers with investors through an online platform. The lending club dataset (LC) contains loan data of its customers from 2007 to 2018. We compare different version of existing dataset of LC and remove all inconsistent and incomplete records. There are different status of loans record in this dataset, we keep 3 types of these record that consist the major part of the dataset (261442 charged off cases, 1035418 fully paid cases and 25757 late cases). We use 20% of the dataset as the testing set. This example shows our algorithm can be use to grant loan given on different risk preference.

Table 3 presents the original accuracy given by neural network on each of these 3 real-world data set. For all dataset, the neural network without using selection mechanism cannot give reliable inference. In mortgage granting, high risk like this can cause significant loss. In medical diagnosis which is healthy issue critical, a diagnosis with miss-diagnose rate as high as 15% is not acceptable. However, if we apply our selective algorithm, we can see that the risk on all dataset sharply reduced. In BUS

---

[1]url:https://www.kaggle.com/datasets/wordsforthewise/lending-club/download?datasetVersionNumber=3

dataset, we can even almost perfectly guarantee the diagnosis result empirically for our most confident cases. These evidence are of practical interest.

Table 3: DNN Original Risk on Each Dataset

|  | Volatility | BUS | LC |
|---|---|---|---|
| Risk | 0.340±0.002 | 0.152±0.008 | 0.392±0.001 |

## C  ABLATION STUDY

### C.1  ABLATION STUDY: EFFECT OF INFORMATIVE RATIO $\alpha$.

We provide several ablation studies that further elucidate the reasons of our advantages. We stick with case that $\lambda = \frac{1}{2}$ so that baselines are able to estimate the value of $\alpha$ according to the 'no label noise' assumption for informative data. Firstly, we try to reduce the total data size by sampling the original dataset. We test each baseline with these partial dataset to evaluate their performance under data shortage scenario. The results are presented in Appendix Table 4 and Table 5. In this study, we follow the setting of our synthetic experiment and fully shuffle the labels of the uninformative data and keep all the informative data intact. We can see that in this setting, our method still outperform all baselines.

Then we conduct experiments where we reveal the ground truth $\alpha$ to each baseline. We test all baselines on both complete dataset and partial dataset. We always completely permute labels of MNIST to impose high label noise in uninformative data and keep informative data clean. The results are presented in Appendix Table 6 - 9. We can see that even in the "easier" scenario, where every algorithm is handed the true $\alpha$, our method still wins out. We believe that this is partially due to the MWU mechanism. We will have a section discussing MWU later.

Finally, we provide a ablation study investigation the effect of the choice of epoch to estimate the $\alpha$. For all baselines, the estimation of $\alpha$ is a non-trivial step, yet it is crucial to the results. We hypothesize that this is the main cause of the suboptimal performance of these baselines. Here we provide additional results by training for 120 epochs instead of 60 in section 7.2. The estimation error for $\alpha$ at 120 epochs is shown in Table 10. One can see that the error's magnitude is significant. This results in worse performance of both selector and classifier (Tables 11 and 12). Our method's result stay the same as it in Table 1, since it doesn't require $\alpha$ as an input.

Table 4: (MNIST - Partial Dataset - Unknown $\alpha$) Uninformative/Informative MNIST/Fashion using 25% shuffled data as proxy for noise.

| Uninformative Data Num. | Informative Data Num. | Criterion | Confidence | SLNet | DeepGambler | Ours |
|---|---|---|---|---|---|---|
| 15000 | 3750 | SR(%) | 79.87 ± 0.40 | 74.03 ± 1.38 | 73.22±0.51 | **20.24 ± 9.25** |
|  |  | Precision | 0.22 ± 0.00 | 0.29 ± 0.02 | 0.30 ± 0.01 | 1.00 ± 0.00 |
|  |  | Recall | 0.85 ± 0.00 | 1.00 ± 0.00 | 1.00 ± 0.00 | 0.88 ± 0.07 |
| 15000 | 7500 | SR(%) | 65.83 ± 0.22 | 57.76 ± 1.97 | 58.13±0.68 | **13.71±0.24** |
|  |  | Precision | 0.38 ± 0.00 | 0.48 ± 0.02 | 0.48 ± 0.01 | 1.00 ± 0.00 |
|  |  | Recall | 0.92 ± 0.01 | 1.00 ± 0.00 | 1.00 ± 0.00 | 0.83 ± 0.01 |
| 15000 | 11250 | SR(%) | 55.99 ± 0.49 | 47.39 ± 1.85 | 46.96±0.50 | **13.06±1.86** |
|  |  | Precision | 0.49 ± 0.01 | 0.60 ± 0.02 | 0.60 ± 0.01 | 0.99 ± 0.00 |
|  |  | Recall | 0.94 ± 0.01 | 1.00 ± 0.00 | 1.00 ± 0.00 | 0.84 ± 0.02 |
| 15000 | 15000 | SR(%) | 48.51 ± 0.25 | 39.23 ± 0.90 | 39.85±0.36 | **19.10±4.50** |
|  |  | Precision | 0.57 ± 0.00 | 0.68 ± 0.00 | 0.68 ± 0.00 | 0.99 ± 0.01 |
|  |  | Recall | 0.95 ± 0.00 | 1.00 ± 0.00 | 1.00 ± 0.00 | 0.91 ± 0.04 |

### C.2  ABLATION STUDY: EFFECT OF MULTIPLICATIVE WEIGHT UPDATE

As mentioned in the discussion of $\alpha$'s effect, we can see that our method win in the "easier" scenario, where every algorithm is handed the true $\alpha$, our method still wins out. We believe that this is partially due to the MWU mechanism. It guides the classifier to put more emphasis on the informative data. This is confirmed via an ablation study reported in Table 13: when MWU is turned off, our algorithm's performance deteriorates.

Table 5: (SVHN - Partial Dataset - Unknown $\alpha$) Uninformative/Informative SVHN using 25% of shuffled classes as proxy for noise.

| Uninformative Data Num. | Informative Data Num. | Criterion | Confidence | SelectiveNet | DeepGambler | Ours |
|---|---|---|---|---|---|---|
| 9200 | 2285 | SR(%) | $74.47 \pm 4.15$ | $64.68 \pm 20.32$ | $34.48 \pm 13.46$ | $\mathbf{14.36 \pm 0.08}$ |
| | | Precision | $0.41 \pm 0.05$ | $0.37 \pm 0.22$ | $0.68 \pm 0.13$ | $0.87 \pm 0.07$ |
| | | Recall | $0.94 \pm 0.01$ | $0.85 \pm 0.23$ | $0.83 \pm 0.24$ | $0.80 \pm 0.02$ |
| 9200 | 4600 | SR(%) | $61.44 \pm 0.64$ | $48.42 \pm 23.42$ | $26.42 \pm 1.68$ | $\mathbf{7.46 \pm 0.87}$ |
| | | Precision | $0.57 \pm 0.00$ | $0.58 \pm 0.17$ | $0.77 \pm 0.02$ | $0.94 \pm 0.07$ |
| | | Recall | $0.95 \pm 0.01$ | $0.97 \pm 0.03$ | $0.96 \pm 0.01$ | $0.85 \pm 0.01$ |
| 9200 | 6900 | SR(%) | $52.41 \pm 0.71$ | $25.64 \pm 3.01$ | $20.83 \pm 2.06$ | $\mathbf{7.25 \pm 0.77}$ |
| | | Precision | $0.65 \pm 0.00$ | $0.76 \pm 0.03$ | $0.82 \pm 0.02$ | $0.94 \pm 0.01$ |
| | | Recall | $0.96 \pm 0.00$ | $0.96 \pm 0.03$ | $0.97 \pm 0.01$ | $0.87 \pm 0.01$ |
| 9200 | 9200 | SR(%) | $49.45 \pm 0.45$ | $24.69 \pm 3.08$ | $18.81 \pm 1.47$ | $\mathbf{7.22 \pm 0.54}$ |
| | | Precision | $0.67 \pm 0.00$ | $0.77 \pm 0.03$ | $0.83 \pm 0.01$ | $0.94 \pm 0.01$ |
| | | Recall | $0.96 \pm 0.00$ | $0.98 \pm 0.00$ | $0.98 \pm 0.00$ | $0.88 \pm 0.01$ |

Table 6: (MNIST - Full Data Setting - Known $\alpha$) Results on a synthetic dataset consisting of uninformative MNIST data and informative Fashion-MNIST data using the entire MNIST.

| Uninformative Data Num. | Informative Data Num. | Criterion | Confidence | SelectiveNet | DeepGambler | Ours |
|---|---|---|---|---|---|---|
| 60000 | 15000 | SR(%) | $10.00 \pm 0.32$ | $10.11 \pm 0.42$ | $9.77 \pm 0.51$ | $\mathbf{9.18 \pm 0.49}$ |
| | | Precision | $0.99 \pm 0.00$ | $1.00 \pm 0.00$ | $1.00 \pm 0.00$ | $1.00 \pm 0.00$ |
| | | Recall | $0.99 \pm 0.00$ | $1.00 \pm 0.00$ | $1.00 \pm 0.00$ | $1.00 \pm 0.00$ |
| 60000 | 30000 | SR(%) | $9.41 \pm 0.20$ | $9.91 \pm 0.48$ | $\mathbf{9.39 \pm 0.26}$ | $\mathbf{9.03 \pm 0.73}$ |
| | | Precision | $0.99 \pm 0.00$ | $1.00 \pm 0.00$ | $1.00 \pm 0.00$ | $1.00 \pm 0.00$ |
| | | Recall | $0.99 \pm 0.00$ | $1.00 \pm 0.00$ | $1.00 \pm 0.00$ | $1.00 \pm 0.00$ |
| 60000 | 45000 | SR(%) | $8.63 \pm 0.23$ | $9.39 \pm 0.35$ | $9.13 \pm 0.51$ | $\mathbf{8.58 \pm 0.26}$ |
| | | Precision | $1.00 \pm 0.00$ | $1.00 \pm 0.00$ | $1.00 \pm 0.00$ | $1.00 \pm 0.00$ |
| | | Recall | $0.99 \pm 0.00$ | $1.00 \pm 0.00$ | $1.00 \pm 0.00$ | $1.00 \pm 0.00$ |
| 60000 | 60000 | SR(%) | $8.11 \pm 0.08$ | $8.21 \pm 0.12$ | $8.16 \pm 0.05$ | $8.04 \pm 0.49$ |
| | | Precision | $1.00 \pm 0.00$ | $1.00 \pm 0.00$ | $1.00 \pm 0.00$ | $1.00 \pm 0.00$ |
| | | Recall | $0.99 \pm 0.00$ | $1.00 \pm 0.00$ | $1.00 \pm 0.00$ | $1.00 \pm 0.00$ |

Table 7: (MNIST - Partial Data Setting - Known $\alpha$) Results on a synthetic dataset consisting of uninformative MNIST data and informative Fashion-MNIST data using 25% of MNIST.

| Uninformative Data Num. | Informative Data Num. | Criterion | Confidence | SelectiveNet | DeepGambler | Ours |
|---|---|---|---|---|---|---|
| 15000 | 3750 | SR(%) | $16.71 \pm 0.31$ | $13.50 \pm 0.30$ | $14.11 \pm 5.65$ | $\mathbf{11.38 \pm 0.49}$ |
| | | Precision | $0.94 \pm 0.02$ | $1.00 \pm 0.00$ | $1.00 \pm 0.00$ | $1.00 \pm 0.00$ |
| | | Recall | $0.90 \pm 0.04$ | $1.00 \pm 0.00$ | $1.00 \pm 0.00$ | $1.00 \pm 0.00$ |
| 15000 | 7500 | SR(%) | $13.93 \pm 0.40$ | $13.34 \pm 1.23$ | $\mathbf{11.20 \pm 0.22}$ | $11.29 \pm 0.44$ |
| | | Precision | $0.96 \pm 0.01$ | $0.99 \pm 0.01$ | $1.00 \pm 0.00$ | $1.00 \pm 0.00$ |
| | | Recall | $0.95 \pm 0.02$ | $0.93 \pm 0.16$ | $1.00 \pm 0.00$ | $1.00 \pm 0.00$ |
| 15000 | 11250 | SR(%) | $12.29 \pm 0.31$ | $11.45 \pm 0.37$ | $\mathbf{10.61 \pm 0.12}$ | $\mathbf{10.42 \pm 0.48}$ |
| | | Precision | $0.97 \pm 0.01$ | $1.00 \pm 0.00$ | $1.00 \pm 0.00$ | $0.99 \pm 0.00$ |
| | | Recall | $0.95 \pm 0.03$ | $1.00 \pm 0.00$ | $1.00 \pm 0.00$ | $1.00 \pm 0.00$ |
| 15000 | 15000 | SR(%) | $11.76 \pm 0.20$ | $11.24 \pm 0.35$ | $10.12 \pm 0.28$ | $\mathbf{9.97 \pm 0.33}$ |
| | | Precision | $0.97 \pm 0.01$ | $1.00 \pm 0.00$ | $1.00 \pm 0.00$ | $0.99 \pm 0.01$ |
| | | Recall | $0.95 \pm 0.02$ | $1.00 \pm 0.00$ | $1.00 \pm 0.00$ | $1.00 \pm 0.00$ |

Table 8: (SVHN - Full Data Setting - Known $\alpha$) Results on a synthetic dataset consisting of uninformative SVHN and informative SVHN using the entire uninformative data.

| Uninformative Data Num. | Informative Data Num. | Criterion | Confidence | SelectiveNet | DeepGambler | Ours |
|---|---|---|---|---|---|---|
| 33800 | 9200 | SR(%) | $6.45 \pm 0.93$ | $58.06 \pm 35.91$ | $4.58 \pm 0.61$ | $\mathbf{4.48 \pm 0.43}$ |
| | | Precision | $0.93 \pm 0.01$ | $0.41 \pm 0.36$ | $0.96 \pm 0.00$ | $0.96 \pm 0.00$ |
| | | Recall | $0.89 \pm 0.01$ | $0.37 \pm 0.34$ | $0.86 \pm 0.02$ | $0.86 \pm 0.01$ |
| 33800 | 18300 | SR(%) | $4.30 \pm 0.31$ | $80.42 \pm 1.55$ | $3.08 \pm 0.15$ | $\mathbf{2.91 \pm 0.21}$ |
| | | Precision | $0.96 \pm 0.00$ | $0.35 \pm 0.00$ | $0.97 \pm 0.00$ | $0.97 \pm 0.00$ |
| | | Recall | $0.91 \pm 0.01$ | $0.38 \pm 0.05$ | $0.90 \pm 0.01$ | $0.88 \pm 0.01$ |
| 33800 | 26200 | SR(%) | $4.49 \pm 0.68$ | $5.22 \pm 0.90$ | $3.86 \pm 0.11$ | $\mathbf{3.65 \pm 0.17}$ |
| | | Precision | $0.96 \pm 0.68$ | $0.95 \pm 0.01$ | $0.97 \pm 0.00$ | $0.96 \pm 0.00$ |
| | | Recall | $0.94 \pm 0.01$ | $0.92 \pm 0.03$ | $0.94 \pm 0.00$ | $0.93 \pm 0.01$ |
| 33800 | 28400 | SR(%) | $10.87 \pm 0.68$ | $8.27 \pm 5.03$ | $7.65 \pm 0.47$ | $\mathbf{6.39 \pm 0.76}$ |
| | | Precision | $0.89 \pm 0.00$ | $0.92 \pm 0.05$ | $0.92 \pm 0.01$ | $0.93 \pm 0.01$ |
| | | Recall | $0.98 \pm 0.00$ | $0.95 \pm 0.04$ | $0.97 \pm 0.01$ | $0.95 \pm 0.01$ |

## C.3 ABLATION STUDY: EFFECT OF NOISE RATIO GAP $\lambda$.

We inject different level of noise into each part of the data according to Definition 1 by setting $\lambda(\boldsymbol{x}) = \lambda$. The higher the $\lambda$ is, the larger the gap of the information noise ratio between informative and uninformative partition. Specifically, for informative data, we inject $100 * (\frac{1}{2} - \lambda)\%$ uniform

Table 9: (SVHN - Partial Data Setting - Known $\alpha$) Results on a synthetic dataset consisting of uninformative SVHN and informative SVHN using 25% uninformative data.

| Uninformative Data Num. | Informative Data Num. | Criterion | Confidence | SelectiveNet | DeepGambler | Ours |
|---|---|---|---|---|---|---|
| 9200 | 2285 | SR(%) | $30.13 \pm 34.94$ | $55.91 \pm 29.49$ | $\mathbf{7.65 \pm 0.48}$ | $11.74 \pm 2.32$ |
| | | Precision | $0.70 \pm 0.35$ | $0.19 \pm 0.10$ | $0.93 \pm 0.01$ | $0.90 \pm 0.02$ |
| | | Recall | $0.67 \pm 0.34$ | $0.16 \pm 0.13$ | $0.78 \pm 0.02$ | $0.80 \pm 0.06$ |
| 9200 | 4600 | SR(%) | $10.41 \pm 0.26$ | $36.46 \pm 28.65$ | $9.70 \pm 1.05$ | $\mathbf{8.12 \pm 0.87}$ |
| | | Precision | $0.91 \pm 0.00$ | $0.68 \pm 0.24$ | $0.90 \pm 0.03$ | $0.93 \pm 0.01$ |
| | | Recall | $0.87 \pm 0.01$ | $0.69 \pm 0.24$ | $0.79 \pm 0.04$ | $0.78 \pm 0.08$ |
| 9200 | 6900 | SR(%) | $8.51 \pm 0.85$ | $34.97 \pm 30.15$ | $\mathbf{7.49 \pm 0.47}$ | $7.67 \pm 0.63$ |
| | | Precision | $0.92 \pm 0.02$ | $0.73 \pm 0.21$ | $0.94 \pm 0.01$ | $0.92 \pm 0.02$ |
| | | Recall | $0.91 \pm 0.02$ | $0.74 \pm 0.22$ | $0.83 \pm 0.04$ | $0.86 \pm 0.01$ |
| 9200 | 9200 | SR(%) | $9.06 \pm 0.82$ | $13.66 \pm 1.39$ | $7.88 \pm 0.29$ | $\mathbf{7.57 \pm 0.51}$ |
| | | Precision | $0.91 \pm 0.01$ | $0.88 \pm 0.02$ | $0.93 \pm 0.01$ | $0.92 \pm 0.01$ |
| | | Recall | $0.94 \pm 0.01$ | $0.92 \pm 0.01$ | $0.88 \pm 0.03$ | $0.91 \pm 0.01$ |

Table 10: Real $\alpha$ and Estimation Error (120 epochs)

| Uninformative Data Num. | Informative Data Num. | $\hat{\alpha} - \alpha$ | $\alpha$ |
|---|---|---|---|
| 60000 | 60000 | $0.05 \pm 0.00$ | 0.50 |
| 60000 | 45000 | $0.08 \pm 0.00$ | 0.33 |
| 60000 | 30000 | $0.11 \pm 0.00$ | 0.25 |
| 60000 | 15000 | $0.17 \pm 0.01$ | 0.20 |
| 15000 | 15000 | $0.33 \pm 0.01$ | 0.50 |
| 15000 | 11250 | $0.40 \pm 0.01$ | 0.33 |
| 15000 | 7500 | $0.49 \pm 0.01$ | 0.25 |
| 15000 | 3750 | $0.63 \pm 0.01$ | 0.20 |

Table 11: (MNIST - Full Dataset - Unknown $\alpha$ - Run 120 Epochs for $\hat{\alpha}$). Results on a synthetic dataset consisting of uninformative MNIST and informative Fashion-MNIST using the entirety of shuffled MNIST as proxy for noise.

| Uninformative Data Num. | Informative Data Num. | Criterion | Confidence | SelectiveNet | DeepGambler | Ours |
|---|---|---|---|---|---|---|
| 60000 | 15000 | SR(%) | $51.81 \pm 0.47$ | $53.0 \pm 1.00$ | $53.0 \pm 1.00$ | $\mathbf{10.35 \pm 0.31}$ |
| | | Precision | $0.54 \pm 0.00$ | $0.52 \pm 0.01$ | $0.53 \pm 0.01$ | $\mathbf{1.00 \pm 0.00}$ |
| | | Recall | $0.94 \pm 0.00$ | $1.00 \pm 0.00$ | $1.00 \pm 0.00$ | $0.85 \pm 0.01$ |
| 60000 | 30000 | SR(%) | $31.86 \pm 0.73$ | $35.00 \pm 1.00$ | $34.0 \pm 1.00$ | $\mathbf{12.03 \pm 1.06}$ |
| | | Precision | $0.76 \pm 0.01$ | $0.72 \pm 0.01$ | $0.74 \pm 0.01$ | $1.00 \pm 0.00$ |
| | | Recall | $0.98 \pm 0.00$ | $1.00 \pm 0.00$ | $1.00 \pm 0.00$ | $0.92 \pm 0.04$ |
| 60000 | 45000 | SR(%) | $22.29 \pm 0.59$ | $26.0 \pm 0.00$ | $24.00 \pm 0.00$ | $\mathbf{12.58 \pm 2.00}$ |
| | | Precision | $0.79 \pm 0.29$ | $0.88 \pm 0.86$ | $0.89 \pm 0.00$ | $1.00 \pm 0.00$ |
| | | Recall | $0.99 \pm 0.00$ | $1.00 \pm 0.00$ | $1.00 \pm 0.00$ | $0.97 \pm 0.03$ |
| 60000 | 60000 | SR(%) | $16.29 \pm 0.42$ | $\mathbf{11.00 \pm 2.00}$ | $18.00 \pm 0.00$ | $\mathbf{11.61 \pm 0.79}$ |
| | | Precision | $0.92 \pm 0.00$ | $0.98 \pm 0.02$ | $0.90 \pm 0.00$ | $1.00 \pm 0.00$ |
| | | Recall | $1.00 \pm 0.00$ | $0.98 \pm 0.02$ | $1.00 \pm 0.00$ | $0.97 \pm 0.01$ |

Table 12: (MNIST - Partial Data Setting - Unknown $\alpha$-Run 120 epochs for $\hat{\alpha}$) Results on a synthetic dataset consisting of uninformative MNIST data and informative Fashion-MNIST data using 25% of MNIST.

| Uninformative Data Num. | Informative Data Num. | Criterion | Confidence | SelectiveNet | DeepGambler | Ours |
|---|---|---|---|---|---|---|
| 15000 | 3750 | SR(%) | $72.06 \pm 0.68$ | $78.00 \pm 1.00$ | $78.00 \pm 0.00$ | $\mathbf{11.38 \pm 0.49}$ |
| | | Precision | $0.29 \pm 0.01$ | $0.24 \pm 0.01$ | $0.24 \pm 0.00$ | $1.00 \pm 0.00$ |
| | | Recall | $0.70 \pm 0.01$ | $1.00 \pm 0.01$ | $1.00 \pm 0.00$ | $1.00 \pm 0.00$ |
| 15000 | 7500 | SR(%) | $54.60 \pm 0.44$ | $65.00 \pm 1.00$ | $64.00 \pm 0.00$ | $\mathbf{11.29 \pm 0.44}$ |
| | | Precision | $0.49 \pm 0.00$ | $0.40 \pm 0.01$ | $0.41 \pm 0.00$ | $1.00 \pm 0.00$ |
| | | Recall | $0.80 \pm 0.01$ | $1.00 \pm 0.00$ | $1.00 \pm 0.00$ | $1.00 \pm 0.00$ |
| 15000 | 11250 | SR(%) | $44.17 \pm 0.72$ | $55.00 \pm 1.00$ | $54.00 \pm 0.00$ | $\mathbf{10.42 \pm 0.48}$ |
| | | Precision | $0.60 \pm 0.01$ | $0.50 \pm 0.01$ | $0.52 \pm 0.00$ | $0.99 \pm 0.00$ |
| | | Recall | $0.85 \pm 0.01$ | $1.00 \pm 0.00$ | $1.00 \pm 0.00$ | $1.00 \pm 0.00$ |
| 15000 | 15000 | SR(%) | $37.08 \pm 0.26$ | $48.00 \pm 1.00$ | $46.00 \pm 0.00$ | $\mathbf{9.97 \pm 0.33}$ |
| | | Precision | $0.68 \pm 0.00$ | $0.58 \pm 0.01$ | $0.61 \pm 0.00$ | $0.99 \pm 0.01$ |
| | | Recall | $0.87 \pm 0.00$ | $1.00 \pm 0.00$ | $1.00 \pm 0.00$ | $1.00 \pm 0.00$ |

label noise (each class has chance $(\frac{1}{2} - \lambda)$ to be flipped into the other classes). For uninformative data, we inject $100 * \frac{1}{2} * (\frac{1}{2} + \lambda)\%$ uniform label noise. We test each baseline on these semi-synthesized dataset. The result is presented in Table 14.

For all baselines, we give $\lambda$ as prior information to calculate $\alpha$ according to the realized accuracy of predictor $\hat{f}$. In Table 14, we can see that our method can effectively recover informative data out of the uninformative ones compared with existing baselines. We put a † on top of selective risk number

Table 13: (MNIST - Full Data Setting - Known $\alpha$ - Turn off MWU) Results on a synthetic dataset consisting of uninformative MNIST data and informative Fashion-MNIST data using the entire MNIST.

| Uninformative Data Num. | Informative Data Num. | SR | Recall | Precision |
|---|---|---|---|---|
| 60000 | 60000 | 0.09±0.00 | 0.99±0.01 | 0.99±0.01 |
| 60000 | 45000 | 0.09±0.00 | 1.00±0.00 | 1.00±0.00 |
| 60000 | 30000 | 0.10±0.00 | 1.00±0.00 | 1.00±0.00 |
| 60000 | 15000 | 0.10±0.00 | 1.00±0.00 | 1.00±0.00 |
| 15000 | 15000 | 0.11±0.01 | 1.00±0.00 | 1.00±0.00 |
| 15000 | 11250 | 0.11±0.00 | 1.00±0.00 | 1.00±0.00 |
| 15000 | 7500 | 0.12±0.00 | 1.00±0.00 | 1.00±0.00 |
| 15000 | 3750 | 0.12±0.00 | 1.00±0.00 | 1.00±0.00 |

where the corresponding algorithm fail to select reasonable amount of data (low recall) and thus result in degenerated performance. All baselines have the same problem with learning given noisy labels. The key input $\alpha$ cannot be properly estimated due to the poor accuracy, which in turn leads to poor selection result. Our method doesn't have this issue because it learns to abstain uninformative data and thus doesn't require knowing $\alpha$.)

Table 14: Synthetic Experiment on MNIST+FashionMNIST: Fix $\alpha$ and Vary $\lambda$

| $\lambda$ | Criterion | Confidence | SLNet | DeepGambler | Ours |
|---|---|---|---|---|---|
| 0.30 | SR(%) | $0.00 \pm 0.00^{\dagger}$ | 38.57±9.95 | 66.67±57.73 | **28.27±4.62** |
| | Precision | 0.67±0.58 | 0.96±0.07 | 0.67±0.58 | 0.78±0.26 |
| | Recall | 0.00±0.00 | 0.08±0.07 | 0.00±0.00 | **0.28±0.08** |
| 0.35 | SR(%) | $0.00 \pm 0.00^{\dagger}$ | 44.11±51.03 | $0.00 \pm 0.00^{\dagger}$ | **27.92±2.39** |
| | Precision | 1.00±0.00 | 0.60±0.53 | 1.00±0.00 | 0.92±0.14 |
| | Recall | 0.00±0.00 | 0.11±0.18 | 0.00±0.00 | **0.51±0.03** |
| 0.40 | SR(%) | 33.33±57.73 | 27.59±6.29 | 33.33±57.73 | **23.46±4.13** |
| | Precision | 1.00±0.00 | 1.00±0.00 | 1.00±0.00 | 0.99±0.01 |
| | Recall | 0.00±0.00 | **0.33±0.57** | 0.00±0.00 | **0.55±0.06** |

## C.4 ABLATION STUDY: CHOICE OF HYPER-PARAMETER

In this section, we provide ablation study on sensitivity of different algorithms w.r.t their hyper-parameter. We study under the same setting as Table 1 and 4. For SelectiveNet, we first fix $\lambda_{sl}$ to 32 (default setting that is recommended by the author in the original paper) and then we vary $a$ from 0.1 to 0.7. Then we fix $a$ to be 0.5 (default setting) and then we vary $\lambda_{sl}$ from 1 to 66. For DeepGambler, we vary $o$ from 1 to 7. For our algorithm, we progressively increase the hyper-parameter $\beta$ from 4 to 10. As presented in Fig 7, Fig 5 and Fig 6, the baselines' performance relies heavily on their key hyperparameters. On the other hand, our method is much more robust w.r.t. its hyperparameter, $\beta$.

We can see that the performance of all baselines are quite sensitive to the choice of hyper-parameters and are expected to experience large fluctuations. In contrast, our algorithm is more stable with regard to the choice of hyper-parameters. This empirical observation supports that the choice of $\beta$ is flexible, as it is stated in Theorem 1. Furthermore, in all scenarios, our algorithm's performance is better than these two baselines as following. Firstly, our selector has better precision such that we can recover almost all informative data while the two baselines cannot. These two baselines tend to select the whole data set indistinguishably (low precision and high recall). Secondly, these baselines consistently show deteriorated risk performance compared against ours, mainly because of their selector fails to pick informative data.

We also present the convergence curve of each evaluation metric for the partial data blind setting where $\alpha = 50\%$ in Fig 8. We pick different combination of $\beta$ and MWU step-size $\eta$. We can see that both recall and precision can converge in a very quick and smooth manner. The performance of our algorithm is very robust against different combination of hyper-parameter. There can be some slightly recall drop and precision increase when $\beta$ is chosen to be some extreme values, e.g., $\beta = 10$. We include such case to illustrate that while the method is not sensitive to hyper-parameter $\beta$, the trade off between precision and recall, controlled by $\beta$, do exists.

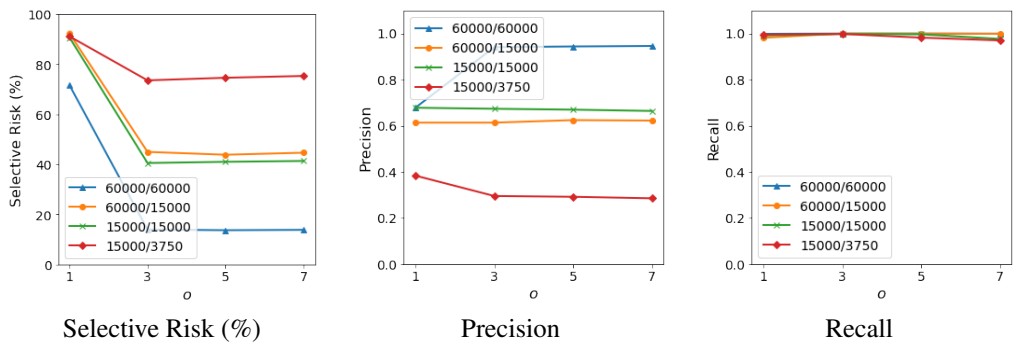

Figure 5: Ablation Study on Hyper-parameter $o$ - DeepGambler.

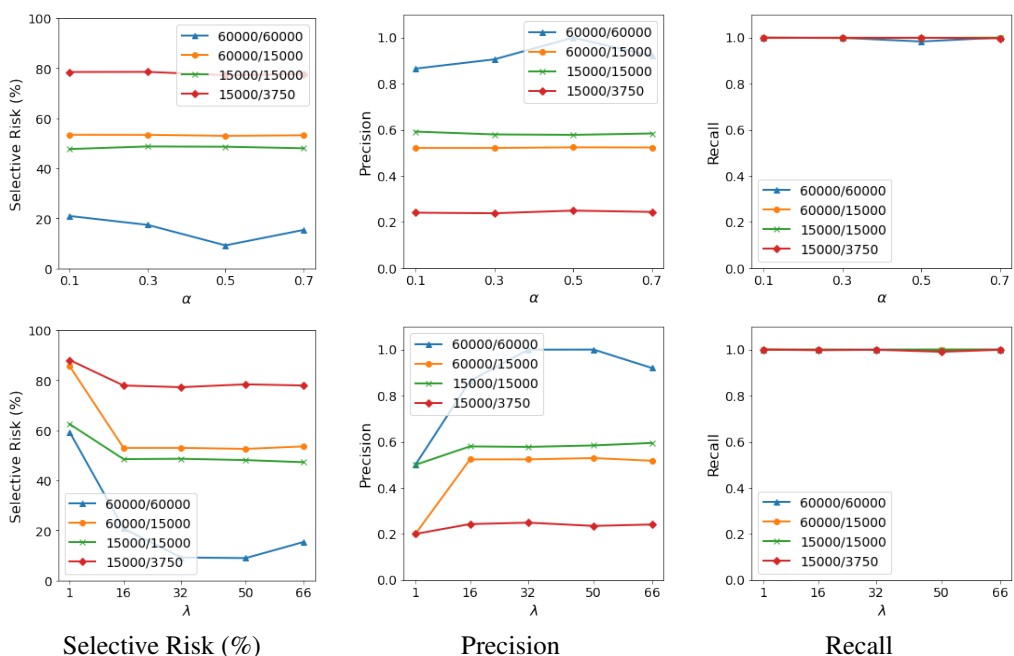

Figure 6: Ablation Study on Hyper-parameter $a$ and $\lambda_{sl}$ - SelectiveNet.

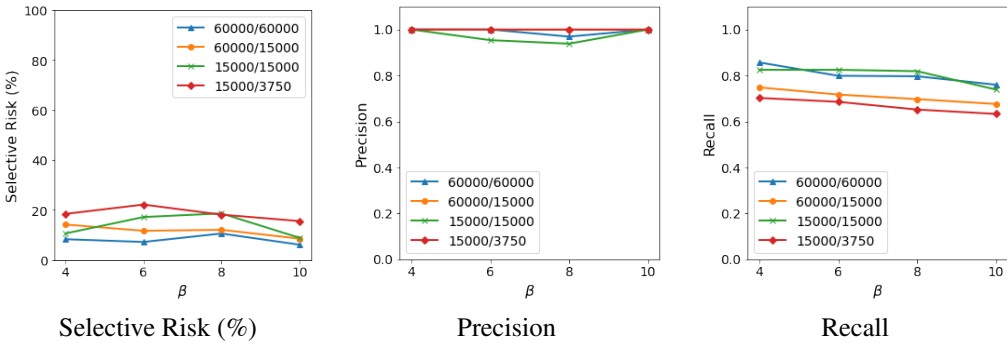

Figure 7: Ablation Study on Hyper-parameter $\beta$ - Our Method.

## C.5  ABLATION STUDY: SELECTIVE RISK $v.s$ COVERAGE LEVEL

In this section, we also the ablation study where we vary the coverage threshold and compare the selective risk of each baseline under the same coverage level (See Figure C.5). Each baseline is trained with a corrupted dataset containing 50% and 20% informative data. We use the same setting

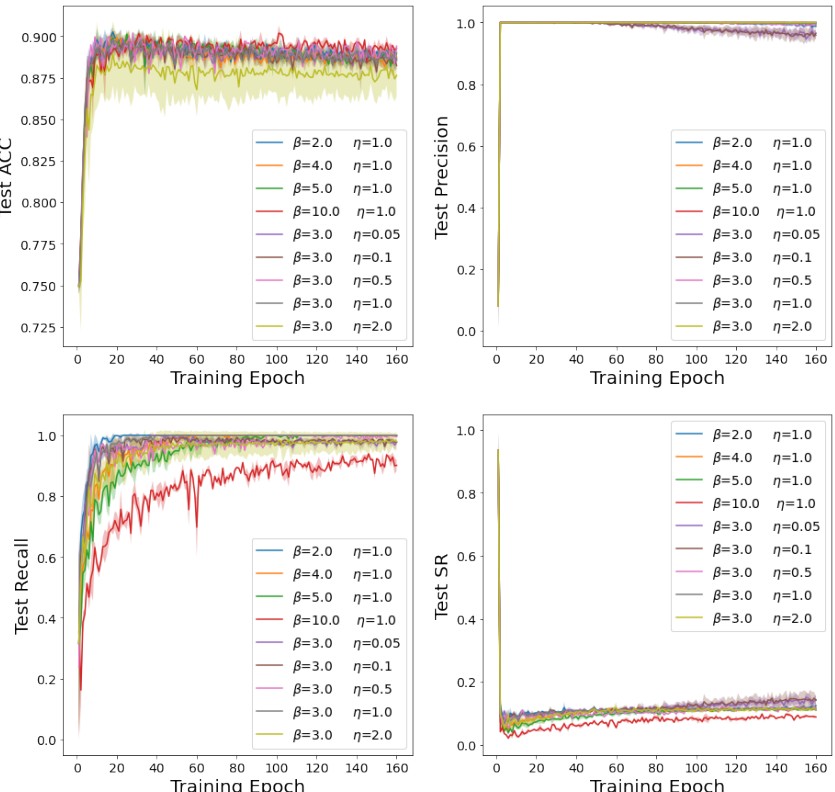

Figure 8: Convergence Curve. Experiment on Partial MNIST with $\alpha = 50\%$

as we did in section 7.2. The selective risk is computed by selecting the top coverage% confident data point. The selection confidence is measured using each baseline's selector module.

The ideal coverage($\alpha$) is indicated by the black dash line on the plot. An selective learning algorithm achieves the ideal coverage if it just cover all informative data. Coverage rate goes beyond ideal level will make the algorithm select uninformative data, which has purely random label in this case. We can see that when coverage ratio is within reasonable range compared to ground truth $\alpha$, our method outperforms all baselines. The advantage of our method is bigger when the noise ratio is high, where all other baselines show a reverse-shape coverage v.s selective risk curve. This curve implies an sever noise-over fitting issue of baseline method given few data under high noise regime.

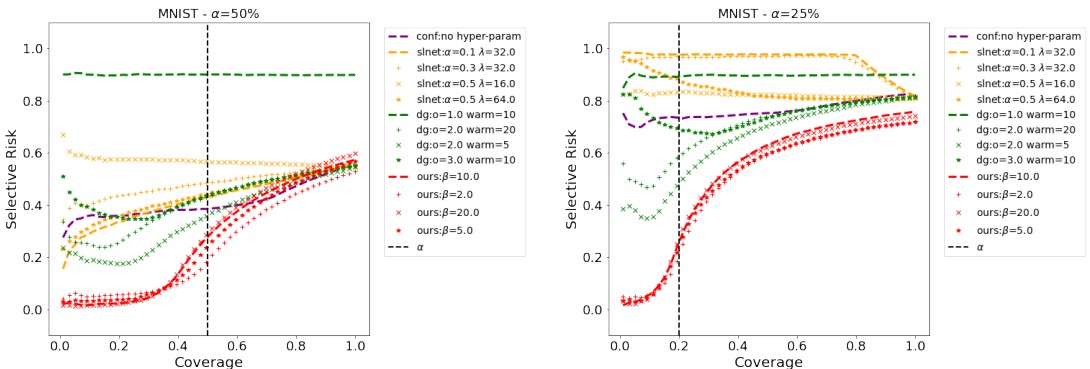

Figure 9: Coverage v.s Selective Risk Curve under Different Hyper-parameter Setting. We refer 'conf' to Confidence, 'slnet' to Selective-Net, 'dg' to DeepGambler. We plot the curve of different methods with varying hyper-parameters, e.g., $\beta$ for our method, $\lambda$ and $\alpha$ for Selective-Net, warm up epoch and $O$ for DeepGambler.

