# OpenReview forum: "Learning to Abstain from Uninformative Data"
_ICLR.cc/2023/Conference — Submitted to ICLR 2023_

### Official Review · Reviewer_3Edq · 2022-10-17

**Confidence:** 3
**Correctness:** 4
**Technical Novelty And Significance:** 3
**Empirical Novelty And Significance:** 4
**Recommendation:** 8

**Clarity, Quality, Novelty And Reproducibility:**

The work is well-written, the theoretical results are of good quality (optimal within an interesting regime), and the studied problem is interesting and (to my knowledge) novel.

**Strength And Weaknesses:**

Learning in settings with low signal-to-noise ratio is an important problem, especially in cases such as healthcare where the wrong diagnosis can be extremely costly. The model studied in this work is novel and theoretically natural, and the authors provide significant progress towards successful algorithms (minimax optimal bounds assuming a known classifier, and strong empirical results without). Furthermore, the paper is well written and provides many helpful comparisons to prior work in selective classification throughout.

The only real weakness in this paper is that the model itself seems fairly restrictive in application and it is not clear to me the extent one could truly apply the approach e.g. in healthcare, but I view this as minor since the work is theoretically interesting and a worthwhile start in modeling/approaching ML in such applications.

Minor complaint: `Shattering’ typically refers to a data subset that can take all possible values. I’d recommend calling your assumption ``G-realizable,’’ since it exactly corresponds to the typical notion of realizable setting in learning.

**Summary Of The Paper:**

The authors study learning with abstention in the regime where a large fraction of data is “uninformative,” both at train and test time. In particular the authors study a natural realizable formalization of this setting where there is a dataset X, a ground truth binary classifier $f \in \mathscr{F}$, and a ground truth `selector’ $g \in \mathscr{G}$ that determines what data is informative. Given $f$ and $g$, examples are then generated (roughly) via the following process: $x \in X$ is sampled from some arbitrary (fixed) marginal distribution, then if g(x)=1 (informative) the label y is the ground truth f(x), while if g(x)=0 (uninformative) the label y is simply Ber(1/2) purely random noise. In fact the authors consider a generalized version of this model which adds an additional Massart-like data-dependent noise parameter $\lambda$ to the second step, such that elements selected by $g(x)$ are Ber(1/2) w.p. $1-\lambda(x)$, while non-selected elements are Ber(1/2) w.p $1/2+\lambda(x)$

Given this setup, the main goal is two-fold: learn a selector g’ close to the ground truth, and a classifier on the support of g’ (informative data) with high accuracy. The authors provide two main results in this direction.

First, the authors provide a minimax optimal strategy to recover the ground truth selector given access to a good estimate of the classifier $f$ (in many settings the latter can be obtained by classical techniques such as ERM). In particular, the authors show that $g$ can be recovered up to $\varepsilon$ classification error in $\frac{\max(VC(G),VC(F))}{\varepsilon}$ along with a matching lower bound in VC(G). The algorithm and analysis are based on empirical risk minimization of a surrogate loss on g based on the accuracy of f itself, as g(x) is not revealed to the learner.

Second, the authors develop a heuristic algorithm inspired by their theoretical analysis for jointly learning a classifier f’ and selector g’ based on the classical multiplicative weights update method. They show that this algorithm outperforms other types of learning with abstention (that typically aim to balance coverage and accuracy rather than learn a ground truth selector) in scenarios where a large fraction (e.g. 80%) of the data us un-informative, both on synthetic and real datasets.

**Summary Of The Review:**

I recommend accepting this work on grounds of a theoretically interesting model for learning in the presence of uninformative data, minimax optimal sample complexity bounds, and an interesting heuristic algorithm + strong empirical performance in the regime of interest.

---

> ### Author Response · Authors · 2022-11-16
> **Response to Reviewer 3Edq**
>
>
> We thank the reviewer for careful evaluation, positive feedback, and insightful questions. We really appreciate the comments on ``$\mathcal{G}$-realizable’’ and we have made the change to make the description precise. Below we discuss the weakness pointed out by the reviewer and some future works toward addressing them.
>
> __R4-Q1:__ "The only real weakness in this paper is that the model itself seems fairly restrictive in application and it is not clear to me the extent one could truly apply the approach e.g. in healthcare, but I view this as minor since the work is theoretically interesting and a worthwhile start in modeling/approaching ML in such applications."
>
> >__R4-Ans1:__ We agree with the reviewer on the importance of empirical verification of the proposed model on real world applications.
> To better understand the merit and limitation of our proposed model, more exploration is needed on real world datasets that may fully/partially  possess   informative/uninformative structures.
> We look forward to thorough empirical studies on financial time series and health care tabular data in future work.

---

### Official Review · Reviewer_vmrJ · 2022-10-24

**Confidence:** 3
**Correctness:** 3
**Technical Novelty And Significance:** 2
**Empirical Novelty And Significance:** 2
**Recommendation:** 5

**Clarity, Quality, Novelty And Reproducibility:**

Clarity

The submission is clearly written and easy to follow.

Novelty

The methodology and the generative model are novel to me. The methodological design are interesting and inspiring.

Reproducibility

The experiment setup is provided to reproduce the experiments.

**Strength And Weaknesses:**

Strength

1. The submission is well written and easy to follow. The notations are clearly defined and the rationale of the whole submission flows smoothly.

2. The results of the submission is thorough and complete with both empirical and theoretical analysis provided for the method.

3. The methodology is also novel under the described probabilistic setting.

Weakness

1. My biggest concern is the probabilistic setting focused on by the submission. The authors assume that the informative and uninformative data are different so that can be distinguished. I could not imagine a real-world case where the data really has informative and uninformative observations, and where the two can be so different. Such a case is not provided in the real-world experiments of the submission either. All of the studied real-world experiments do not seem to satisfy the assumed data-generating model. Further, the only example mentioned that seems to satisfy this strong model, the gene mutation example, is not thoroughly studied but just mentioned in the intro.

Therefore, I am wondering whether it might be possible for authors to empirically study one real-world case, where it can be either theoretically or empirically shown that the assumed generative model is consistent with the data-generating process?

2. I am wondering what might be the main technical challenge of the proof given the assumed model and a fixed good $f$ estimator? After fixing $f$, what is the difference between the considered problem compared with a empirical risk minimization or generative model estimation? Given that the model assumed is a little strong (comment 1), would it be possible for authors to provide some theoretical analysis on the performance of the algorithm 1, without fixing the $f$ as a good estimator but learned by the algorithm? This can further strengthen the theoretical contribution of the submission.

**Summary Of The Paper:**

The submission aims to distinguish informative data and uninformative data under a designed probabilistic setting. Under this setting, the true labels of informative v.s. uninformative are missing. To deal with this issue, the authors propose a learning method simultaneously learning the predictor and the selector, with a novel loss function for the selector constructed using the learned predictor.

Both theoretical analysis and empirical results are provided to support the proposed method.

**Summary Of The Review:**

The submission is in high quality with novelty. The results are rich to support the proposed method, and the writing is also good. However, the studied problem setting (the assumed data-generating model) is a not shown to be realistic enough. There were not enough support for this assumed model in the submission. (comment 1)

On the other hand, if considered as a pure theoretical work, I am not quite sure whether the theoretical contribution of the submission is strong enough for comment 2.

---

> ### Author Response · Authors · 2022-11-16
> **Response to Reviewer vmrJ - Part II**
>
>
> __R3-Q4:__ "What might be the main technical challenge of the proof given the assumed model and a fixed good  estimator?"
>
> >__R3-Ans4:__ We do not assume a fixed good estimation of $\widehat{f}$. Our theorem is general so that ANY good classifier $\widehat{f}$ can be applied to recover selector $g$. We do not focus on recovering $\widehat{f}$ since this problem is thoroughly studies in the literature. We have a discussion on such assumption in Remark 2. Such assumption provides flexibility in terms of choosing a classifier. We thank the reviewer for raising this question. To improve the clarity, we have elaborated more and strengthened this point in Remark 1.
>
> __R3-Q5:__ "After fixing $f$, what is the difference between the considered problem compared with a empirical risk minimization or generative model estimation?"
>
> >__R3-Ans5:__ Since we do not fix  $\widehat{f}$ and the theorem holds for a family of  $\widehat{f}$, the analysis is different compared against standard ERM. To elaborate more, the  selector risk depends on the empirical error of classifier $\widehat{f}$. The empirical error of$ \widehat{f}$ may not be iid if $\widehat{f}$ is learned using the same training set. We thus use localization type technique (see [1],[2]) to bound the deviation of empirical error of $\widehat{f}$  from $f^*$ and use the error of $f^*$ in the analysis. That is why we need Lemma 2 and Lemma 8.
>
> __R3-Q6.__ "Would it be possible for authors to provide some theoretical analysis on the performance of the algorithm 1, without fixing the $\widehat{f}$ as a good estimator but learned by the algorithm?"
>
> >__R3-Ans6:__  Good point. Since this is (to the best of our knowledge) the first work  towards modeling and learning with uninformative/informative data, our analysis has only covered the informative-theoretic perspective of the problem, providing sufficient conditions for sample efficient PAC learning as well as statistical limitations of learning methods.
>
> >We agree that a theoretical analysis on Algorithm 1 would be interesting. We look forward to such extension in future work. We believe the framework in [3] could be useful in the analysis. In the interim, we also offer a purely empirical study in Figure 8 in the Appendix.
>
>
> [1] Shahar Mendelson. Improving the Sample Complexity Using Global Data. IEEE Transaction on Information Theory, 2002.
>
> [2] Peter L Bartlett, Olivier Bousquet, and Shahar Mendelson. Local Rademacher complexities. The Annals of Statistics, 2005
>
> [3] Moritz Hardt, Ben Recht, Yoram Singer. Train faster, generalize better: Stability of stochastic gradient descent. ICML 2015

---

> ### Author Response · Authors · 2022-11-16
> **Response to Reviewer vmrJ - Part I**
>
> We thank the reviewer for characterizing the paper 'well written' and 'novel',  as well as considering the method 'interesting' and inspiring'.  We hope to address reviewer's concern on the probabilistic setting via  some empirical evidence and  additional explanations on the noisy model assumption. To address reviewer's concern on the  the significance of  our theoretical contribution, we provide additional explanations to clarify our theoretical results.
>
> __R3-Q1:__  "The authors assume that the informative and uninformative data are different so that can be distinguished. I could not imagine a real-world case where the data really has informative and uninformative observations, and where the two can be so different."
>
> >__R3-Ans1:__ Indeed, we do not assume that informative and uninformative observations are 'so different'. The only assumption we make to distinguish informative and uninformative observations is a margin gap $\lambda>0$ in the label noise ratio, otherwise it replicates the celebrated Massart Noise model. We believe such problem setting is fairly relaxed since for small values of lambda, the informative and uninformative observations are indeed not quite distinguishable. As lambda goes to zero, the assumption is completely gone and our model replicates the classical Massart Noise model.(see discussion below Definition 1)
>
> >We would appreciate if the reviewer could elaborate more on concerns on the probabilistic setting. For example, the specific assumption that leaves reviewer under the impression that informative and uninformative observations are assumed to be very different. We look forward to improved our Noise generative model and further relax those assumptions accordingly in future work.
>
> __R3-Q2:__ "I could not imagine a real-world case where the data really has informative and uninformative observations, and where the two can be so different. Such a case is not provided in the real-world experiments of the submission either. All of the studied real-world experiments do not seem to satisfy the assumed data-generating model."
>
> >__R3-Ans2:__ For real world scenarios, we agree that no real world scenario will match our assumed model perfectly. However, we do believe the model is able to capture some structure present in real data. For example, some data points - outcomes of hyper-competitive bidding with lots of agents, for example - may be hard to predict, while others - less competitive situations with few participants - can exhibit more structure. We believe that such tensions are present in the real world datasets explored in the paper. One can see empirically that the original risk in Table 3 is significantly higher than the selective risk in Table2, which suggests that some data points are naturally more challenging to predict, which is what we aim to capture in the proposed model.
>
> __R3-Q3.__ "Is it possible for authors to empirically study one real-world case, where it can be either theoretically or empirically shown that the assumed generative model is consistent with the data-generating process?"
>
> >__R3-Ans3:__  Thank you for the helpful suggestion. We next present some empirical evidence that suggests the structure of informative and uninformative data may exists in the real world. We statistically verify the assumption through white noise test on the volatility dataset data studied in the experiments. We use 30 days as our sequence length in our paper real-world dataset experiments. For each such 30-day sequence, we perform Ljung correlation test to verify autocorrelations. We use Shapiro-Wilk test to verify the normality. These two together should roughly verify the independence and symmetry of distribution. Out of 143752 data points, there are 12594 (around 9\%) points where we reject both null hypothesis at 95\% significance level, which we could consider informative. For the rest 131158 data points, there are 89813 (62\%) points that fail to reject both tests. These points present some certain evidence to say they are independently distributed Gaussian signals and thus fulfill our uninformative data generation model. From the statistical perspective, we can consider this dataset as containing a majority of uninformative data.
>
> >We believe a rigorous verification of such assumptions itself could be a challenging and non-trivial work, which requires tremendous effort and deserves another paper.

---

> ### Author Response · Authors · 2022-11-28
> **Looking forward to further discussion**
>
> Dear Reviewer vmrJ,
>
> We really appreciate your time and effort during this busy time. We sincerely hope our discussions and updated manuscript have addressed your concerns.  If there are any further questions, please do not hesitate to let us know.  We remain attentive to your feedback.
>
> Sincerely,
>
>  Paper 4766 authors

---

### Official Review · Reviewer_UQnf · 2022-10-25

**Confidence:** 2
**Correctness:** 2
**Technical Novelty And Significance:** 3
**Empirical Novelty And Significance:** 2
**Recommendation:** 5

**Clarity, Quality, Novelty And Reproducibility:**

Clarity: the method in Algorithm 1 is clear. The metrics are not.

Novelty: I cannot comment on the theoretical novelty. The method as a whole is novel, but the iterative optimization of selector and predictor seems like a natural extension of the problem.

Reproducible: Yes, as the authors promise to release the code with the final submission.

**Strength And Weaknesses:**

Strengths:
- Theoretically grounded
- The method seems to perform better than baselines on synthetic data

Weaknesses:
- I find the metrics confusing. Selective Risk by definition seems to favor methods with high-precision and low recall, so I'm not sure if by itself this is such a useful metric
- Relatedly, we also see that their method has high-precision but low recall. In Table 1: their method out-performs baselines in selective risk, but has lower recall and higher precision. The same is also seen in Table 2: the method is out-performing baselines at low coverage values, but drops off at 20% and higher coverage numbers are not provided. I'm not sure if the results validate the claim.
- Only two baselines are compared. One related baseline to use could be DAC (Combating Label Noise in Deep Learning Using Abstention, Sunil Thulasidasan et. al., 2019).

**Summary Of The Paper:**

The authors propose a method for learning when there is high amount of label noise present, using an iterative bi-level optimization. Their method in an iterative fashion first trains a classifier (predictor), then defines a pseudo-labels for the selector (abstain) classifier using confidence of the trained predictor on the target class. These pseudo-labels are used to train the selector classifier. The selector classifier then updates sample weights for the training of the next round of predictor.
They also formulate a noisy generative process to describe how the data was generated and provide bounds on the accuracy of the selector given a reasonably good predictor.

Their method is validated using both synthetically generated datasets (by combining MNIST and Fashion MNIST as well as a noisy SVHN) and real datasets (breast ultrasound images, lending club, and Oxford realized volatility). The method is however only compared with three baselines: DeepGambler, SelectiveNet, and a simple (singe-level) confidence-based selector.

**Summary Of The Review:**

I cannot comment on the theory and am not an expert in this field, but the empirical results are not convincing for me. The model can simply trade-off lower recall for higher-precision to get low Selective Risk. A metric such as average precision could have been more convincing. I would also like to know how the selector and predictor adapt as the number of iterations (T) increases. I am not confident that this paper reaches the threshold for acceptance.

---

> ### Author Response · Authors · 2022-11-16
> **Response to Reviewer UQnf - Part II**
>
> __R2-Q2:__ I would also like to know how the selector and predictor adapt as the number of iterations ($T$) increases.
>
> >__R2-Ans2:__ Thank you for pointing this out. Indeed reviewer rpje also raises the question about the convergence of the algorithm. We now have included some additional empirical evidence to characterize the convergence behavior of the algorithm in Figure-8 in the appendix. As one can see, the algorithm is not sensitive to hyper-parameter $\eta$ and it saturates and stabilizes when $T$ is reasonably large. The results also suggests that the method is not sensitive to hyper-parameter $\beta$ within a wide range. Please refer to section C.4 in the appendix for more details.
>
> __R2-Q3:__ One related baseline to use could be DAC (Combating Label Noise in Deep Learning Using Abstention, Sunil Thulasidasan et. al., 2019).
>
> >__R2-Ans3:__ Thank you for suggesting the extra baseline. We present the experiment result for DAC in Table 3-7. We use the challenging partial data setting as we use in the paper. We adopt the same blind setting and assume both methods don't know the exact informative-uninformative ratio $\alpha$. DAC will abstain a datapoint whenever its extra neuron dominates the rest output neurons. Due to the time limit, we only show experiments for $\alpha=0.5$ and $\alpha=0.20$. We will add more values for $\alpha$ in the final version of the paper.
>
> >We can see that our method has better performance than DAC on MNIST while DAC has better performance than ours on SVHN.  We found that the performance of DAC can be very sensitive to the warm-up period in majority noise setting. Our method is not sensitive to hyper-parameter $\beta$ within reasonable range according to Theorem 1.
>
> >We also test the baseline on the 3 real world datasets, where we optimize the hyper-parameter of each method using 50 random search trials on volatility and BUS datastes, and 30 random search trials on LC dataset. We can see that our method has better performance than DAC for each coverage level.
>
> >Table 3. DAC v.s. Ours - Selective Risk
> | DataSet | Uninformative Data Num. | Informative Data Num. | DAC | Ours |
> |---|---|---|---------|----------------|
> | MNIST   | 15000   | 3750      |  46.04$\pm$0.03  | 20.24$\pm$9.25 |
> | MNIST   | 15000   | 15000     |  19.90$\pm$0.02  | 19.10$\pm$4.50 |
> | SVHN    | 9200    | 2285      |  10.06$\pm$0.03  | 14.36$\pm$0.08 |
> | SVHN    | 9200    | 9200      |   5.33$\pm$0.01  |  7.22$\pm$0.54 |
>
> >Table 4. DAC v.s. Ours - Precision
> | DataSet | Uninformative Data Num. | Informative Data Num. | DAC | Ours|
> |---------|--------------|-----------|---------|---------------|
> | MNIST   | 15000    | 3750       |  0.62$\pm$0.03    | 1.00$\pm$0.00 |
> | MNIST   | 15000    | 15000      |  0.88$\pm$0.01    | 0.99$\pm$0.01 |
> | SVHN    | 9200     | 2285       |  0.91$\pm$0.02    | 0.87$\pm$0.07 |
> | SVHN    | 9200     | 9200       |  0.95$\pm$0.01    | 0.94$\pm$0.01 |
>
> >Table 5. DAC v.s. Ours - Recall
> | DataSet | Uninformative Data Num. | Informative Data Num. | DAC | Ours|
> |---------|---------------|---------|---------|---------------|
> | MNIST   | 15000       | 3750      | 0.63$\pm$0.04   | 0.88$\pm$0.07 |
> | MNIST   | 15000       | 15000     | 0.93$\pm$0.02   | 0.91$\pm$0.04 |
> | SVHN    | 9200        | 2285      | 0.83$\pm$0.01   | 0.80$\pm$0.02 |
> | SVHN    | 9200        | 9200      | 0.94$\pm$0.01   | 0.88$\pm$0.01 |
>
> >Table 6. DAC v.s. Ours - Average F1
> | DataSet | Uninformative Data Num. | Informative Data Num. | DAC | Ours|
> |---------|---------------|---------|---------|---------------|
> | MNIST   | 15000       | 3750      | 0.62   | 0.94 |
> | MNIST   | 15000       | 15000     | 0.90   | 0.95 |
> | SVHN    | 9200        | 2285      | 0.87   | 0.83 |
> | SVHN    | 9200        | 9200      | 0.94   | 0.91 |
>
> >Table 7. DAC v.s. Ours - Selective Risk on 3 Real-world Dataset
> | Coverage | Volatility-DAC | Volatility-Ours | BUS-DAC | BUS-Ours        | LC-DAC | LC-Ours |
> |----------|------------|--------|------|--------|-------|-------------|
> | 0.02     |  0.421$\pm$0.013 | 0.046$\pm$0.002 | 0.583$\pm$0.312 | 0.000$\pm$0.000 | 0.539$\pm$0.004  | 0.136$\pm$0.013 |
> | 0.05     |  0.411$\pm$0.018 | 0.073$\pm$0.003 | 0.667$\pm$0.118 | 0.000$\pm$0.000 |  0.529$\pm$0.004 | 0.177$\pm$0.013 |
> | 0.10     |  0.397$\pm$0.012 | 0.116$\pm$0.004 | 0.563$\pm$0.088 | 0.042$\pm$0.059 |  0.519$\pm$0.005 | 0.221$\pm$0.010 |
> | 0.20     |  0.388$\pm$0.016 | 0.192$\pm$0.005 | 0.615$\pm$0.082 | 0.083$\pm$0.029 |  0.503$\pm$0.005 | 0.271$\pm$0.007 |

---

> ### Author Response · Authors · 2022-11-18
> **Response to Reviewer UQnf - Part I**
>
> We thank the reviewer for acknowledging the theoretical contribution and novelty of the method as strength of the paper as well as considering heuristic algorithm as a natural extension of the problem. In the response below we hope to address reviewer's concern with additional explanations and empirical results.
>
> __R2-Q1:__ " Selective risk .. I'm not sure if by itself is such a useful metric."..."Their algorithm can trade recall for better precision"..."average precision could have been more convincing"..."In real world experiment, they outperform only at 20\% level and don't provide the result when coverage > 20\%"
>
> >__R2-Ans1:__ Instead of F1 and average precision, we focus on precision and recall in order to better analyze our method’s benefit with regard to the false positive/negative error. We have included F1 score in Table 1 and 2. As one can see, the advantage of our method in F1 score is significant especially in settings where data is limited.
>
> >As for the higher recall of the baselines, it is not surprising and actually implies a loss of specificity of their selector. These methods indeed tend to select all data points indistinguishably to achieve perfect recall. To put it in another way, most baselines sacrifices precision to achieve good recall, whereas our method strikes a good balance between the two. In Figure 9 in the appendix we present the selective risk of all methods under different coverage levels. While most methods have similar selective risk at coverage=1, our method significantly outperforms other baselines when coverage is within reasonable range w.r.t the ground truth information data ratio, suggesting its superior capability in picking out informative data.
>
> >In the real-world dataset experiments, we mainly show the strength of our method in capturing highly informative data, which is of practical interest in areas like finance and healthcare, where negative consequences caused by some decision are significantly more severe than that caused by others.
>
>
> >Table 1: Average F1 Score on Complete Synthetic Dataset
> | Dataset | Informative Data | Uninformative Data | Confidence (F1) | SLNet (F1) | DeepGambler (F1) | Ours (F1) |
> |---------|------------------|--------------------|-----------------|------------|------------------|-----------|
> | MNIST   | 60000            | 15000              | 0.62            | 0.76       | 0.76             | **0.92**  |
> | MNIST   | 60000            | 30000              | 0.80            | 0.89       | 0.89             | **0.96**  |
> | MNIST   | 60000            | 45000              | 0.88            | 0.94       | 0.94             | **0.98**  |
> | MNIST   | 60000            | 60000              | 0.92            | 0.96       | 0.97             | **0.98**  |
> | SVHN    | 33800            | 9200               | 0.69            | 0.66       | 0.87             | **0.89**  |
> | SVHN    | 33800            | 18300              | 0.82            | 0.84       | **0.93**         | 0.91      |
> | SVHN    | 33800            | 26200              | 0.88            | 0.93       | **0.94**         | 0.93      |
> | SVHN    | 33800            | 28400              | 0.88            | 0.93       | **0.95**         | 0.92      |
>
> >Table 2: Average F1 Score on Partial Synthetic Dataset
> | Dataset | Informative Data | Uninformative Data | Confidence (F1) | SLNet (F1) | DeepGambler (F1) | Ours (F1) |
> |---------|------------------|--------------------|-----------------|------------|------------------|-----------|
> | MNIST   | 15000            | 3750               | 0.35            | 0.45       | 0.46             | **0.94**  |
> | MNIST   | 15000            | 7500               | 0.54            | 0.65       | 0.65             | **0.91**  |
> | MNIST   | 15000            | 11250              | 0.64            | 0.75       | 0.75             | **0.91**  |
> | MNIST   | 15000            | 15000              | 0.71            | 0.81       | 0.81             | **0.95**  |
> | SVHN    | 9200             | 2285               | 0.57            | 0.52       | 0.75             | **0.83**  |
> | SVHN    | 9200             | 4600               | 0.71            | 0.73       | 0.85             | **0.89**  |
> | SVHN    | 9200             | 6900               | 0.78            | 0.85       | 0.89             | **0.90**  |
> | SVHN    | 9200             | 9200               | 0.79            | 0.86       | 0.90             | **0.91**  |

---

> ### Author Response · Authors · 2022-11-28
> **Looking forward to further discussion**
>
> Dear Reviewer UQnf,
>
> We really appreciate your time and effort during this busy time. We sincerely hope our discussions and updated manuscript have addressed your concerns.  If there are any further questions, please do not hesitate to let us know.  We remain attentive to your feedback.
>
> Sincerely,
>
> Paper4766 Author

---

### Official Review · Reviewer_rpJe · 2022-10-25

**Confidence:** 2
**Correctness:** 3
**Technical Novelty And Significance:** 3
**Empirical Novelty And Significance:** 2
**Recommendation:** 5

**Clarity, Quality, Novelty And Reproducibility:**

The paper is dense and rather long for a short reviewing period. I did not fully check all the mathematical derivations.

No source code is submitted so I could not comment on the reproducibility.

**Strength And Weaknesses:**

Strength:
-Theory: the paper provides a minimax risk bound for the selector $g$
-Practice: the paper contains an iterative algorithm to find the pair of predictor-selector (f, g).
-  Even though the algorithm is heuristics, the performance is relatively well.

Weakness:
-Theorem 1 depends on the parameter $\lambda$, which is rarely known.
-The risk bound is only for the selector $\hat g$, and there is no joint risk bound for the pair $(\hat f, \hat g)$.
-There is a huge gap between Section 5 and Section 6. The algorithm in Section 6 contains several relaxations, including (i) cross-entropy loss replacing the binary loss, (ii) continuous output of $f$ and $g$ instead of binary. Thus, it is unclear how the bound in Section 5 can provide any guarantees on the performance of the predictor-selector pair found in Section 6.
-There is no convergence guarantee for Algorithm 1. It is not clear how to choose $T$ and $\eta$ so that we can have favorable convergence results.
- It is desirable to provide experimental comparisons against Gangrade et al. (2021) [Selective classification via one-sided prediction] because the method therein outperforms both DeepGambler and SelectiveNetwork.


**Summary Of The Paper:**

This paper studies the Selective classification problem when data contains many noisy samples which should be filtered out. The authors present a novel loss function to train the selector g (used to classify whether data samples are informative or uninformative) given predictor f. They also prove that the optimal selector can be approximated when training with the proposed loss function. Finally, this paper presents an iterative algorithm that trains the selector in parallel with the predictor and provides empirical results demonstrating promising performance over baselines in both small datasets (Fashion Mnist + mnist, SVHN), semi-synthetic datasets and real-world datasets (breast ultrasound images, lending club dataset, and Oxford realized volatility dataset).


**Summary Of The Review:**

The paper is heavy on the theoretical side (Theorem 1), however, it is not clear whether this bound is of significance for scientific understanding and future research.

There is a big gap between the theory (Section 5) and the practice (Section 6). I wish the paper can establish a coherent setup throughout with minimal relaxations.

---

> ### Author Response · Authors · 2022-11-16
> **Response to Reviewer rpJe - Part III**
>
> __R1-Q7.__ "It is desirable to provide experimental comparisons against Gangrade et al. (2021) [Selective classification via one-sided prediction] because the method therein outperforms both DeepGambler and SelectiveNetwork."
>
> >__R1-Ans7:__ Per reviewer's request, we present experiment results of one-sided prediction in Table 1-5. We use the challenging partial data setting as we use in the paper. We adopt the same blind setting and assume both methods don't know the exact informative to uninformative ratio $\alpha$. The one-sided prediction adopts the same estimation procedure as all other baselines used in our paper to estimate this ratio.  Due to the time limit, we only show experiments under $\alpha=0.5$ and $\alpha=0.20$. We will add more values for $\alpha$ in the final version of the paper.
>
> >We can see that our method outperforms one-sided prediction by a large margin on both selective risk and precision. One-sided prediction still has the same informative data identification difficulties shared by other baselines, thus it fails to make a reasonable selection. The observed high recall actually indicates this issue.
>
> >One can also evaluate the bi-criterion (precision and recall) results through the F1 score in Table 4, which balances these two criterias. As we can see, our method has higher F1 score than one-sided prediction.
>
> >Table 1. Oneside v.s. Ours - Selective Risk
> | DataSet | Uninformative Data Num. | Informative Data Num. | OneSide | Ours |
> |---|---|---|---------|----------------|
> | MNIST   | 15000   | 3750      |  77.31$\pm$0.009 | 20.24$\pm$9.25 |
> | MNIST   | 15000   | 15000     |  44.08$\pm$0.001 | 19.10$\pm$4.50 |
> | SVHN    | 9200    | 2285      |  44.35$\pm$0.471 | 14.36$\pm$0.08 |
> | SVHN    | 9200    | 9200      |  41.04$\pm$0.256 |  7.22$\pm$0.54 |
>
> >Table 2. Oneside v.s. Ours - Precision
> | DataSet | Uninformative Data Num. | Informative Data Num. | OneSide | Ours|
> |---------|--------------|-----------|---------|---------------|
> | MNIST   | 15000    | 3750       |  0.26$\pm$0.01 | 1.00$\pm$0.00 |
> | MNIST   | 15000    | 15000      |  0.62$\pm$0.01 | 0.99$\pm$0.01 |
> | SVHN    | 9200     | 2285       |  0.58$\pm$0.04 | 0.87$\pm$0.07 |
> | SVHN    | 9200     | 9200       |  0.62$\pm$0.29 | 0.94$\pm$0.01 |
>
> >Table 3. Oneside v.s. Ours - Recall
> | DataSet | Uninformative Data Num. | Informative Data Num. | OneSide | Ours|
> |---------|-------------|------------------|---------|---------------|
> | MNIST   | 15000       | 3750      |   0.90$\pm$0.03 | 0.88$\pm$0.07 |
> | MNIST   | 15000       | 15000     |   0.96$\pm$0.00 | 0.91$\pm$0.04 |
> | SVHN    | 9200        | 2285      |   0.89$\pm$0.01 | 0.80$\pm$0.02 |
> | SVHN    | 9200        | 9200      |   0.94$\pm$0.10 | 0.88$\pm$0.01 |
>
> >Table 4. Oneside v.s. Ours - F1 (Calculated using the averaged precision and averaged recall value)
> | DataSet | Uninformative Data Num. | Informative Data Num. | OneSide | Ours|
> |---------|-------------|------------------|---------|---------------|
> | MNIST   | 15000       | 3750      | 0.40 | 0.94 |
> | MNIST   | 15000       | 15000     | 0.75 | 0.95 |
> | SVHN    | 9200        | 2285      | 0.70 | 0.83 |
> | SVHN    | 9200        | 9200      | 0.74 | 0.91 |
>
> >Table 5. Oneside v.s. Ours - Selective Risk on 3 Real-world Dataset
> | Coverage | Volatility-OneSide | Volatility-Ours | BUS-OneSide | BUS-Ours        | LC-OneSide | LC-Ours         |
> |----------|------------|--------|------|--------|-------|-------------|
> | 0.02     |   0.061$\pm$0.002     | 0.046$\pm$0.002 | 0.000$\pm$0.000     | 0.000$\pm$0.000 | 0.430$\pm$0.016  | 0.136$\pm$0.013 |
> | 0.05     |   0.084$\pm$0.003     | 0.073$\pm$0.003 |    0.042$\pm$0.059 | 0.000$\pm$0.000 |  0.289$\pm$0.007         | 0.177$\pm$0.013 |
> | 0.10     |   0.132$\pm$0.001    | 0.116$\pm$0.004 |     0.083$\pm$0.059| 0.042$\pm$0.059 |  0.252$\pm$0.003 | 0.221$\pm$0.010 |
> | 0.20     |   0.174$\pm$0.002    | 0.192$\pm$0.005 |     0.177$\pm$0.090| 0.083$\pm$0.029 |  0.263$\pm$0.005 | 0.271$\pm$0.007 |

---

> ### Author Response · Authors · 2022-11-16
> **Response to Reviewer rpJe - Part II**
>
> __R1-Q5.__ "There is no convergence guarantee for Algorithm 1. It is not clear how to choose $\eta$ and $T$ and  so that we can have favorable convergence results."
>
> >__R1-Ans5:__ Thank you for pointing this out. Indeed reviewer Uqnf also raises the question about the convergence of the algorithm. We have added new empirical evidence to characterize the convergence behavior of the algorithm in Figure-8 in the appendix. As we can see, algorithm saturates and stabilizes when $T$ is reasonably large. The results also suggests that the method is not sensitive to hyper-parameter $\eta$  and  $\beta$ within a wide range.  Please refer to section C.4 in the appendix for more details.
>
> >We appreciate reviewer's suggestion regarding the convergence analysis, we believe this could be an interesting and important work. Nevertheless, this is the (to the best of our knowledge) first work towards modeling and learning with uninformative/informative data. Our analysis mainly focus on the information-theoretic perspective of the problem, providing sufficient conditions for sample efficient PAC learning as well as statistical limitations of learning methods. As pointed out by the reviewer, the paper is already dense, thus we look forward to analyzing the convergence of the algorithm in future work.
>
> __R1-Q6.__ "The paper is heavy on the theoretical side (Theorem 1), however, it is not clear whether this bound is of significance for scientific understanding and future research."
>
> >__R1-Ans6:__ We want to stress that this paper studies an important yet understudied problem in ML. We propose a natural noisy model, closely related to the celebrated Massart label noise model, to characterize a situation where some data points are naturally noisier than the others. The theoretical analysis suggests that the selector loss proposed in the paper is not only the 'correct' loss, it also achieves minimax optimal sample complexity rate. We believe the theoretical analysis provide thorough answers for questions on the information theoretic perspective of the noisy model. There are several future directions that can be explored given our theoretical results. For example, extending the analysis of binary loss to convex surrogate losses such as hinge loss and cross-entropy loss that can be efficiently optimized, as suggested by the reviewer.

---

> ### Author Response · Authors · 2022-11-16
> **Response to Reviewer rpJe - Part I**
>
> We thank the reviewer for acknowledging the theoretical contribution and good performance of heuristic algorithm as strengths of the paper. Below we address the reviewer's concerns via 1) providing clarifications regarding our theoretical results, and 2) providing additional empirical results according to the suggestions. We will incorporate these clarifications and results into our manuscript.
>
>
> __R1-Q1.__ "Theorem 1 depends on the parameter $\lambda$, which is rarely known. "
>
> >__R1-Ans1:__ You are correct that $\lambda$ is a problem-dependent constant and is rarely known. We introduce it in the sample complexity analysis in order to derive a tight rate, which matches the information theoretic lower bound. The existence of a problem-dependent bound is common in statistical learning theory research in order to characterize the complexity tightly (see [1,2,3,4] etc). We would like to stress that $\lambda$ is not a hyper-parameter required by our algorithm: corollary 1 suggests that choice of $\beta=3$ ensures the PAC guarantee, regardless the value of lambda.
>
> __R1-Q2.__ "The risk bound is only for the selector, and there is no joint risk bound for the pair $f$ and $g$"
>
> >__R1-Ans2:__ We do not focus on the risk bound for $f$ since it is well studied in the literature [2,3]. As stated in remark 2, instead of estimating $\widehat{f}$ in a specific way, the assumption that $f$ is a good estimator makes our theorem more general; the strategy to achieve such $f$ can be flexible. We thank the reviewer for raising this question. To improve the clarity, we have elaborated more and strengthened this point in Remark 1.
>
>
> __R1-Q3.__ "There is a huge gap between Section 5 and Section 6. The algorithm in Section 6 contains several relaxations, including (i) cross-entropy loss replacing the binary loss, (ii) continuous output of  and  instead of binary. ..."
>
> >__R1-Ans3:__ We believe it is a common practice to analyze using 0-1 loss, but implement cross-entropy loss as a convex surrogate in DL research. This is the case in [5,6] and in the suggested baseline (Gangrade et al. 2021). The reason is that the binary loss is a natural choice for PAC learning in statistical learning theory, whereas cross entropy is a better choice in practice, because 0-1 loss is uniformly bounded and has nice concentration properties. We do agree with the reviewer that a theoretical analysis generalizing from binary to some convex surrogate loss could be an important and non-trivial work, which we wish to explore in future.
>
>
> __R1-Q4.__ "It is unclear how the bound in Section 5 can provide any guarantees on the performance of the predictor-selector pair found in Section 6."
>
> >__R1-Ans4:__ We agree with the reviewer that these bounds are not directly applicable to the practical algorithm due to several relaxations and heuristics introduced for practical purposes. However, our algorithm is motivated by and derived from our theoretical analysis. We would like to highlight the following connections between the theorem and the algorithm:
>
> >1) Theorem 1 suggests that the ground-truth selector model $g$ can be 'correctly' learned using the selector loss. The algorithm applies the cross-entropy loss as a convex surrogate of the selector loss analyzed in the Theorem.
> >2) The theorem admits any good classifier $f$, which allows us to use the iterative re-weighting heuristic algorithm to train $f$ beyond standard ERM.
> >3) Theorem 1 provides guidance in choosing hyper-parameter beta ($\beta=3$) and suggests that the choice of beta can be flexible. Empirical study supports the claim that the algorithm's performance is not sensitive to the choice of the hyper-parameter $\beta$. (see Figure 7 in the paper's appendix)
> >4) Empirical study supports Theorem 1 as the algorithm recovers $g$ reasonably well. We also observe coherent relationship between recovery error and problem dependent constant $\lambda$ as suggested in Theorem 1 in Figure 2.
>
> [1] Shahar Mendelson. Improving the sample complexity using global data.  IEEE transactions on Information Theory, 2002
>
> [2] Peter L Bartlett, Olivier Bousquet, and Shahar Mendelson. Local rademacher complexities. Local Rademacher complexities. The Annals of Statistics, 2005
>
> [3] Pascal Massart and Élodie Nédélec. Risk bounds for statistical learning. The Annals of Statistics, 2006
>
> [4] Ilias Diakonikolas, Themis Gouleakis, and Christos Tzamos. Distribution-independent pac learning of halfspaces with massart noise.  Advances in Neural Information Processing Systems, 2019
>
> [5] Nontawat Charoenphakdee, Zhenghang Cui, Yivan Zhang, Masashi Sugiyama. Classification with Rejection Based on Cost-sensitive Classification, ICML 2019
>
> [6] Jiacheng Cheng, Tongliang Liu, Kotagiri Ramamohanarao, Dacheng Tao. Learning with bounded instance and label-dependent label noise, ICML 2020

---

> ### Author Response · Authors · 2022-11-28
> **Looking forward to further discussion**
>
> Dear Reviewer rpJe,
>
> We really appreciate your time and effort during this busy time. We sincerely hope our discussions and updated manuscript have addressed your concerns.  If there are any further questions, please do not hesitate to let us know.  We remain attentive to your feedback.
>
> Sincerely,
>
> Paper4766 Author

---

### Official Review · Reviewer_8f3z · 2022-12-02

**Confidence:** 3
**Correctness:** 3
**Technical Novelty And Significance:** 2
**Empirical Novelty And Significance:** 3
**Recommendation:** 5

**Clarity, Quality, Novelty And Reproducibility:**

The paper is well written except some further discussions and clarifications are needed. See section above.

Novelty lies in including a different cost associated with the abstained points, deriving new theory and algorithms under the high noise setting.

The results seem reproducible.


**Strength And Weaknesses:**

Strengths of the paper

1. Introduce an interesting loss to quantify the benefits of the selector g according to the correctness of the predictor f.
2. Authors present both theoretical and empirical aspects of the framework.
3. Proposed algorithm optimizes simultaneously the predictor and selector function.

Weakness of the paper:

Theoretically:
1. Theorem 1 is only a statement about selecting the selector g given that there is an \hat{f} that is a good predictor. Having a joint result on (f,g) would greatly improve the paper. Even though a good predictor \hat{f} may be achieved on the noisy data under some assumptions, couldn’t an even better predictor be learned on the data selected by g, that is on data that is ideally less noisy?
2. More detailed discussion on how a good predictor \hat{f} can be achieved is needed. They claim that it could be achieved  “under some margin condition” – are these additional assumptions or can it be part of assumption 1?
3. There is a large gap between the theoretical analysis and the presented algorithm as a series of relaxations are undertaken. In the theory section, they claim that “In practice, one can also apply some methods beyond ERM to obtain \hat{f} ”, yet they do not apply these methods in the presented algorithm and instead try to solve a joint optimization over f and g.
4. Compared to the abstention loss in Cortes et al., the main difference is that the cost of abstention is replaced by 1_{f(x) = y}. This should be highlighted in the main body. It is also unclear why a joint analysis on (f,g) as was done in Cortes et al could not be extended in this case since having a term 1_{f(x) = y} 1_{g>0} is as difficult to deal with as the already present term 1_{f(x) != y} 1_{g<0} in the abstention loss.

Experimentally:

1. Metrics are not clearly defined.
2. I may have misunderstood the metrics since they are not clearly defined, but as far as I can tell the experiments using the semi-synthetic data for Q2 are not convincing since recall for their method is always lower than other methods. If their algorithm selects very few examples – leading to low recall, then it will have naturally high precision and high SR. Thus it is hard to compare the methods.
3. Experiments using real world datasets only test 3 datasets and the proposed method outperforms DeepGambler on only 1 dataset.



**Summary Of The Paper:**

 Authors propose a method for learning with high label noise that is based on learning a predictor and a selector function. They introduce a loss function that weights the abstained points by the predictor’s correctness. They present a theoretical analysis in the noisy setting for picking the best selector function given the availability of a good predictor f. They derive an algorithm that jointly optimizes the predictor and selector along with some empirical evidence.

**Summary Of The Review:**

Even though I commend the authors for deriving both theoretical analysis and testing their proposed methods in practice, both the theoretical and experimental sections admit several weaknesses that combined together place the paper below acceptance threshold. In the theory section, they claim that predictor f and selector g do not need to be optimized jointly and present a result based on the availability of a good prediction function f, but then in the empirical section proceed to present an algorithm that optimizes f and g jointly. This conceptual mismatch needs to be addressed. I do believe that if the authors strengthen their empirical section by showing more results in the real world dataset regime, this paper will be easily accepted into a top tier conference.

---

> ### Author Response · Authors · 2022-12-04
> **Response to Reviewer 8f3z II**
>
> __R0-Q4__: "There is a large gap between the theoretical analysis and the presented algorithm as a series of relaxations are undertaken. In the theory section, they claim that “In practice, one can also apply some methods beyond ERM to obtain $\hat{f}$ ”, yet they do not apply these methods in the presented algorithm and instead try to solve a joint optimization over $f$ and $g$."
>
> >__R0-Ans4__: Indeed it is not obvious to us why the gap is considered large.  The crucial relaxation involved is basically 1) binary loss 2) ERM oracle to iterative SGD. We believe such relaxation is a convention in the ML community. (see response to Q3 of Reveiwer rpJe). What might leave readers an impression on the gap is the MWU schema. The MWU is introduced solely for the sake of better classifiers $f$. Which is essentially the ‘method beyond ERM’ discussed in previous discussions regarding theoretical challenges (Q2). We would like to stress that the MWU schema does not contradict our theorem at all since our theorem is conditional on ‘any’ good classifier.
>
> ### Concern Regarding Experiments
>
> __R0-Q5__: "Metrics are not clearly defined."
>
> >__R0-Ans5__: Thank you for the suggestion. We will make sure to include the exact formula of the metrics, instead of using plain language. We use selective risk, prediction, and recall. Per Reviewers’ request, we also added an F1 score. The selective risk is borrowed from the selective learning literature. All others are pretty standard. The formulations are: SR=$\frac{| \\{x | \hat{f}(x)=y , \hat{g}(x)>0.5 \\} |}{| \\{ x | \hat{g}(x)>0.5 \\} |}$,  Precision=$\frac{ | \\{ x | \hat{g}(x)>0.5,  g^*(x)>0.5 \\} | }{ | \\{ x | \hat{g}(x)>0.5 \\} | }$, Recall=$\frac{ | \\{ x | \hat{g}(x)>0.5,  g^*(x)>0.5 \\} | }{ | \\{ x | g^*(x)>0.5 \\} | }$ and F1=$\frac{2*\text{Precision}*\text{Recall}}{\text{Precision}+\text{Recall}}$.
>
> __R0-Q6__: "I may have misunderstood the metrics since they are not clearly defined, but as far as I can tell the experiments using the semi-synthetic data for Q2 are not convincing since recall for their method is always lower than other methods. If their algorithm selects very few examples – leading to low recall, then it will have naturally high precision and high SR. Thus it is hard to compare the methods."
>
> >__R0-Ans6__: In the rebuttal period we have added F1 score in Table 1 and 2 in per request of Reviewer UQnf (in his/her Q1).
> When using the harmonic mean of precision of recall (i.e., F1), the benefit of our method is pretty significant especially in limited-data settings. The higher recall of the baselines is not surprising, and actually implies a lack of specificity of their selector. These methods indeed tend to select all data points indistinguishably to achieve perfect recall. To put it in another way, most baselines sacrifice precision to achieve good recall, whereas our method strikes a good balance between the two (and thus a better F1). In Figure 9 in the appendix we present the selective risk of all methods under different coverage levels. When coverage is close to the ground truth information data ratio, our method significantly outperforms other baselines. This implies that our method is superior in picking out the informative data.
>
> __R0-Q7__: Experiments using real world datasets only test 3 datasets and the proposed method outperforms DeepGambler on only 1 dataset.
>
> >__R0-Ans7__: Our method outperforms Deep Gambler on one dataset and remains consistently competitive (on par) on other two datasets. We would like to highlight that our method mainly stands out when there is a label noise ratio gap in the generative process. In the semi-synthetic experiments, where such conditions hold, our method outperforms all other baselines by a large margin. We believe this is the case for the Volatility dataset (see response to Q3 of Reviewer  vmrJ), and thus our method outperforms the others.
>
> >As a final note, the real world data can be highly heterogeneous and no single model can explain all the noise. Our work focuses on a specific aspect and provides a theoretically guaranteed solution. We empirically showed that when this aspect dominates the noise (semi-synthetic and Volatility), our method performs significantly better. This provides a solid contribution and one step towards a better modeling of the real world noisy data.

---

> ### Author Response · Authors · 2022-12-04
> **Response to Reviewer 8f3z I**
>
> We thank the reviewer for considering our idea interesting. We truly appreciate the reviewer's suggestion on our theorem and empirical results. We will address the concerns in the following response and improve our manuscript accordingly.
>
> ### Concern Regarding Theorem
>
> __R0-Q1__.  Why not study joint risk bound like the one in Cortes et al.
>
> >__R0-Ans1__: Thank you for the insightful question. This indeed points towards one of the core challenges and contributions of our method. We will add these discussions in our paper.
>
> >First, we would like to point out that even if we can successfully apply the analysis framework in Cortes et al to our method, the risk bound  won’t be improved. The bound in Cortes et al has $1/\sqrt{n}$ generalization rate while our Theorem 1 already implies a better $1/n$ generalization rate, matching our information theoretic lower bound.
>
> >More importantly, the joint optimization from Cortes et al is not applicable here. Different from Cortes et al, our selector loss can not be viewed as a joint loss for a classifiers-selector pair $(f,g)$. The classifier $f$ and selector $g$ have to be learnt separately using different losses. Therefore, a joint risk bound similar to Cortes et al is not possible here. Let us elaborate below.
>
> >A major difference between our method and Cortes et al is due to the different motivations. In selective learning (Cortes et al), the selector pays a constant penalty for any unselected data in spite of the classifier $f$. In our problem, the selector tries to select data so that on the selected ones $f$ tends to be correct and on the unselected ones $f$ tends to be incorrect. If we convert our selector loss into a joint loss of $f$ and $g$, the loss will push $f$ to be consistently __incorrect__ on the unselected data ($\{x|g(x)<0\}$); essentially adding `negative’ weights to risks on unselected data. This is intrinsically contradictory to the risk minimization schema of $f$, and thus cannot be unified with the optimization of $f$. Note this is not an issue for the joint loss of Cortes et al, in which $f$ has zero risk on unselected data. As an analogy, the graph min-cut problem is polynomially solvable with non-negative edge weights, but NP-hard when the graph has negative edge weights.
>
> >Due to this constraint, the best we could do is an iterative optimization to repeatedly optimize the classifier and then optimize the selector. The analysis of the ‘joint’ risk bound in this iterative schema has to be decoupled into the risk bound of $\widehat{f}$ and the risk bound of $g$. This is indeed what we do in practical algorithms. Our theorem 1 is essentially a one-shot bound for the iterative algorithm. To extend the theorem to encompass all iterations can be extremely challenging, as will be discussed in R0-Q2. We would like to consider it as a separate paper in the future.
>
> >Finally, we’d like to clarify that a joint risk bound of $(\hat{f},\hat{g})$ is already naturally implied by the current theorem 1 by incorporating the empirical risk minimization (ERM) bound of $\widehat{f}$, as has been noted in the paper (see discussion in Remark 1 and 2)
>
>
> __R0-Q2__: "Even though a good predictor $\hat{f}$ may be achieved on the noisy data under some assumptions, couldn’t an even better predictor be learned on the data selected by $g$, that is on data that is ideally less noisy?"
>
> >__R0-Ans2__: Thank you for the suggestion. It is true that the risk can potentially be improved. However, it is highly challenging to prove a risk bound of $f$ for a $g$-reweighted loss that is better than the standard ERM bound. One has to consider the bias and variation of risk weighted  by $g$ and how they affect $f$’s estimation. We would like to emphasize that pursuing better risk bounds for classifier $f$ is not the focus of this work. Given the existing dense results on the novel noisy model formulation, highly non-trivial statistical limitation result  and minimax risk bound, we consider it justifiable to leave the work to the next paper.
>
> __R0-Q3__: "More detailed discussion on how a good predictor $\hat{f}$ can be achieved is needed. They claim that it could be achieved “under some margin condition” – are these additional assumptions or can it be part of assumption 1?"
>
> >__R0-Ans3__: Thank you for pointing this out. We will elaborate this in the paper. The ‘margin condition‘, e.g ,Massart noise condition, Tsybakov noise conditions, are not implied in our assumption 1. The margin condition is introduced mainly to achieve a fast generalization rate to match the lower bound. We note these conditions are very standard for analyzing ERM classifiers $f$. They are not due to our selector-classifier formulation.

---

> ### Author Response · Authors · 2022-12-09
> **Looking forward to further discussion**
>
> Dear Reviewer 8f3z,
>
> We really appreciate your time and effort in reviewing our paper. We have provided our response to your concerns in existing works, potential relaxation, clarification of empirical results. We hope to know if these responses addressed your concerns. If not, we wish we could have more discussion during the remaining openreview time so that we can improve our paper accordingly.
>
> Sincerely,
>
> Paper 4766 Author

---

### Author Response · Authors · 2022-11-16
**Overall Response**

We sincerely thank all reviewers for their time and valuable suggestions. We have improved our manuscript based on constructive feedbacks from reviewers (changes highlighted in blue). We will address each reviewer's concerns separately.

---

### Decision · Program_Chairs · 2023-01-20

**Decision:**

Reject

**Justification For Why Not Higher Score:**

Although there are some interesting and potential important theoretical results provided by this work, there is jump to get to the practical setting which is analyzed empirically.  Neither the theory or empirical studies are strong enough to stand on their own and their connection would need to be made stronger to truly be coherent.

**Justification For Why Not Lower Score:**

N/A

**Metareview: Summary, Strengths And Weaknesses:**

This work analyzes a learning with abstention setting, which assumes the data distribution is generated as a mixture of informative and uninformative points. Under this setting, the authors provide an "selector loss" to train an abstaining function "g" and, furthermore show minimizing the loss (in the presence of a good classifier "f") results in an abstainer that is near optimal.  Finally, the authors present a practical iterative algorithm to simultaneously train f and g and demonstrate that the proposed method provides benefits in the highly selective regime, i.e., when it is assumed a large number of examples are noisy and, thus, only a small fraction of points should receive predictions.

The AC would like to acknowledge and commend the authors detailed responses during the discussion phase, which includes additional empirical evidence. Although the submission does make valuable contributions, the shortcomings, when considered together, keep this paper just below the bar for publication at this venue.

(1) The theory is presented wrt fixed "f" and consider the optimization of "g", however, the practical algorithm optimizes both f and g in an alternating fashion.  As the authors say in their feedback, they present "one-shot bound for the iterative algorithm" and a bound on the iterative algorithm is significantly more challenging. But going from the single-shot to iterative setting is a big jump and fundamental questions, like "does the alternating optimization converge (and to what solution)?", are critical to understand in a theoretical work.

(2) If we take the algorithm as a heuristic inspired by the single-shot theory, then the expectation is to see a very thorough and compelling empirical analysis. The authors do show on several benchmarks that the suggested algorithm is superior to a few other baselines in settings where higher precision (at lower recall) is desired, in particular, when few predictions are required (low coverage setting). This is a useful observation, but it's not clear from the experiments (or theory) why that is the case, or if there is any way to adjust the balance of precision/recall for the different target coverage level settings.